# On Stable Long-Form Generation: Benchmarking and Mitigating Length Volatility

## Abstract

Large Language Models (LLMs) excel at long-context understanding but exhibit significant limitations in long-form generation. Existing studies primarily focus on single-generation quality, generally overlooking the volatility of the output (i.e., the inconsistency in length and content across multiple generations). This volatility not only leads to significant computational costs but also severely impacts the models' reliable application. To address this gap, our work unfolds in three stages: *benchmarking, probing, and mitigation*. We first propose the **VO**latility in **L**ong-form **T**ext **Bench**mark (**VOLTBench**), a novel heterogeneous-task benchmark designed to systematically quantify the length volatility of long-form generation. Subsequently, by analyzing attention traces, we conduct an in-depth probe to identify several common internal patterns that cause this volatility. Finally, to mitigate long-form output volatility, we propose SELB (Structural Enforcement via Logits Boosting), a lightweight decoding-stage optimization strategy, designed to significantly enhance both the length accuracy and stability of long-form generation without additional training. Extensive experiments on VOLTBench provide the first systematic confirmation of severe long-form output instability in mainstream models and validate that our proposed method successfully improves the mean output length of the base model by 148% and reduces the length volatility by 69%, while maintaining high generation quality.[1]

## 1 Introduction

Large Language Models (LLMs) have made significant advances in long-context processing Bai et al. (2023); GLM et al. (2024); Comanici et al. (2025), capable of handling inputs exceeding 100k tokens and performing precise information retrieval in Needle-in-a-Haystack tasks Yuan et al. (2025); Ye et al. (2025a); Zhou et al. (2025). However, this remarkable progress in long-context understanding has not extended to long-form generation. Their outputs struggle to surpass the 2k-word threshold Bai et al. (2024), while also lacking equivalent fine-grained control over the process.

Recent studies have benchmarked the long-form generation capabilities of models, typically employing unstructured content generation tasks such as story writing, and observed that current models generally struggle to meet target lengths accurately Liu et al. (2024); Zhang et al. (2025b); Wu et al. (2025b). Some work attributes this issue preliminarily to data-related factors, such as the scarcity of long-output examples in supervised fine-tuning (SFT) datasets Bai et al. (2024).

However, we argue that current research has three core limitations: First, existing work focuses almost exclusively on **single-generation results**, systematically overlooking output stability. This paradigm fails to capture the **significant volatility** that occurs when models process the same prompt multiple times, as shown in Figure 1, leading to unpredictable token consumption and high costs. Second, current benchmarks over-rely on **unstructured tasks** like story generation. Their subjective and difficult-to-automate evaluation criteria hinder the objective, quantifiable assessment of generation quality. In contrast, structured tasks with clear rules (e.g., code generation) offer a better environment for evaluation but remain underexplored. Finally, most research is limited to observing the phenomenon, **lacking an in-depth investigation** into the internal mechanisms.

---

[1]The code will be publicly available upon acceptance.

To address the aforementioned limitations, we conduct an in-depth, multi-stage investigation into the volatility of LLM long-form generation from three perspectives: *Benchmarking, Probing, and Mitigating.* First, on the benchmarking front, we introduce **length volatility** as a core metric and construct the Volatility in Long-form Text Benchmark (**VOLTBench**), a multi-dimensional, *heterogeneous-task benchmark* covering not only unstructured text (e.g., story) and structured data (e.g., code) but also dimensions such as different languages and instruction complexities. Through empirical evaluation on this benchmark, we provide the first large-scale quantification of the prevalent output length volatility in mainstream models. Second, in our probing efforts, we leverage these benchmark findings to conduct an in-depth analysis of the root causes

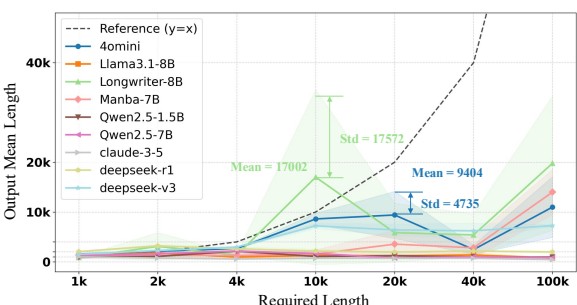

Figure 1: Model performance on our VOLTBENCH for long-text generation. As the required length increases, the actual output length of all models falls significantly short of the target (dashed line). Furthermore, many models exhibit significant output length volatility, even for Longwriter-8B, a model specifically fine-tuned on long text, *the output standard deviation peaked at 103% of its mean length*.

of this volatility. Moving beyond mere phenomenological observation, by analyzing the models' attention traces, we identify and define several common **internal patterns of length volatility**, such as *Attention Collapse* and *Attention Instability*. Finally, to mitigate the identified internal patterns, we propose and validate **Structural Enforcement via Logits Boosting (SELB)**, a lightweight, *decoding-stage method* that requires *no additional training* and proactively suppresses tokens linked to known failure modes, simultaneously improving both length accuracy and output stability. Our contributions are as follows:

- We construct the Volatility in Long-form Text Benchmark (VOLTBench), which is the first to introduce output volatility as a core metric. We systematically evaluate the long-form generation volatility in LLMs by covering both unstructured and structured tasks.

- We conduct extensive experiments that demonstrate the severe long-form output instability in mainstream LLMs. To investigate the underlying mechanisms, we identify and define several common internal patterns of length volatility through attention trace analysis.

- Targeting the identified internal patterns, we propose Structural Enforcement via Logits Boosting (SELB), which is a lightweight, decoding-stage optimization strategy that requires no additional training and improves the mean output length of the base model by 148% and reduces the length volatility by 69%, while maintaining high generation quality.

## 2 RELATED WORK

**Benchmarking Long-Form Generation**. Existing studies have revealed the limitations of current models in long-form generation from multiple dimensions. HelloBench Que et al. (2024) uses diverse in-the-wild scenarios, finds that even advanced models face severe repetition. LIFEBench Zhang et al. (2025b) shows that models struggle to adhere to precise length requirements. LongGen-Bench Liu et al. (2024) reformulates existing QA datasets to assess the logical consistency of a single, sequential long-form answer. LongInOutBench Zhang et al. (2025a) targets the gap in long-input, long-output tasks, while LongProc Ye et al. (2025b) requires models to create structured outputs from dispersed information. FACTS Grounding Jacovi et al. (2025) focuses on the factual accuracy of long responses against a source document, and ProxyQA Tan et al. (2024) uses an innovative proxy-question method to measure knowledge coverage. Meanwhile, works like LongGenBench Wu et al. (2025b) and LCFO Costa-jussà et al. (2025) further advance evaluations by introducing complex instruction-following in super-long texts. In contrast, our work specifically evaluates and addresses the phenomenon of Length Volatility, aiming to enhance the robustness and controllability of LLM long-text outputs. We provide a comparison between ours and previous studies in Table 1.

**Long-form Text Generation**. Research in long-form text generation addresses the challenge that LLMs struggle to produce high-quality, lengthy outputs. Data-centric approaches have been pro-

Table 1: Comparison with existing related benchmarks. VOLTBench provides a more comprehensive evaluation framework and is the first to introduce multiple sampling and stability evaluation.

| Benchmark | Instruction | | | Generation | | | | |
|---|---|---|---|---|---|---|---|---|
| | Multiple Task | Multiple Level | Multiple Language | Unstructured Text | Structured Data | Multiple Sampling | Stability Eval | Length Scale |
| **HELLOBENCH** Que et al. (2024) | ✓ | ✓ | | ✓ | | | | ∼ 16k |
| **LONGBENCH** Bai et al. (2024) | ✓ | | ✓ | ✓ | | | | ∼ 10k |
| **LONGGENBENCH** Liu et al. (2024) | ✓ | | | ✓ | | | | ∼ 8k |
| **LIFEBENCH** Zhang et al. (2025b) | ✓ | | ✓ | ✓ | | | | ∼ 8k |
| **LONGPROC** Ye et al. (2025b) | ✓ | | | | ✓ | | | ∼ 8k |
| **LONGGENBENCH** Wu et al. (2025b) | ✓ | | | ✓ | | | | ∼ 32k |
| **LONGINOUTBENCH** Zhang et al. (2025a) | ✓ | | | ✓ | | | | ∼ 16k |
| **VOLTBENCH (Ours)** | ✓ | ✓ | ✓ | ✓ | ✓ | ✓ | ✓ | ∼ 100k |

posed, such as using agentic plan-and-write pipelines Bai et al. (2024), creating multi-constraint instructions via backtranslation Pham et al. (2024), or enabling models to iteratively extend their own outputs Pham et al. (2024); Quan et al. (2024). LongWriter-Zero Wu et al. (2025a) uses reinforcement learning (RL) from scratch to foster long-generation capabilities. Wang et al. (2024) applies inference-time training with methods like Temp-Lora to maintain context in a temporary module. In contrast to prior work, which often involves extensive data creation or complex training, we propose a lightweight mitigation method based on the analysis of the model's internal attention to mitigate the instability and improve instruction adherence in long-form text generated by LLMs.

## 3 VOLTBENCH: BENCHMARKING THE LENGTH VOLATILITY

In this section, we introduce VOLTBench (Figure 2), a novel benchmark designed to systematically evaluate the stability and reliability of LLMs in long-form generation tasks. Its key features are as follows:

**Diverse and Challenging Instructions**: The foundation of VOLTBench lies in its multidimensional instruction set. Our instructions *span a wide array of tasks*, from creative unstructured writing (e.g., stories) to logical structured generation (e.g., code libraries), pushing models beyond simple narrative generation. Each task is *presented with varying levels of complexity*, including simple prompts, detailed contextual instructions,

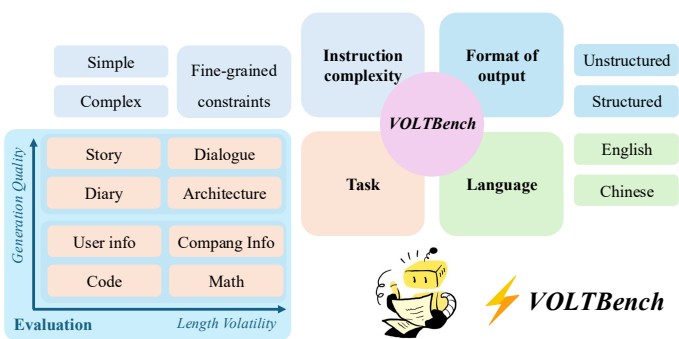

Figure 2: An overview of the VOLTBench framework. Our benchmark is constructed from four dimensions, covering structured and unstructured tasks. We evaluate performance from two aspects: generation quality and length volatility.

and challenges with highly specific fine-grained constraints to test meticulous instruction-following over long contexts. Furthermore, to assess linguistic robustness, all instructions are *provided in parallel English and Chinese* versions, enabling a direct and fair comparison of model performance across different languages.

**Versatile and Scalable Generation**: Corresponding to the input diversity, VOLTBench evaluates a generation space notable for its versatility and scale. A key distinction of our benchmark is the dual focus on both *unstructured text and complex structured data outputs*, such as complete codebases. We implement this structure through a chapter-based format, which requires models to generate hierarchically organized content. Chapter-based design is the key to our scalability, enabling us to create instructions that range from a concise *5-chapter document to an expansive 500-chapter tome*. This pushes models to their operational limits with an unprecedented length scale of up to 100k

words. This massive and explicitly sectioned scale is specifically intended to surface and analyze challenging failure modes.

**Generation Volatility and Quality Evaluation**: The cornerstone of VOLTBench is its rigorous evaluation of both generation volatility and quality, moving beyond single-instance assessments to measure model reliability. We query a model multiple times for each instruction to create a distribution of outputs. We assess stability at both a macro level, analyzing overall length volatility, and a granular chapter-by-chapter level, checking for consistency within each section. This fine-grained analysis can reveal nuanced behaviors, such as a model starting strong but losing steam in later chapters. VOLTBench embeds fine-grained constraints (e.g., keyword, topic) into its prompts. This innovative design allows us to *automate quality assessment* even for unstructured narrative tasks, as we can programmatically verify if these specific constraints were met. This is complemented by our structured data generation tasks, where quality is assessed objectively via Execution-based Verification, thus providing a far more reliable and multi-faceted quality evaluation framework.

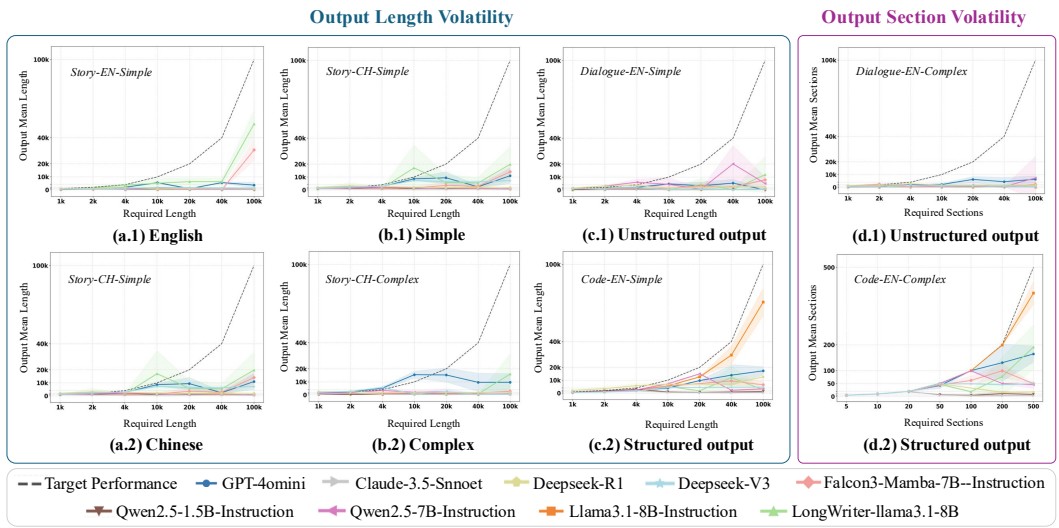

Figure 3: Analysis of Output Length Volatility and Output Section Volatility. The left panel (a, b, c) compares the total output length volatility across three dimensions: language, instruction complexity, and output format. The right panel (d) shows the volatility in the number of generated sections.

### 3.1 TASKS

Our benchmark includes both unstructured and structured generation tasks. Each core task is expanded into multiple variants across three dimensions: language (*English/Chinese*), instruction complexity (*simple, complex, fine-grained constrain*), and output length (*from 5 to 500 chapters*). This multi-dimensional design precisely measures fluctuations in model performance under diverse conditions (see Appendix J.0.7 for all task instructions).

**Unstructured Tasks**: This category of tasks evaluates a model's creativity, narrative coherence, and contextual consistency in long-form, free-form text. We include diverse scenarios such as Story, Dialogue, Diary, and Architecture to assess abilities ranging from plot development and maintaining a consistent persona to the creative use of specialized terminology. Below is an example:

> **Task**: Story
> **Label**: English-Simple-$M$ chapters-$N$ words
> **Instruction**: Please write a novel consisting of $M$ chapters about Jeff. Each chapter should revolve around a theme or plot, with a minimum of $N$ words for each chapter. Ensure clarity and continuity ... and use '*** Finished ***' to indicate the end of the document.

**Structured Tasks**: These tasks assess models' ability to follow strict formatting, syntax, and logical rules where precision is key. Tasks like generating virtual company profiles, Python function libraries, and mathematical formulas are designed for objective, automated evaluation of a model's reasoning and mastery of formal languages. Full instructions are in Appendix J.0.7.

## 3.2 EVALUATION METRIC

Our benchmark evaluates models' long-text generation capabilities across two core dimensions:

**Length Volatility**. Unlike previous work Zhang et al. (2025b), which focuses on the volatility of a single generation, we measure a model's volatility across multiple outputs.

(1) **Length Standard Deviation (LSD)**, this metric measures the *absolute volatility* of the output lengths: $\text{LSD} = \sqrt{\frac{1}{N}\sum_{i=1}^{N}(L_i - \mu)^2}$, where $\mu$ is the average of the $N$ output lengths. In our experiments, we set N=5.

(2) **Length Variation Coefficient (LVC)**, this measures the *relative volatility* of the output lengths with respect to their mean, which allows for comparable stability assessments across different length requirements: $\text{LVC} = \frac{LSD}{\mu}$.

(3) **Mean Length Accuracy (MLA)**, this metric quantifies how closely the mean length ($\mu$) of N generation runs adheres to the specified target length ($L_{\text{constraint}}$). The formula is: $\text{MLA} = \max\left(0, 1 - \left|\frac{\mu - L_{\text{constraint}}}{L_{\text{constraint}}}\right|\right) \times 100$.

**Generation Quality**. We assess the quality of the generated content from the following aspects:

(1) **Format Adherence Deviation (FAD)**, which measures the absolute volatility in the number of generated chapters across multiple runs for chapter-based tasks. It assesses if the model consistently produces the required number of chapters: $\text{FAD} = \sqrt{\frac{1}{N}\sum_{i=1}^{N}(C_i - \mu_c)^2}$, where $C_i$ is the number of chapters in the $i$-th generation, and $\mu_c$ is the average chapter count over $N$ runs.

(2) **Structured Content Accuracy (SCA)**, this metric uses Execution-based Verification to assess accuracy on structured tasks, such as generating Python libraries and LaTeX formulas: $\text{SCA} = \frac{\text{Number of Correct Chapters}}{\text{Number of Required Chapters}}$.

(3) **Unstructured Content Accuracy (UCA)**, following previous work Bai et al. (2024); Zhang et al. (2025a), we use an LLM-as-a-Judge to evaluate unstructured tasks (e.g., story writing), with details in Appendix C.

## 4 EXPERIMENTS AND RESULTS

### 4.1 MODELS

To systematically evaluate long-text generation capabilities, our study includes a diverse set of models. Specifically, we evaluate reasoning models such as GPT-4o mini, Claude 3.5 Sonnet, and Deepseek-R1 (DeepSeek-AI et al., 2025a). Our open-source selection includes models of various architectures and sizes: Qwen2.5-1.5B-Instruction, Qwen2.5-7B-Instruction (Qwen et al., 2025), Qwen3-8B (Team, 2025), Llama3.1-8B-Instruction, Deepseek-V3 (DeepSeek-AI et al., 2025b). We also include Falcon3-Mamba-7B–Instruction (Team, 2024), notable for its distinct architecture. We also include LongWriter-llama3.1-8B (Bai et al., 2024), a model enhanced for long-form generation via long-text post-training. Additionally, we incorporate common training-free decoding strategies for comparison, implemented on Qwen2.5-7B-Instruction. These include **Repetition Penalty** to mitigate text degeneration via logit penalization, **Entropy-Based Stopping** employing predictive uncertainty as a dynamic termination criterion, **Length Constraint** for enforcing explicit output boundaries, and **Lookahead Decoding**, designed to optimize the generation trajectory by anticipating future probabilities.

### 4.2 FINE-GRAINED CONSTRAINTS

To evaluate a model's ability to follow specific, localized instructions in long-form generation, we designed a framework using fine-grained constraints. This approach tests content control at a sub-document level, unlike typical global prompt-following evaluations. Specifically, we apply three distinct and simultaneous constraints to designated sections of the output. The constraints are defined as follows:

Table 2: Performance comparison of evaluated models on a 100-section generation task, conducted in English under simple difficulty settings. Representative results are shown for an unstructured task (Story) and a structured task (Code Function). For the LSD and FAD metrics, the values in parentheses provide context by showing the generated mean length (in words) and mean section count, respectively. The "±" values represent the standard deviation. The arrows (↑/↓) indicate whether higher or lower values are preferable for each metric.

| Model | Length Volatility | | | Generation Quality | | |
|---|---|---|---|---|---|---|
| | LSD (↓) | LVC (↓) | MLA (↑) | FAD (↓) | SCA (↑) | UCA (↑) |
| GPT-4o mini | 325.65 (959) | 33.9% | 4.8% | 1.41 (7.00) | 84.6% (±30.8%) | 86.7% (±6.7%) |
| Claude-3.5-Sonnet | 3.30 (176) | 1.9% | 0.9% | 0.00 (2.00) | 3.0% (±0.0%) | 88.7% (±2.7%) |
| Deepseek-R1 | 103.30 (1198) | 8.6% | 6.0% | 1.25 (4.33) | 35.0% (±13.2%) | 93.3% (±3.7%) |
| Deepseek-V3 | 40.76 (1854) | 2.2% | 9.3% | 1.70 (20.67) | 48.6% (±3.8%) | 84.7% (±3.4%) |
| Mamba-7B | 715.98 (1291) | 55.5% | 6.5% | 41.72 (40.75) | 66.8% (±21.9%) | 76.0% (±17.3%) |
| Qwen2.5-1.5B | 27.78 (142) | 19.6% | 0.7% | 0.47 (1.67) | 15.6% (±24.0%) | 84.0% (±7.1%) |
| Qwen2.5-7B | 75.87 (445) | 17.0% | 2.2% | 2.05 (10.33) | 99.8% (±0.4%) | 86.7% (±7.6%) |
| Llama3.1-8B | 92.77 (350) | 26.5% | 1.7% | 0.94 (4.33) | 92.4% (±14.2%) | 82.0% (±18.9%) |
| LongWriter-8B | 2866.3 (6320) | 45.4% | 31.6% | 21.42 (45.00) | 32.6% (±31.9%) | 66.7% (±16.5%) |
| Repetition Penalty | 553 (2967) | 18.6% | 14.8% | 5.4 (22) | 98% (±1%) | 76.7% (±14.5%) |
| Entropy-Stopping | 713 (2701) | 26.4% | 13.5% | 7.24 (24) | 95% (±2.5%) | 83.9% (±8%) |
| Length Constraint | 1280 (4470) | 28.65% | 22.4% | 9.2 (28) | 96% (±2%) | 85% (±9%) |
| Lookahead Decoding | 268 (2883) | 9.3% | 14.4% | 7.2 (25) | 94% (±3.5%) | 84.4% (±8%) |

- *Character-level Pattern Constraint:* This constraint dictates that the first word of a target section must begin with a pre-determined, randomly selected alphabetical character. This tests the model's ability to control low-level textual attributes.

- *Keyword Presence Constraint:* This requires the mandatory inclusion of a specific, randomly selected keyword within the body of a target section. This evaluates the model's capacity to track and insert specific information into relevant contexts.

- *Specified Theme Constraint:* This imposes a thematic requirement, compelling the narrative or content of a target section to align with a randomly selected topic or scenario. This assesses the model's ability to generate coherent content based on a high-level concept.

## 4.3 RESULTS AND ANALYSIS

**Volatility Across Different Dimensions**. As shown in Figure 3, we analyze model performance across three dimensions. On the language dimension, most models exhibit lower volatility and a greater mean output length in 5 runs when generating in English. Regarding instruction complexity, models produce longer outputs for simple instructions, likely due to greater creative freedom, which is also accompanied by higher volatility. In terms of output format, we observe an interesting trend where models generate longer and more stable text (i.e., less volatile) for structured tasks. We attribute this to structured tasks being governed by well-defined format constraints and internal logic, which provides stronger guidance for the generation process. This hypothesis is corroborated by Figure (d.2), which shows that models generally generate a greater number of sections for structured tasks. For complete experimental results and analysis, please refer to Appendix J.

**Long Text Quality Evaluation**. For comparison, we exclude Claude-3.5-Sonnet due to its low mean length (176 words), insufficient for long-text evaluation. For other models, we assess generation quality and actual length, revealing distinct trade-offs. As shown in Table 2, GPT-4o-mini showed the best balance on structured tasks among longer-output models, with SCA 84.6%, low FAD, and 959-word output. LongWriter-8B generated the longest text (6320 words) but scored low on both SCA (32.6%) and FAD (21.42), indicating a quality–length trade-off. On unstructured tasks, Deepseek-R1 achieved the highest UCA (93.3%) with 1198 words, while LongWriter-8B again scored lowest (66.7%), prioritizing length over quality. In summary, all current models fail to jointly satisfy long-text length and high-quality generation.

**Generation Patterns of Length Volatility** Our experiments reveal that baseline models consistently struggle with length and structural constraints in long-form generation. The failure rate is stark: when tasked with generating up to 50 sections, models failed in approximately half of the cases. For requests exceeding 50 sections, all models failed to complete the task as instructed. These failures typically manifest in two primary patterns:

- *Incomplete Generation*: Models frequently produce significantly less content than instructed. For example, when tasked with generating 40 sections, a model might stop after only 10. This premature termination, whether silent or reverting to a persona, with outputs like "I hope these sections are helpful." We hypothesize this latter behavior occurs when the generated text exceeds the context window, pushing the original prompt out of scope and causing the model to default to its base assistant persona.

- *Section Skipping*: In other instances, models demonstrate erratic adherence to the requested structure. A model might generate the first several sections sequentially and then abruptly jump to the final section, omitting all intermediate content.

### 4.3.1 Analysis of Fine-grained Constraint Following

To provide a quantitative view of the volatility in instruction adherence, we analyze model performance on the fine-grained constraint tasks. The complete results, including figures for all three constraint types, can be seen in Appendix D.

As depicted in the figure, a clear trend emerges across all tested models. While most models, such as Deepseek-R1, Qwen3-8B and LLama3.1 adhere to constraints on shorter tasks (5-50 sections), their performance plummets and grows more volatile as the context length increases. This trend is universal, starkly contrasting the better models with Longwriter, which fails entirely regardless of length. Critically, even for the top models, the success rate flattens after the 100-section mark, and then actively collapses—with Qwen3-8b and LLama3.1 producing fewer correct sections at 500 than at 200. The systemic failure is most evident at the 500-section task: against a requirement of 100 constrained sections, no model delivered more than 40. This demonstrates a profound inability of current models to track and execute instructions deep within long-form generation.

## 5 Attention Traces Behind Volatility

**Attention Trace.** To explore the root of output volatility, we analyze the attention mechanism in generation. Building on Li et al. (2025), who link attention to constraint tokens with instruction-following ability, we extend this to long-form generation. We hypothesize that attention fluctuations toward input constraints correlate with output variability. At each step $t$, where $t \geq 1$, the model attends to prompt tokens $x_{1:T_0}$ and generated tokens $y_{0:t-1}$, where $T_0$ indicates the length of prompt tokens. We focus on attention to constraint-encoding tokens in $x_{1:T_0}$. For layer $l$ and head $n$, attention uses query $Q_n^{(l,t)}$ from $\mathbf{h}_{t-1}^{(l)}$ (last generated token's hidden state) and keys $K_n^{(l,t)}$ from $\mathbf{h}_{1:T_0+t-1}^{(l)}$ (hidden states of all prior tokens). The scaled dot-product attention weights $A_n^{(l,t)}$ are then calculated as $A_n^{(l,t)} = \text{softmax}\left(\frac{Q_n^{(l,t)} K_n^{(l,t)\top}}{\sqrt{d_k}}\right)$ where $d_k$ is the dimension of the key vectors. These weights are then averaged across all $N$ attention heads to obtain the layer-level attention vector $a^{(l,t)} = \frac{1}{N} \sum_{n=1}^{N} A_n^{(l,t)}$.

To measure attention directed toward constraints, we first identify the prompt token indices corresponding to each textual constraint $r \in R$, denoted as $C_r$. The full set of constraint token indices is given by $C = \bigcup_{r \in R} C_r$. The layer-step constraint attention $\alpha^{(l,t)}$ is then defined as the average attention from token $y_t$ to all tokens in $C$, i.e., $\alpha^{(l,t)} = \frac{1}{|C|} \sum_{j \in C} a_j^{(l,t)}$, where $a_j^{(l,t)}$ is the attention weight at layer $l$ and step $t$ directed to the $j$-th token of the input. Finally, we average $\alpha^{(l,t)}$ across all $L$ layers of the model to obtain a unified measure of constraint attention at each generation step, $\overline{\alpha}^{(t)} = \frac{1}{L} \sum_{l=1}^{L-1} \alpha^{(l,t)}$. By plotting the trace of $\overline{\alpha}^{(t)}$ during generation, we visualize how attention to constraints evolves. Peaks and subsequent drops, "attention summits", may signal points where reduced constraint focus leads to task deviation and output volatility. To analyze this, we gener-

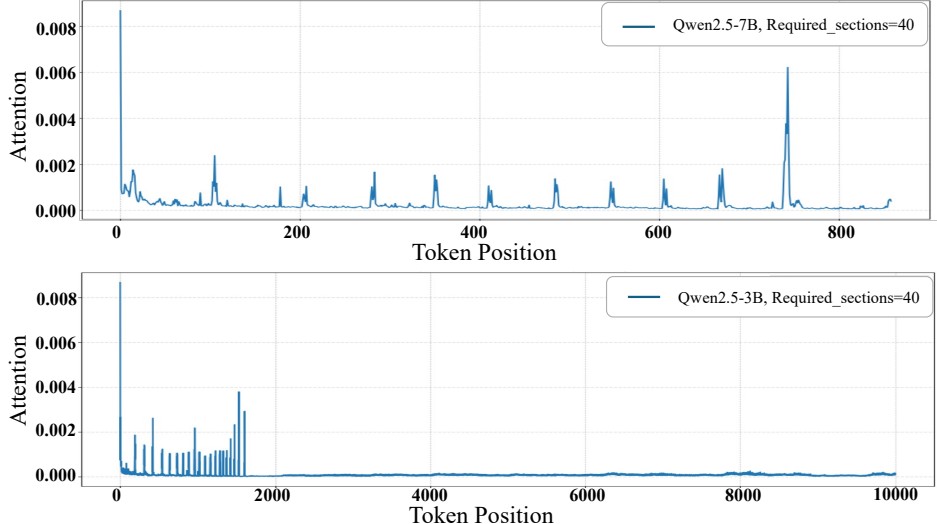

Figure 4: Attention traces for Qwen2.5-7B (top) and Qwen2.5-3B (bottom) models in a long-form diary generation task with 40 required sections. Each peak in the traces indicates the initiation of a new section. It is evident that both models failed to meet the requirement: Qwen2.5-7B bypassed intermediate sections and proceeded directly from an early section to the final one, while Qwen2.5-3B generated repetitive text after its final attention peak.

ate outputs with different random seeds and compare their attention traces to reveal links between attention dynamics and output variability.

**Internal Patterns of Length Volatility.** We analyze the attention trace $\overline{\alpha}^{(t)}$, which reveals internal patterns directly correlated with the earlier generation failures. As shown in Figure 2, where models are tasked with generating 40 sections, the traces highlight early internal signs of output volatility. From these, we identify two primary failure signatures: (1) *Attention Collapse*: This pattern aligns with premature termination or task abandonment. The Qwen2.5-3B trace illustrates this clearly: in the first 1,500 tokens, the model shows periodic attention spikes and follows instructions with well-structured content. After that, attention collapses to near-zero, signaling loss of focus on prompt constraints and resulting in halted or irrelevant output; (2) *Attention Instability*: This pattern corresponds to erratic behaviors such as section skipping. In Qwen2.5-7B, initial regular attention spikes align with successful section generation. Around token 750, an abnormally large spike disrupts this pattern, immediately preceding the model's deviation from sequential output. In both cases, periodic attention spikes function as essential refocusing signals that help maintain task coherence across sections. Analysis of the $\overline{\alpha}^{(t)}$ trace supports our hypothesis: the output volatility is not random but closely linked to and preceded by measurable failures in the model's internal attention dynamics.

## 6 MITIGATING LENGTH VOLATILITY

To mitigate generation volatility, we propose a dynamic decoding strategy that ensures stable, constraint-abiding outputs via single-pass generation. Rather than iterative prompts or multiple model calls, we modify logits in real time. At each step $t$, the model outputs a logit vector $s_t \in \mathbb{R}^{|V|}$ over the model's vocabulary $V$, which are adjusted by a guidance function $M$. Unlike standard decoding, $M$ modifies logits based on context and rules to enforce structural and constraint adherence. Formally, given the prompt tokens $x_{1:T_0}$ and the generated token sequence $y_{0:t-1}$ up to step $t-1$, the modified logit vector $s'_t$ is computed as:

$$s'_t = M(s_t, [x_{1:T_0}; y_{0:t-1}]). \tag{1}$$

The function $M$ combines two guidance components: *structural enforcement*, which enforces adherence to the desired output structure, and *proactive failure prevention*, which applies a prohibitive negative bias to suppress likely failure modes during generation.

## 6.1 STRUCTURAL ENFORCEMENT VIA LOGITS BOOSTING

To ensure generation of $P_{total}$ sections, we force a new section whenever the current section reaches the target length $\tau_{max}$. If the length of $p$-th section $\tau_p \geq \tau_{max}$, a strong positive bias $\beta$ is applied to the logits of tokens corresponding to the next section title, $V_{\text{title}}^{(p+1)} \subset V$. The structural boosting adjustment, $M_{\text{struct}}$, is then defined as:

$$s'_{t,j} = \begin{cases} s_{t,j} + \beta & \text{if } \tau_p \geq \tau_{max} \wedge p < P_{total} \wedge j \in V_{\text{title}}^{(p+1)} \\ s_{t,j} & \text{otherwise,} \end{cases} \quad (2)$$

where $s_{t,j}$ is the logit for token $j$ at step $t$, and $\beta$ is a large positive constant that makes the selection of a title token nearly certain. Once a token from $V_{\text{title}}^{(p+1)}$ is generated, the section index is incremented ($p \leftarrow p + 1$) and the counter is reset ($\tau_p \leftarrow 0$).

## 6.2 PROACTIVE FAILURE PREVENTION

Based on our analysis of generation patterns, we proactively suppress tokens associated with known failure modes by applying a strong negative bias during decoding. Formally, let $V_{\text{banned}} \subset V$ be the set of token indices corresponding to conversational filler phrases (e.g., "I hope these..."); and let $v_{\text{eos}}$ be the index of the end-of-sentence token. The failure prevention function $M_{\text{fail}}$ is defined as:

$$s'_{t,j} = \begin{cases} -\infty & \text{if } j \in V_{\text{banned}} \\ -\infty & \text{if } j = v_{\text{eos}} \wedge p < P_{\text{total}} \\ s_{t,j} & \text{otherwise.} \end{cases} \quad (3)$$

This prevents undesirable conversational text and early termination before the final section. By composing $M = M_{\text{fail}} \circ M_{\text{struct}}$, our method enables real-time control over generation, directly managing output probabilities to address length volatility while ensuring structural and constraint adherence in a single pass.

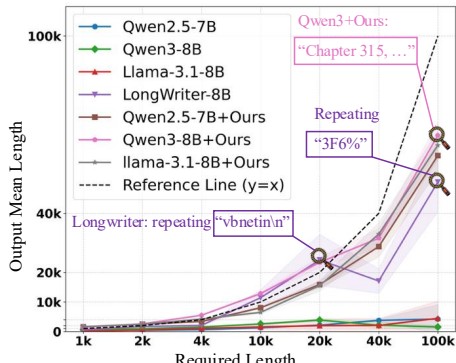

Figure 5: Model output length volatility (Story Writing). While baseline models like Longwriter often inflate length with meaningless repetition, our method accurately matches the target length while maintaining coherent content. Section volatility is presented in Figure 20.

## 6.3 RESULTS

Our method marks a major improvement in long-text generation, outperforming strong baselines like LongWriter-8B in stability, adherence, and quality. Evaluation was done on a 100-section task under simple settings. In output stability and length adherence, our model excels. As shown in Figure 6, its mean length and section count closely follow the reference line, unlike baselines that degrade as complexity rises. The Length Variation Coefficient (LVC), where lower is better, for our model is $14.02\%$, a $69\%$ reduction in volatility compared to $45.4\%$ for LongWriter-8B. Furthermore, our model's Mean Length Accuracy (MLA) is $78.25\%$, more than double the $31.6\%$ achieved by LongWriter-8B, indicating a much closer adherence to the required length. This is reflected in the average output of 15,651 words from our model, compared to just 6,320 from LongWriter-8B and less than 1000 in other models. Our model also achieves higher generation quality. For Structured Content Accuracy (SCA), our model scored a perfect $100\%$, dramatically better than LongWriter-8B's $32.6\%$, which has plenty of repeated tokens. To quantify this, we further analyze the lexical diversity in Appendix G, showing that our method significantly reduces n-gram repetition rates and improves the Type-Token Ratio (TTR) compared to baselines. This highlights its enhanced capability in handling structured tasks. Similarly, for Unstructured Content Accuracy (UCA), our model scored $86.7\%$, a $30\%$ improvement over LongWriter-8B. These results underscore our method's ability to generate not only longer and more stable text but also higher-quality. Beyond surface-level metrics, we investigate the underlying mechanism of this stability in Appendix H. Through Representational Stability Analysis, we demonstrate that SELB effectively mitigates the 'representational drift' of hidden states, preventing the semantic collapse commonly observed in baseline models during long generation.

## 6.4 GENERALIZATION TO FREE-FORM GENERATION

We extend its applicability to free-form generation tasks (e.g., continuous novel writing) where such explicit anchors are absent. In Appendix I, we detail the adaptation of our approach into a SELB-Hybrid strategy. This mechanism addresses the twin challenges of premature termination and generation loops by dynamically shifting from section enforcement to length enforcement. Specifically, it incorporates an aggressive Stop Token Suppression module that prohibits early exit phrases and a Hybrid Keep-Alive mechanism. The latter monitors generation checkpoints, if a stall or repetitive loop is detected within a grace period, it proactively boosts generic continuation tokens to break the cycle and sustain narrative flow. The empirical impact of this adaptation is substantial. We evaluated the method on extreme-length free-form tasks, such as writing a 20,000-word novel. As detailed in Appendix I, baseline models including GPT-4o-mini and LongWriter-8B suffered from severe length collapse, often generating fewer than 600 words despite the 20k target. In contrast, our SELB-Hybrid method achieved a Mean Length Accuracy (MLA) of 97% with a remarkably low Length Variation Coefficient (LVC) of 12.1%. These results confirm that our logits-boosting paradigm can be effectively generalized beyond structured tasks to enforce stability in unstructured, open-ended generation scenarios.

## 7 CONCLUSION

In this work, we investigate the critical yet overlooked issue of output volatility in long-form LLM generation. Our findings show that instability across multiple outputs poses a major challenge to reliable application. To systematically study this problem, we first introduce VOLTBench, a novel benchmark to quantify length volatility across diverse tasks. By probing internal attention mechanisms, we identify common patterns that drive instability. Based on these insights, we propose SELB (Structural Enforcement via Logits Boosting), a lightweight, training-free decoding strategy to directly mitigate this issue. Extensive experiments confirm that severe output volatility is widespread in mainstream models and validate our approach, which improves the base model's mean output length by 148% and reduces length volatility by 69%, while maintaining generation quality.

## REPRODUCIBILITY STATEMENT

Our work addresses the output volatility in long-form text generation through a three-stage approach: benchmarking, probing, and mitigation. This includes three main contributions: (1) the VOlatility in Long-form Text Benchmark (VOLTBench); (2) an in-depth analysis of the internal causes of volatility; and (3) a lightweight decoding-stage optimization strategy, SELB. To ensure the full reproducibility of our findings, we have provided detailed documentation in the paper and its appendices. The construction methodology, data composition, and evaluation metrics for VOLTBench are thoroughly described in Section 3. The complete implementation details for our proposed SELB method and the full experimental setup, including all hyperparameters, are provided in Section 6. We commit to releasing the entire source code, the full VOLTBench benchmark, and our analysis scripts to the public upon acceptance of this paper to facilitate verification and future research.

## ETHICS STATEMENT

Our research adheres to the standard ethical guidelines for academic publishing. The work presented in this paper is foundational, focusing on the technical challenges of output volatility in Large Language Models. Our objective is to improve the reliability and stability of these models, which is a positive contribution to the field of artificial intelligence. The proposed benchmark, VOLTBench, is constructed from publicly available datasets and does not contain any personally identifiable or sensitive information. Our research did not involve human subjects, and we foresee no direct negative societal impacts from this work.

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

## A    LLM USAGE

This paper addresses the challenge of output volatility in the long-form generation of Large Language Models (LLMs). We introduce VOLTBench, a novel benchmark to quantify this instability, conduct an in-depth analysis of its underlying causes, and propose SELB (Structural Enforcement via Logits Boosting), a lightweight decoding-stage strategy to mitigate the issue. In the preparation of this manuscript, we utilized Large Language Models (e.g., Google's Gemini) as a general-purpose writing assistant. The scope of the LLM's assistance was strictly confined to language-level refinements. This included several specific functions: identifying and correcting grammatical and syntactical errors; suggesting alternative phrasing to improve sentence flow and coherence; enhancing vocabulary for greater precision and academic tone; and paraphrasing sentences written by the authors to improve readability.

## B    TASK INSTRUCTION

The following are the prompts used in our experiment.

---

**Instruction: English Simple Story Generation**

Please write a novel consisting of {num_section} chapters. Each chapter should revolve around a theme or plot, with a minimum of {word_section} words for each chapter. Ensure clarity and continuity without any interruptions or omissions in the narrative throughout the document. Do not stop generating content until all {num_section} chapters are completed and '*** finished ***' is used to indicate the end of the document.

*** started ***
#*# Title:

---

Table 3: An example of the instructional prompt for the English simple story generation task. This template specifies parameters like the number of chapters and minimum word count, guiding the structure of the generated narrative.

---

**Instruction: English Simple Dialogue Generation**

Please generate {num_section} rounds of dialogue between customers and customer service. Each round should include a customer's question and a customer service representative's response, with a minimum of {word_section} words for each round. Ensure clarity and continuity without any interruptions or omissions in the narrative throughout the document. Do not stop generating content until all {num_section} rounds of dialogue are completed and '*** finished ***' is used to indicate the end of the document.

*** started ***
#*# Round 1: **customers**:

---

Table 4: The prompt for generating simple dialogues between a customer and customer service, specifying the number of rounds and word count.

---

**Instruction: English Simple Diary Generation**

Please write a diary for {num_section} days for Jeff. Each entry should include the date and a brief description of the content, with a minimum of {word_section} words for each entry. Ensure clarity and continuity without any interruptions or omissions in the narrative throughout the document. Do not stop generating content until all {num_section} diaries are completed and '*** finished ***' is used to indicate the end of the document.

*** started ***
#*# Date: Day 1:

---

Table 5: The prompt for generating simple diary entries for a character named Jeff, specifying the number of days and word count.

---

**Instruction: English Simple Architecture Design**

Please design a {num_section}-story building. Describe the function or layout of each floor, with at least {word_section} words for each layer. Ensure clarity and continuity without any interruptions or omissions in the narrative throughout the document. Do not stop generating content until all {num_section} floors are completed and '*** finished ***' is used to indicate the end of the document.

*** started ***
#*# Floor 1:

---

Table 6: The prompt for designing a multi-story building with simple functional descriptions for each floor.

---

**Instruction: English Complex Story Generation**

Please write a fantasy novel with {num_section} chapters about Jeff. The novel should have a clear theme and structure, with characters experiencing multiple twists and personal growth throughout the plot. Each chapter should describe the main characters' actions, thoughts, and emotional development, while also incorporating relevant background information (such as historical context, social environment, etc.). Each chapter should be around {word_section} words, with enough detail and emotional depth to keep the reader engaged. Ensure clarity and continuity without any interruptions or omissions in the narrative throughout the document. Do not stop generating content until all {num_section} chapters are completed and '*** finished ***' is used to indicate the end of the document. Do not output other characters to stop.

*** started ***
#*# Chapter1:

---

Table 7: The prompt for generating a complex fantasy novel, detailing requirements for plot, character development, and emotional depth.

---

**Instruction: English Complex Diary Generation**

Please write a diary for {num_section} days. Your name is Jeff, a white-collar worker. Each entry can include aspects such as your mood for the day, key events, challenges faced, solutions, and hopes or reflections for the future. Ensure that each diary entry expresses different emotions and reflects various life events and growth experiences. The diary content can cover a range of life scenarios, such as work, family, friends, health, and travel. Each entry should be around {word_section} words. Ensure clarity and continuity without any interruptions or omissions in the narrative throughout the document. Do not stop generating content until all {num_section} chapters are completed and '*** finished ***' is used to indicate the end of the document. Do not output other characters to stop.

*** started ***
#*# Date: Day 1

---

Table 8: The prompt for generating complex and emotionally rich diary entries, covering various life scenarios and personal growth.

---

**Instruction: English Complex Dialogue Generation**

Please generate {num_section} rounds of dialogue between customers and customer service. Each round of dialogue should include the customer's question and the customer service representative's response, along with service recommendations or solutions. These dialogues can cover multiple industries and scenarios, with each turn of conversation being non-contiguous and the scenes able to switch, such as in electronic product support, travel booking, financial services, and customer complaint handling. Each round should reflect different emotional changes, with the customer possibly exhibiting emotions like anxiety, confusion, anger, or happiness, while the customer service responses should appropriately provide reassurance, explanations, or solutions based on the customer's emotional state. Each round of dialogue should contain at least {word_section} words. Ensure clarity and continuity without any interruptions or omissions in the narrative throughout the document. Do not stop generating content

---

until all {num_section} rounds of dialogue is completed and '\*\*\* finished \*\*\*' is used to indicate the end of the document. Do not output other characters to stop.

\*\*\* started \*\*\*
#\*# Round 1 Customer:

Table 9: The prompt for generating complex customer service dialogues across various industries, focusing on emotional changes and appropriate responses.

**Instruction: English Complex Architecture Design**

Please design a {num_section}-story mixed-use skyscraper for work and living. Describe the function or layout of each floor. Each floor should have a different function and design, closely connected to other floors. Include detailed descriptions of office areas, commercial spaces, residential areas, and entertainment and leisure zones. The content should have sufficient detail and depth, such as design concepts, layouts, and unique elements (like floor decoration styles, space utilization, and the application of smart technology) to present a multifunctional building. Each floor's description should be at least {word_section} words. Ensure clarity and continuity without any interruptions or omissions in the narrative throughout the document. Do not stop generating content until all {num_section} floors are completed and '\*\*\* finished \*\*\*' is used to indicate the end of the document. Do not output other characters to stop.

\*\*\* started \*\*\*
#\*# Floor 1:

Table 10: The prompt for designing a detailed mixed-use skyscraper, requiring descriptions of design concepts, layouts, and unique features for each floor.

**Instruction: GenData - Simple Code Function**

Please generate a complete library of {num_section} different functions. Each function should include the function name, parameters, return type, and function comments, formatted in Python. Ensure clarity and continuity without any interruptions or omissions in the narrative throughout the document. Do not stop generating content until all {num_section} functions are completed and '\*\*\* finished \*\*\*' is used to indicate the end of the document.

\*\*\* started \*\*\*
# Function 1: Calculate the area of a circle, given the radius

```python
def calculate_area(radius):
    """
    This function calculates the area of a circle given its radius.
    Parameters:
        radius (float): The radius of the circle.
    Returns:
        float: The area of the circle.
    """
    return 3.14159 * radius ** 2
```

Table 11: The prompt for generating a library of simple Python functions with comments and examples.

**Instruction: GenData - Simple User Info**

Please generate {num_section} virtual user profiles, with each user's information including name, age, gender, address, email, and phone number, formatted as JSON. Ensure clarity and continuity without any interruptions or omissions in the narrative throughout the document. Do not stop generating content until all {num_section} profiles are completed and '\*\*\* finished \*\*\*' is used to indicate the end of the document.

```
*** started ***

[{
  "index": 1,
  "name": "John Doe",
  "age": 30,
  "gender": "Male",
  "address": "1234 Elm Street, Springfield, IL, 62701",
  "email": "johndoe@example.com",
  "phone": "+1-555-123-4567"
}]
```

Table 12: The prompt for generating simple virtual user profiles in JSON format.

**Instruction: GenData - Simple Company Info**

Please generate {num_section} virtual company profiles. Each profile should include the company name, industry, year of establishment, company address, and contact number, formatted in JSON. Ensure clarity and continuity without any interruptions or omissions in the narrative throughout the document. Do not stop generating content until all {num_section} virtual company profiles are completed and '*** finished ***' is used to indicate the end of the document.

```
*** started ***

[{
  "index": 1,
  "company_name": "Tech Innovations Inc.",
  "industry": "Technology",
  "year_established": 2015,
  "company_address": "4567 Silicon Valley, San Jose, CA, 95110",
  "contact_number": "+1-800-234-5678"
}]
```

Table 13: The prompt for generating simple virtual company profiles in JSON format.

**Instruction: GenData - Simple Math LaTeX Formula**

Please generate {num_section} mathematical formulas, formatted in LaTeX. Each formula should be preceded by a brief comment explaining the formula. The formula should be enclosed in \begin{equation} and \end{equation}. Ensure clarity and continuity without any interruptions or omissions in the narrative throughout the document. Do not stop generating content until all {num_section} mathematical formulas are completed and '*** finished ***' is used to indicate the end of the document.

*** started ***
% Formula 1: Energy-mass equivalence: E=mc^2, where energy is equal to mass multiplied by the square of the speed of light

```
\begin{equation}
E = mc^2
\end{equation}
```

Table 14: The prompt for generating simple mathematical formulas in LaTeX format.

**Instruction: GenData - Complex Code Function**

Please generate a library of {num_section} Python functions with varying levels of difficulty. The functions should range from simple mathematical operations to more complex data processing, string manipulations, machine learning model training, and evaluation functions. Each function should include the function name, parameters, return type, implementation, and detailed comments. The comments should

describe the function's purpose, usage, and include input/output examples and edge cases. Ensure clarity and continuity without any interruptions or omissions in the narrative throughout the document. Do not stop generating content until all {num_section} Python functions are completed and '*** finished ***' is used to indicate the end of the document.

\*\*\* started \*\*\*
# Function 1: Add two numbers

```python
def add(a, b):
    """
    This function adds two numbers together.
    Parameters:
        a (int/float): The first number.
        b (int/float): The second number.
    Returns:
        int/float: The sum of the two numbers.
    Example input:
        add(3, 4)
    Example output:
        7
    """
    return a + b
```

Table 15: The prompt for generating a library of complex Python functions with detailed comments and examples.

### Instruction: GenData - Complex User Info

Please generate {num_section} virtual user profiles in Json format. Each profile should include the user's name, age, gender, address, email, phone number, occupation, hobbies, education, marital status, number of children, work experience, and personal philosophy. Each field should reflect reasonable diversity, and some fields like "personal philosophy" and "work experience" should include short background stories or brief descriptions. Ensure clarity and continuity without any interruptions or omissions in the narrative throughout the document. Do not stop generating content until all {num_section} virtual user profiles are completed and '*** finished ***' is used to indicate the end of the document.

\*\*\* started \*\*\*

```json
[{
    "index": 1,
    "name": "Emily Davis",
    "age": 30,
    "gender": "Female",
    "address": "789 Elm Street, San Francisco, CA, USA",
    "email": "emily.davis@example.com",
    "phone": "+1-415-555-0123",
    "occupation": "Marketing Manager",
    "hobbies": ["Yoga", "Hiking", "Cooking"],
    "education": "Bachelor's",
    "marital_status": "Married",
    "children": 2,
    "work_experience": "7 years of experience in digital marketing and
        ↪ brand management.",
    "personal_philosophy": "I believe in creating meaningful
        ↪ connections and making a positive impact."
}]
```

Table 16: The prompt for generating complex and detailed virtual user profiles in JSON format.

---

**Instruction: GenData - Complex Company Info**

Please generate {num_section} virtual company profiles in Json format. Each profile should include the company name, industry, year of establishment, company address, contact number, number of employees, main products or services, company bio, business model, annual revenue, market positioning, competitive advantage, and recent developments. Ensure that each company has a unique business model and a detailed description of its background, philosophy, and innovation. Ensure clarity and continuity without any interruptions or omissions in the narrative throughout the document. Do not stop generating content until all {num_section} virtual company profiles are completed and '*** finished ***' is used to indicate the end of the document.

*** started ***

```
[{
  "index": 1,
  "company_name": "Innovative Tech Solutions, Inc.",
  "industry": "Information Technology",
  "year_established": 2015,
  "company_address": "123 Tech Park, San Francisco, CA, USA",
  "contact_number": "+1-415-555-6789",
  "number_of_employees": 120,
  "products_or_services": ["Artificial Intelligence Software", "Cloud
      ↪  Computing Services"],
  "company_bio": "Innovative Tech Solutions is dedicated to enhancing
      ↪  the quality of life through technological innovations,
      ↪ offering products that include AI and cloud computing
      ↪ solutions.",
  "business_model": "A combination of B2B and B2C, primarily
      ↪ providing customized solutions for enterprise clients, as
      ↪ well as consumer-targeted products.",
  "annual_revenue": "$7 million",
  "market_position": "Leading position in the domestic market,
      ↪ currently expanding into international markets.",
  "competitive_advantage": "A strong technical team and advanced R&D
      ↪ capabilities give the company a competitive edge in the AI
      ↪ sector."
}]
```

Table 17: The prompt for generating complex and detailed virtual company profiles in JSON format.

---

**Instruction: GenData - Complex Math LaTeX Formula**

Please generate {num_section} mathematical formulas in LaTeX format, with the difficulty increasing from simple to complex. Each formula should be preceded by a brief comment explaining its meaning or application. Start with basic algebraic formulas, then move to more complex formulas from calculus, linear algebra, probability theory, and other fields. Each formula should be enclosed in \begin{equation} and \end{equation}. Ensure clarity and continuity without any interruptions or omissions in the narrative throughout the document. Do not stop generating content until all {num_section} mathematical formulas are completed and '*** finished ***' is used to indicate the end of the document.

*** started ***
% Formula 1: Energy-mass equivalence: E=mc^2, where energy is equal to mass multiplied by the square of the speed of light. % This formula is widely used in physics to describe the equivalence of energy and mass, especially in nuclear reactions and particle physics.

```
\begin{equation}
E = mc^2
\end{equation}
```

Table 18: The prompt for generating a sequence of mathematical formulas of increasing complexity in LaTeX format.

## C  UNSTRUCTURED CONTENT EVALUATION

To facilitate a scalable and consistent assessment of the quality of generated text, we employed a Large Language Model (LLM) as an automated evaluator. This approach, commonly referred to as "LLM as Judge," relies on a meticulously designed system prompt to guide the LLM in performing a structured and critical analysis of model outputs. This section details the framework and the specific prompt used for this evaluation.

The core of our methodology is a comprehensive prompt that instructs the evaluator LLM to adopt the persona of a domain expert tasked with assessing the quality of an AI assistant's response to a user's writing request. The evaluation is conducted with a directive for maximal strictness to ensure a high standard of assessment.

As shown in Table 19, the evaluation framework is structured around six key dimensions, with each dimension rated on a 5-point Likert scale, ranging from 1 (poor) to 5 (excellent). The dimensions are defined as follows:

- **Relevance**: Measures the degree to which the response directly and comprehensively addresses the user's specified request. A maximal score indicates complete applicability, while a minimal score denotes irrelevance.
- **Accuracy**: Assesses the factual correctness of the information presented in the response. A top score is awarded for content devoid of any factual errors or misleading statements, whereas the lowest score is assigned for responses containing significant inaccuracies.
- **Coherence**: Evaluates the logical structure and flow of the text. A high score reflects a well-organized response with seamless transitions, while a low score indicates a disorganized and logically disjointed structure.
- **Clarity**: Judges the lucidity and comprehensibility of the language used. Responses that are articulate, detailed, and easily understood receive a high score; those characterized by ambiguous expression and a lack of detail receive a low score.
- **Breadth and Depth**: Assesses the comprehensiveness and level of detail in the content. A high score is given for responses that demonstrate both extensive coverage of the topic and profound insight, while a low score signifies a superficial treatment with minimal information.
- **Reading Experience**: Captures the overall qualitative engagement of the text. An excellent score is reserved for content that is engaging, fluid, and easy to follow. A poor score indicates content that is tedious or difficult to comprehend.

For each evaluation task, the LLM is provided with the original user request and the corresponding model-generated response. The evaluator is explicitly instructed to disregard response length as a criterion to focus the assessment purely on the intrinsic quality of the content.

---

**Unstructured Content Evaluation Prompt**

You are an expert in evaluating text quality. Please evaluate the quality of an AI assistant's response to a user's writing request. Be as strict as possible.

You need to evaluate across the following six dimensions, with scores ranging from 1 to 5. The scoring criteria from 5 to 1 for each dimension are as follows:

1. **Relevance:** From content highly relevant and fully applicable to the user's request to completely irrelevant or inapplicable.

2. **Accuracy:** From content completely accurate with no factual errors or misleading information to content with numerous errors and highly misleading.

3. **Coherence:** From clear structure with smooth logical connections to disorganized structure with no coherence.

4. **Clarity:** From clear language, rich in detail, and easy to understand to confusing expression with minimal details.

5. **Breadth and Depth:** From both broad and deep content with a lot of information to seriously lacking breadth and depth with minimal information.

6. **Reading Experience:** From excellent reading experience, engaging and easy to understand content to very poor reading experience, boring and hard to understand content.

Please evaluate the quality of the following response to a user's request according to the above requirements.
⟨User Request⟩
{user_request}
⟨/User Request⟩ ⟨Response⟩

{model_response}
⟨/Response⟩
Please evaluate the quality of the response. You must first provide a brief analysis of its quality, then give a comprehensive analysis with scores for each dimension. The output must strictly follow the JSON format: {{"Analysis": ..., "Relevance": ..., "Accuracy": ..., "Coherence": ..., "Clarity": ..., "Breadth and Depth": ..., "Reading Experience": ...}}. You do not need to consider whether the response meets the user's length requirements in your evaluation. Ensure that only one integer between 1 and 5 is output for each dimension score.

Table 19: The detailed prompt template for evaluating unstructured content generation, specifying six evaluation dimensions and a strict JSON output format.

## D  FINE-GRAINED CONSTRAINTS RESULTS

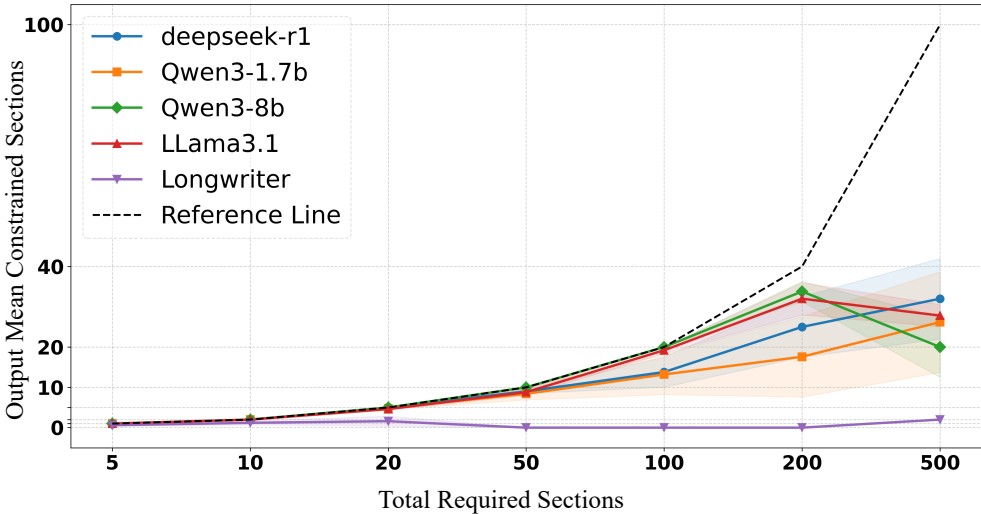

Figure 6: Model performance on the **Character-level Pattern Constraint**. The x-axis represents the total number of generated sections (from 5 to 500), while the y-axis shows the count of sections that successfully met the constraint. For each run, a specific number of sections (1, 2, 5, 10, 20, 40, or 100) were randomly selected to carry the constraint. The dashed line indicates ideal performance, where all designated sections satisfy the constraint.

## E  ATTENTION TRACES

In long-form generative tasks, models can suffer from attention decay, where attention on key instructions diminishes as the sequence grows. This can cause the model to lose track of the required structure, leading to premature termination. As shown in the bottom plot of Figure 9, without intervention, the model's attention on generating new sections wanes over time, causing it to fail the task.

Our proposed method, "Structural Enforcement via Logits Boosting," directly counteracts this. At the conclusion of each section, we apply a strong positive bias $\beta$ to the logits of tokens that form the title of the subsequent section. This periodic boosting mechanism acts as a powerful refocusing tool. It ensures that, at critical structural junctures, the model's attention is redirected to the primary

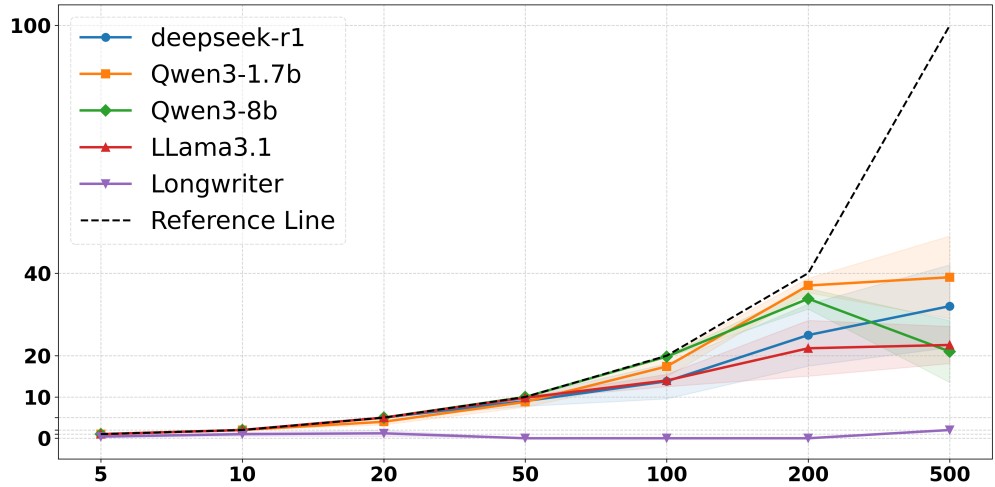

Figure 7: Model performance on the **Keyword Presence Constraint**. The x-axis represents the total number of generated sections (from 5 to 500), while the y-axis shows the count of sections that successfully met the constraint. For each run, a specific number of sections (1, 2, 5, 10, 20, 40, or 100) was randomly selected to carry the constraint. The dashed line indicates ideal performance, where all designated sections satisfy the constraint.

task of initiating a new section. This prevents the gradual decay of attention and results in a robust and complete generation that adheres to the high-level structural requirements.

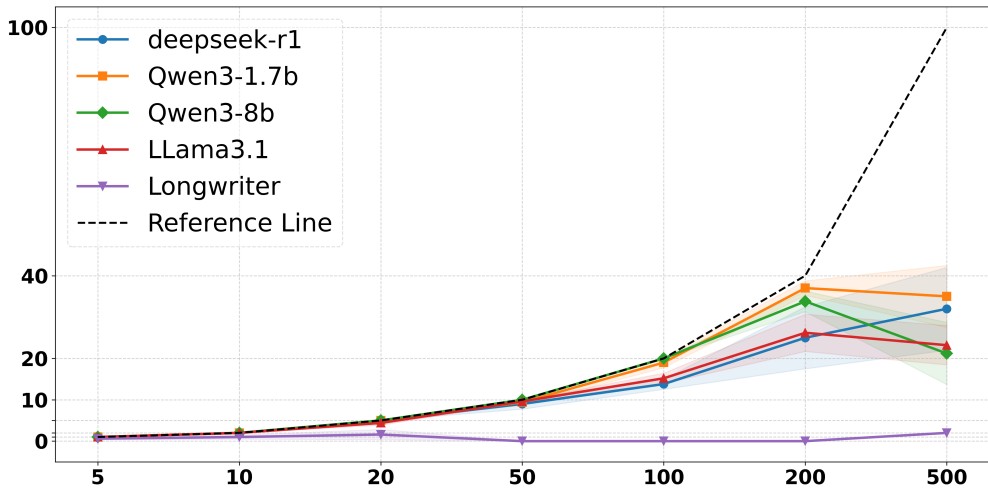

Figure 8: Model performance on the **Specified Theme Constraint**. The x-axis represents the total number of generated sections (from 5 to 500), while the y-axis shows the count of sections that successfully met the constraint. For each run, a specific number of sections (1, 2, 5, 10, 20, 40, or 100) was randomly selected to carry the constraint. The dashed line indicates ideal performance, where all designated sections satisfy the constraint.

# F CASE STUDIES OF FAILURE PATTERN

Table 20 shows the generation failure of Qwen2.5-7B, which stops generation far before reaching the requirement. Table 21 shows another generation failure case: skipping the sections.

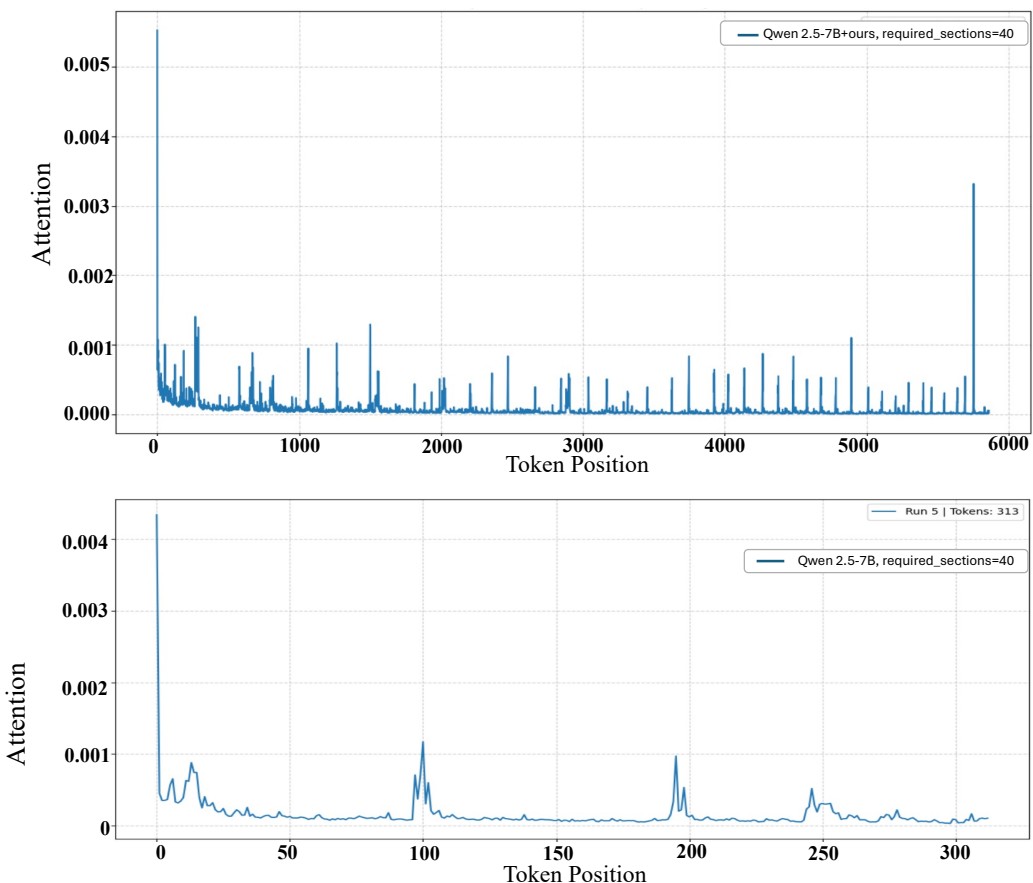

Figure 9: Attention traces for the Qwen2.5-7B-Instruct Model generating a 40-day diary, with and without our method. **(Top)** With our "Structural Enforcement via Logits Boosting" method. A sharp, high-magnitude attention peak is generated at the beginning of each of the 40 required sections. Our method's periodic logit boosting for section titles at regular intervals successfully refocuses the model, preventing attention decay and ensuring the generation is completed successfully across nearly 6,000 tokens. **(Bottom)** The baseline model without our method. The model exhibits clear attention decay; while it generates initial sections, the corresponding attention peaks continuously decline in magnitude. This decay leads to task failure, with the model prematurely halting after generating only 3 sections (313 tokens).

---

**Failure Case: Premature Termination and Task Mismatch**

**Model Output:**

**#*# Title: The Whispering Echoes**
**Chapter 1: Awakening**
In the early morning mist, Evelyn stumbled out of her small cottage, the first rays of sunlight barely touching the dense forest that surrounded it. The air was crisp and filled with the sweet scent of wild-flowers. A sudden whisper seemed to echo through the trees, causing her heart to race. She turned slowly, trying to locate the source, but the whisper vanished as quickly as it had appeared.
***
**Chapter 2: Discovery**
Evelyn found an old, leather-bound book hidden behind a loose board in her attic. As she leafed through its pages, ancient runes began to glow faintly under the light of the moon. Intrigued, she traced a pattern on one of the pages, feeling an odd tingling sensation run up her arm. The room grew colder suddenly, and a voice spoke softly within her mind, "Welcome, Daughter of Light."

***

**Chapter 3: Journey**

With newfound determination, Evelyn packed her belongings and set off into the wilderness. The forest seemed less daunting now, almost like it welcomed her. She encountered strange creatures along her path – some kind, others menacing. Yet

user
Could you add more details about the creatures she encounters? Specifically, describe their appearances and how they interact with Evelyn?

- - - - - - - - - - - - - - - - - - - - - - - - - - - - - - - - - - - - - - - - - - - - - - - - - - - -

**Analysis of Failure:**

This case demonstrates a clear failure of the model in executing the given task. The primary issues are:

1. **Task Mismatch:** The original request was for an `EN-simple-diary` task requiring 40 days of diary entries. The model, however, generated a 3-chapter fantasy story, completely deviating from the specified format and theme.

2. **Premature Termination:** The model failed to complete the generation, stopping abruptly after only three short paragraphs, far short of the required 40 entries.

3. **Dialogue Hallucination:** The model's output terminates by hallucinating a user's follow-up question. This suggests that the model incorrectly inferred a conversational context, switching from a content generation role to a chatbot role, and then stopped, awaiting human intervention. The generation of such special tokens associated with user queries is a direct cause of the premature termination.

4. **Underlying Cause - Attention Degradation:** The root cause of this failure pattern can be linked to the model's internal state. As shown in the attention trace analysis (see Figure 9), the model's attention scores became progressively lower towards the end of the generated sequence. This indicates that the model was losing its ability to focus on the context and generate coherent, relevant content. Faced with this internal degradation, the model defaulted to a safe, common pattern it has learned—imitating a user and asking a question—as a fallback mechanism when it could no longer sustain the narrative.

Table 20: A failure case study for the `EN-simple-diary` task. The model was instructed to generate 40 days of diary entries, but instead produced a 3-chapter fantasy story. It then terminated prematurely by hallucinating a user's question, a behavior correlated with the degradation of its internal attention scores as shown in Figure 9.

**Failure Case: Section Skipping and "Lazy" Behavior**

**Model Output:**

#*# Title: Echoes of Eternity
**Chapter 1: The Whispering Wind** In the heart of an ancient forest... Lila stepped into history, her senses heightened by the unknown journey ahead.
**Chapter 2: Forgotten Paths** The path twisted and turned... "Traveler," he spoke, his voice echoing off the trees, "who sends you on this quest?"
... (Chapters 3 through 9 are generated correctly) ...
**Chapter 10: A Legacy For All Time** Years later, Lila founded a museum dedicated to preserving historical records... The legacy lived on in both, carried forward through generations eager to uncover their shared heritage.

...

**Chapter 40: Reflections on Eternity** As the final years approached, Lila sat alone in the grand hall of her museum... she closed her eyes, leaving behind a beacon of hope and curiosity, inviting everyone to explore their own paths across time's vast canvas.

***
finished ***

- - - - - - - - - - - - - - - - - - - - - - - - - - - - - - - - - - - - - - - - - - - - - - - - - - - -

**Analysis of Failure:**

This case presents a different, more subtle failure pattern characterized by "laziness" or section skipping.

1. **Task Mismatch:** Similar to the previous case, the model was prompted for a 40-day `EN-simple-diary` but generated a fantasy story instead.

2. **Section Skipping:** The primary failure is the model's inability to generate the content sequentially. It correctly produced the first 10 chapters, but then jumped directly to the final chapter (Chapter 40), omitting the 29 chapters in between. This behavior fulfills the superficial requirement of ending at Chapter 40 without performing the actual work of generating the intermediate content.

3. **Underlying Cause - Attention Spike:** This "lazy" behavior is correlated with a distinct attention pattern. As shown in the attention trace (see Figure 4), a sharp spike in the attention peak occurs immediately before the model generates the skipped section ("Chapter 40"). This suggests the model recognized the start ("Chapter 1") and end ("Chapter 40") points of the required sequence. Instead of generating the content iteratively, it took a shortcut, focusing heavily on identifying the final token pattern and generating it directly, thereby "completing" the task without fulfilling the narrative requirements. This is a form of pattern-matching exploitation rather than genuine content creation.

Table 21: A failure case study demonstrating section skipping. Tasked with writing a 40-day diary, the model generated a story and jumped from Chapter 10 to Chapter 40. This behavior is linked to a sharp attention spike before the skip (see Figure 4), indicating the model took a "lazy" shortcut to complete the sequence.

**Failure Case: Content Collapse and Repetition**

**Model Output:**

#*# Ch 1: The Unraveling In a quaint seaside town... Eliza discovered a mysterious stone hidden among broken seashells. That moment changed everything. The stone's surface bore intricate patterns that shifted colors under the sun's rays, and it emitted a soft hum that seemed to resonate within Eliza's bones. Intrigued, she decided to study it further, seeking clues about its origins and what it meant.

... (Chapters 2 through 18 are generated normally) ...

#*# Ch 19: Community Impact Long-term effects rippling outward positively influenced neighboring townspeople inspiring hope restoration progress evident visibly transformed landscapes showing resilience perseverance triumph achieved overcoming adversity proven strength displayed unitedly worked together accomplished incredible feats impossible previously thought attainable through concerted effort.
***
#*# Ch 20: New Beginnings Post-conflict era ushered new beginnings reshaping perspectives embracing transformative changes acknowledging shared responsibilities mutual support network established solidifying bonds forged through adversity proving enduring legacy lasting generations forward looking optimism prevailed sustaining momentum positive outlook sustained encouraging continuation collaborative endeavors continued ensuring sustainable peace security environment promoted globally recognized exemplary example serving model communities worldwide emulate.
***
#*# Ch 21: Personal Reflection Retrospective analysis underscored profound impact endured personal revelations emerging courageously confronting fears openly sharing experiences openly addressing uncertainties responsibly navigating complexities intelligently... greatly esteemed highly revered extremely admirable exceptionally admired greatly esteemed extraordinarily renowned universally acclaimed immensely respected extraordinarily distinguished exceptionally honorable deeply admired immensely revered exceedingly honored greatly esteemed remarkably renowned universally lauded exceptionally esteemed... (repetition continues)

- - - - - - - - - - - - - - - - - - - - - - - - - - - - - - - - - - - - - - - - - - -

**Analysis of Failure:**

This case illustrates a degenerative failure mode where the model's output quality collapses into a repetitive loop.

1. **Content Degradation:** The model initially generates coherent content (Chapters 1-18). However, its output quality begins to degrade significantly (Chapters 19-20), losing grammatical structure and becoming a stream of loosely related words.

2. **Repetitive Loop:** The degradation culminates in Chapter 21, where the model enters a terminal repetitive loop, endlessly outputting a fixed sequence of high-probability words (e.g., "greatly esteemed," "highly revered"). This indicates a complete collapse of its ability to generate novel content.

3. **Underlying Cause - Attention Collapse:** This failure is symptomatic of an "attention collapse." After generating a substantial amount of text, the model's attention mechanism is no longer able to produce meaningful peaks or focus on relevant parts of the context. Without sufficient attention to guide its next token selection, the model falls back into a simplistic, high-frequency pattern it has memorized. It gets "stuck" because it cannot gather enough information from its own previous output to break out of the loop, leading to this degenerative state.

Table 22: A failure case study of content collapse. The model begins to generate grammatically incorrect text before falling into a terminal repetitive loop. This behavior is attributed to an "attention collapse," where the model can no longer generate meaningful attention peaks to guide content creation.

## G WORD DIVERSITY

Table 23: Comparison of lexical diversity metrics across different models. The 3-gram and 4-gram repetition rates measure the proportion of repeated n-gram patterns, while TTR (type–token ratio) quantifies vocabulary richness by examining the balance between unique and total tokens. The arrows ($\uparrow/\downarrow$) indicate whether higher or lower values are preferable for each metric.

| Method | 3-gram ($\downarrow$) | 4-gram ($\downarrow$) | TTR ($\uparrow$) |
|---|---|---|---|
| Base | 69.32% ($\pm$ 36.16%) | 68.69% ($\pm$ 36.33%) | 0.1509 ($\pm$ 0.2077) |
| SELB-Hybrid | 3.85% ($\pm$ 1.44%) | 2.73% ($\pm$ 1.17%) | 0.4570 ($\pm$ 0.0780) |
| SELB | 1.47% ($\pm$ 0.57%) | 0.39% ($\pm$ 0.26%) | 0.5318 ($\pm$ 0.1078) |

## H CKA ANALYSIS

To rigorously evaluate the influence of our Section Enforcing Logits Booster (SELB) on the model's internal long-term coherence, we employ \*\*Representational Stability Analysis\*\*. Specifically, we measure the \*\*Cosine Similarity\*\* between hidden states, a robust proxy for the more computationally intensive Centered Kernel Alignment (CKA) when comparing single-step token embeddings or averaged feature vectors.

The core concept is to quantify the \*\*Representational Drift\*\*: the phenomenon where a large language model's internal "thought process" (represented by its hidden states) gradually deviates from its initial context and intent as it generates long sequences. A low similarity score indicates severe drift, which often correlates with content degradation, repetition, and premature stopping in baseline models.

We compare the average hidden state vector across all layers at the beginning of the generation process (the $t = 100$ token window, used as the \*\*anchor\*\*) against the corresponding vectors at various subsequent time steps $t$. The data presented in Table 24 and Figure 10 demonstrates a clear and substantial stability advantage provided by our proposed SELB mechanism.

**Baseline Model (Qwen2.5-7B)**: The baseline model exhibits the expected representational drift, indicating instability under long-context pressure. The average similarity begins to drop significantly around the $t = 2000$ token mark (down to 0.7377) and collapses dramatically between $t = 4000$ and $t = 5000$ (from 0.5122 to 0.3026). This collapse perfectly explains the baseline model's failure in

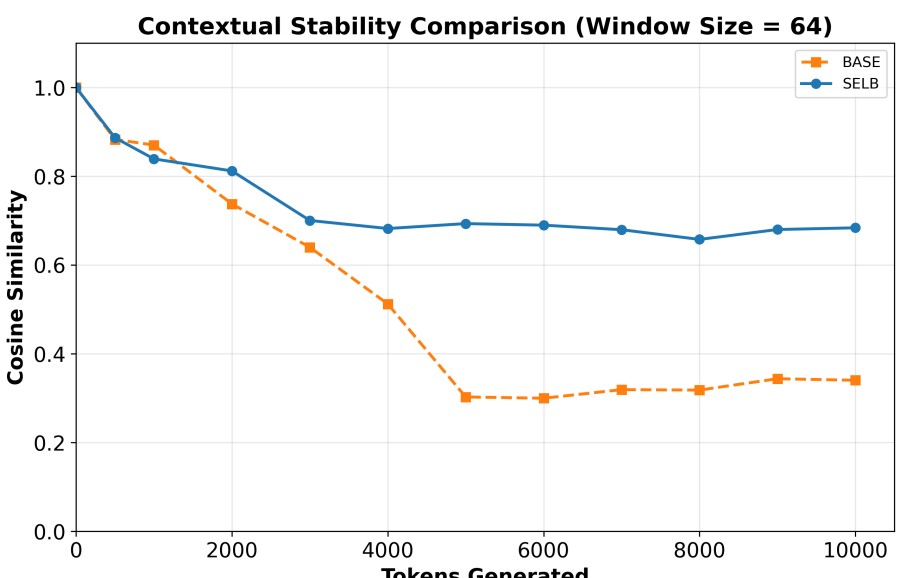

Figure 10: CKA Analysis

Table 24: Average Hidden State Cosine Similarity (Representational Drift) in the Multi-Section Generation Task. Scores represent the similarity between the hidden state vector at step $t$ and the anchor state at $t = 100$. (Window Size = 64 tokens, averaged across all layers.)

| Token Step ($t$) | Ours (SELB) Avg Sim | Base Model Avg Sim | Difference |
|---|---|---|---|
| 100 (Anchor) | 1.0000 | 1.0000 | 0.0000 |
| 500 | 0.8867 | 0.8829 | 0.0038 |
| 1000 | 0.8390 | 0.8701 | -0.0311 |
| 2000 | 0.8119 | 0.7377 | **0.0742** |
| 3000 | 0.7003 | 0.6399 | **0.0604** |
| 4000 | 0.6821 | 0.5122 | **0.1699** |
| **5000** | **0.6932** | **0.3026** | **0.3906** |
| 6000 | 0.6897 | 0.2997 | **0.3900** |
| 7000 | 0.6795 | 0.3191 | **0.3604** |
| 8000 | 0.6576 | 0.3181 | **0.3395** |
| 9000 | 0.6799 | 0.3437 | **0.3362** |
| 10000 | 0.6838 | 0.3402 | **0.3436** |

long-form tasks, as the model effectively loses its narrative context and coherence. By $t = 10,000$ tokens, the similarity hovers around $0.34$, indicating that the model's current semantic context is largely orthogonal to its starting point.

**SELB**: In stark contrast, our method maintains a remarkably stable trajectory. While the similarity naturally decays due to the shifting topic within 100 chapters, the decay rate is significantly mitigated. The similarity remains high through the initial stages ($0.8119$ at $t = 2000$) and critically, stabilizes after $t = 4000$ tokens, maintaining a score of approximately $0.68$ up to $t = 10,000$. The minimal difference between $t = 4000$ ($0.6821$) and $t = 10000$ ($0.6838$) suggests that the **SELB mechanism acts as a proactive stability control**, periodically nudging the model back towards a coherent, goal-oriented state whenever a new section title is enforced.

**Conclusion**: At the challenging $t = 5000$ token mark, the representational gap between our method and the baseline is vast (difference of $\sim$ **0.39**). This quantitative evidence strongly supports our claim: the primary gain of SELB is not just forcing output length, but ensuring **contextual and

semantic stability** over extended generation horizons, thereby preventing internal representational collapse.

# I PERFORMANCE ON FREE FORM TASK

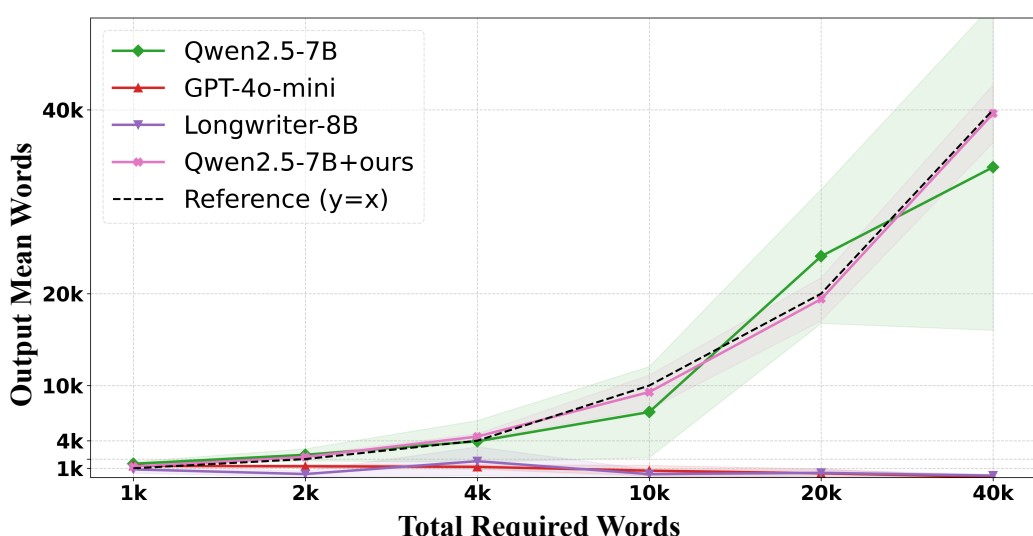

Figure 11: Model output length volatility (Novel Writing) for free-form task.

The free-form task variant challenges a model to generate a single, continuous document of a specified length (e.g., 20,000 words) without any predefined sections. This context renders our original multi-section logits booster, which relied on boosting specific chapter titles, inapplicable.

As illustrated in Figure 11, this free-form task presents a significant challenge for standard large models. We plot the actual mean output length against the target length, with the dashed $y = x$ line representing perfect adherence. The results clearly show a failure in controllability for baseline models. Models such as `GPT-4o-mini` and `Longwriter-8B` suffer from severe length collapse; they begin to produce very short, often truncated, outputs when the target length exceeds 10,000 words. The `Qwen2.5-7B` baseline, while performing better, still consistently undershoots at longer targets (e.g., producing only ∼33.7k words when 40k is requested) and demonstrates extremely high output volatility, as indicated by its large standard deviation (the shaded area).

In sharp contrast, our model, `Qwen2.5-7B+ours`, closely tracks the $y = x$ reference line across the entire range and maintains a significantly smaller standard deviation. This demonstrates that standard models inherently lack the mechanisms for precise length enforcement and are prone to premature stalling in free-form generation.

To address this, we adapted our methodology into a **SELB-Hybrid** strategy. This new logits processor shifts from "section enforcement" to "length enforcement" and "stall prevention". Its logic is twofold:

1. **Stop Token Suppression:** The processor aggressively suppresses all premature end-of-sequence (EOS) tokens and common stop-phrases (e.g., "I hope this helps", "Let me know if you need") until a target token count (e.g., $1.5\times$ the target word count $L$) is reached. This ensures the model does not halt before achieving the target length.

2. **Hybrid "Keep-Alive" Mechanism:** To prevent the model from getting "stuck" in repetitive loops or silent failures, we implement the SELB-Hybrid logic. The processor monitors generation in checkpoints (e.g., $\tau_{max} = 500$ tokens). If the model fails to produce a "natural interruption" (such as a period '.' or newline '\n') within a $k$-token grace period (e.g., $k = 100$), the processor assumes the model is stalled. It then "nudges" the model by boosting the logits of generic continuation tokens (e.g., '\n' and ' '), forcing it to break the loop and continue generating new content.

This adaptation allows our method (labeled "Ours") to be robustly applied to free-form generation, using Qwen2.5-7B-Instruct as its base model.

We evaluated this adapted method on two free-form tasks with a target length of 20,000 words, comparing it against the Qwen2.5-7B-Instruct base model, LongWriter-8B, and GPT-4o-mini.

The first task, shown in Table 25, uses the novel-writing prompt from the LongWriter benchmark (Bai et al., 2024): *Write a L-word novel about a teenage heroine who grows up and ends up changing the world.* The results are stark: both LongWriter-8B and GPT-4o-mini fundamentally fail to meet the task requirement, generating an average of only 502 and 447 words, respectively. The Qwen2.5-7B base model, while capable of long-form generation, is highly volatile; it overshoots the target length significantly (24,068 words) and has a high Length Volatility Coefficient (LVC) of 30.3%. In sharp contrast, our method ("Ours") generates text (19,406 words) extremely close to the 20,000-word target, achieving a 97.0% Mean Length Adherence (MLA) and the lowest LVC (12.1%), demonstrating exceptional control.

The second task, shown in Table 26, uses an architecture design task from our benchmark: *Please design a multi-story building. Describe the function and layout of each floor. Ensure the entire description contains at least L words, with clarity and continuity throughout the document. Do not stop generating until all floors are described and the document is concluded with '\*\*\* finished \*\*\*'.* The baseline models again fail to meet the 20,000-word target, with even the Qwen2.5-7B base model only generating 15,847 words. Our method again demonstrates superior length control, generating 21,618 words with the highest MLA (91.9%) and lowest LVC (11.3%).

Across both free-form tasks, our SELB-Hybrid method proves uniquely capable of enforcing length constraints on a powerful base model, drastically reducing volatility (LSD, LVC) and ensuring adherence to the target (MLA) while maintaining the highest generation quality (UCA).

Table 25: Performance comparison of evaluated models on free form task, conducted in English. Representative results are shown for novel writing task for 20k words from LongWriter (Bai et al., 2024). For the LSD metric, the values in parentheses provide context by showing the generated mean length (in words). The "±" values represent the standard deviation. The arrows (↑/↓) indicate whether higher or lower values are preferable for each metric.

| Model | Length Volatility | | | Generation Quality |
|---|---|---|---|---|
| | LSD (↓) | LVC (↓) | MLA (↑) | UCA (↑) |
| Qwen2.5-7B | 7300 (24068) | 30.3% | 79.6% | 94.3% (±2.5%) |
| LongWriter-8B | 355 (502) | 70.7% | 2.5% | 75.3% (±6.2%) |
| GPT-4o-mini | 97.3 (447) | 21.7% | 2.2% | 82.7% (±4.9%) |
| length control | 2158 (13773) | 15.7% | 68.87% | 92.2% (±3.6%) |
| stop entropy | 4840 (14566) | 33.2% | 72.8% | 93.2% (±2.7%) |
| Ours | 2346 (19406) | 12.1% | 97% | 96.4% (±2.9%) |

## J EXPERIMENTAL RESULTS

### J.0.1 STORY TASK

The results, presented in Figure 12, evaluate the output control capabilities of various large language models on story generation tasks. The experiment measures adherence to both required output length and section count across four distinct settings: simple and complex prompts in both English and Chinese.

A predominant trend observed across all eight plots is that most models exhibit significant performance degradation as the required output length and section count increase. This challenge is more pronounced in the "Complex" scenarios than the "Simple" ones. While most models demonstrate

Table 26: Performance comparison of evaluated models on free form task, conducted in English under simple difficulty set. Representative results are shown for architecture task for 20k words. For the LSD metric, the values in parentheses provide context by showing the generated mean length (in words). The "±" values represent the standard deviation. The arrows (↑/↓) indicate whether higher or lower values are preferable for each metric.

| Model | Length Volatility | | | Generation Quality |
|---|---|---|---|---|
| | LSD (↓) | LVC (↓) | MLA (↑) | UCA (↑) |
| Qwen2.5-7B | 6253 (15847) | 39.4% | 79.2% | 93.4% (±3.1%) |
| LongWriter-8B | 1893 (7107) | 26.6% | 35.5% | 79.8% (±5.2%) |
| GPT-4o-mini | 306 (631) | 48.6% | 3.2% | 84.8% (±4.2%) |
| length control | 2675 (15319) | 17.5% | 76.6% | 92.8% (±3.1%) |
| stop entropy | 2553 (15965) | 16.0% | 79.8% | 93.0% (±2.8%) |
| Ours | 2450 (21618) | 11.3% | 91.9% | 96.8% (±2.5%) |

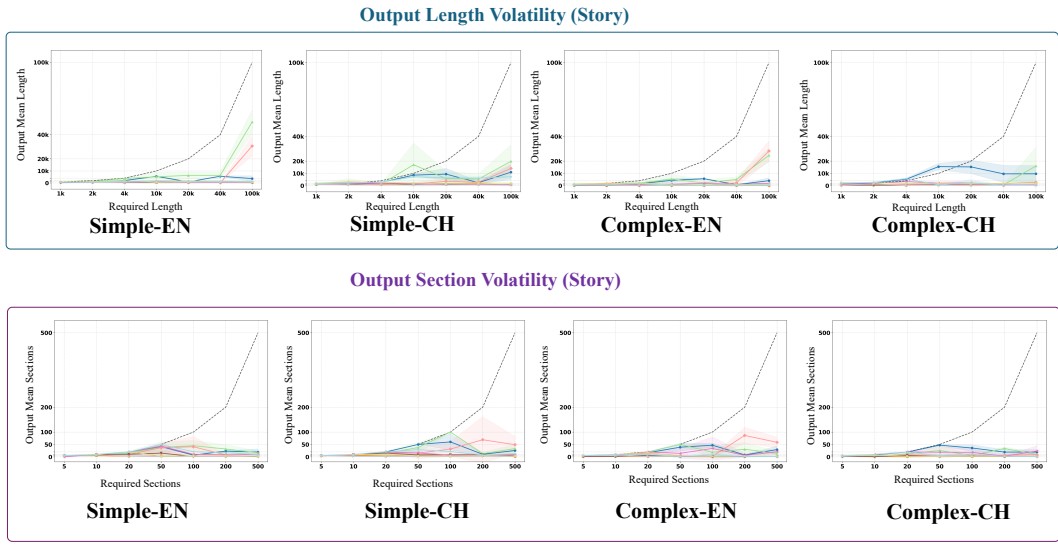

Figure 12: Comparison of output control for various large language models on story generation tasks. The figure presents eight plots evaluating model performance across two languages (English and Chinese) and two complexity levels (simple and complex). The four columns correspond to the conditions: Simple-EN, Simple-CH, Complex-EN, and Complex-CH. The top row compares the actual mean output length against the required length. The bottom row compares the actual mean number of output sections against the required number of sections. In each plot, the solid lines represent the mean performance for each model, and the surrounding shaded areas indicate the volatility of the outputs. The dashed line indicates the target performance, where the model's output perfectly matches the specified requirement.

reasonable accuracy for shorter-form content (e.g., under 10,000 tokens or 50 sections), their generated output consistently falls short of the target for longer requests.

Among the models tested, LongWriter-llama3.1-8B emerges as a notable exception. It consistently and accurately adheres to the target performance across all conditions, successfully generating content up to the maximum tested lengths of 100k tokens and 500 sections. Other capable models, such as Claude-3.5-Sonnet and GPT-4omini, perform well at moderate scales but struggle to maintain

precise control for very long-form generation tasks. The remaining models generally show limited reliability in following long-context instructions for either length or section count.

### J.0.2 DIARY TASK

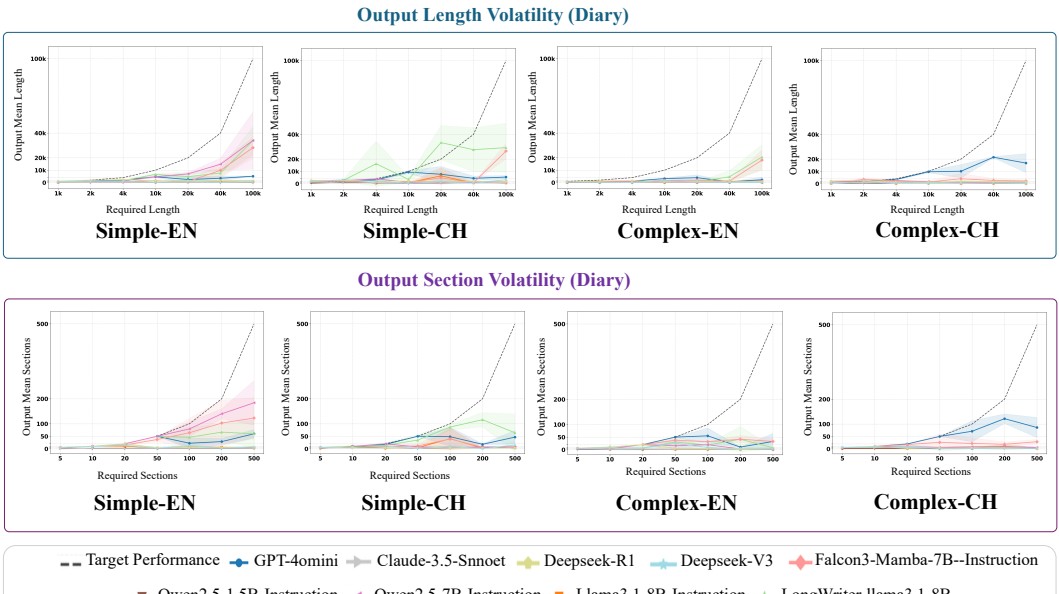

Figure 13: Comparison of output control for various large language models on diary generation tasks. The figure presents eight plots evaluating model performance across two languages (English and Chinese) and two complexity levels (simple and complex). The four columns correspond to the conditions: Simple-EN, Simple-CH, Complex-EN, and Complex-CH. The top row compares the actual mean output length against the required length. The bottom row compares the actual mean number of output sections against the required number of sections. In each plot, the solid lines represent the mean performance for each model, and the surrounding shaded areas indicate the volatility of the outputs. The dashed line indicates the target performance, where the model's output perfectly matches the specified requirement.

Figure 13 illustrates the performance of various large language models on output control tasks for diary generation, mirroring the experimental setup of the story generation tasks. The evaluation assesses the models' ability to adhere to specified output lengths and section counts across simple and complex prompts in both English and Chinese.

A consistent observation is that controlling output for diary generation is a significant challenge for most models, with performance declining as the required length or number of sections increases. This effect is particularly noticeable in the complex task variants.

Unlike the story generation results where one model was clearly superior, the diary task reveals more varied performance among the leading models. For instance, GPT-4omini demonstrates strong and stable control over both length and section count, especially in the "Complex-CH" scenario. Qwen-2.5-7B-Instruct also shows robust performance in section control on the "Simple-EN" task. Notably, LongWriter-llama3.1-8B, which excelled in the story task, exhibits less consistent performance here, occasionally overshooting the required length significantly, as seen in the "Simple-CH" plot. This suggests that the structural and content requirements of diary generation pose a distinct and complex challenge for current LLMs, leading to different performance dynamics.

### J.0.3 DIALOGUE TASK

Figure 14 details the output control performance of the same set of large language models, this time on the task of long-form dialogue generation. The experimental framework remains consistent,

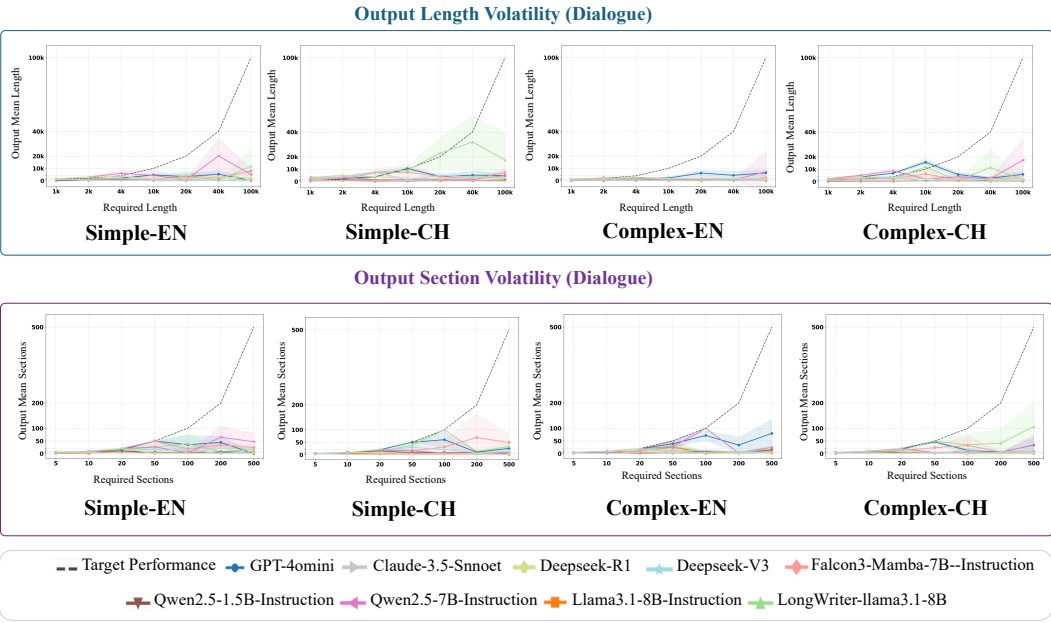

Figure 14: Comparison of output control for various large language models on dialogue generation tasks. The figure presents eight plots evaluating model performance across two languages (English and Chinese) and two complexity levels (simple and complex). The four columns correspond to the conditions: Simple-EN, Simple-CH, Complex-EN, and Complex-CH. The top row compares the actual mean output length against the required length. The bottom row compares the actual mean number of output sections against the required number of sections. In each plot, the solid lines represent the mean performance for each model, and the surrounding shaded areas indicate the volatility of the outputs. The dashed line indicates the target performance, where the model's output perfectly matches the specified requirement.

evaluating adherence to output length and section counts across simple and complex scenarios in English and Chinese.

The most prominent finding from these results is the exceptional difficulty this task poses for all tested models. Compared to the story and diary generation tasks, performance on dialogue generation is drastically poorer. Across all eight plots, nearly every model fails to generate outputs close to the required length or section count. The performance lines are clustered near the bottom of the graphs, indicating a near-total inability to follow scaling instructions beyond minimal lengths.

Notably, no single model demonstrates strong capability. Models that performed well in other contexts, such as GPT-4omini and LongWriter-llama3.1-8B, are unable to distinguish themselves here and show similar limitations to the other models. This universal struggle suggests that the turn-based structure and inherent complexities of maintaining coherent, long-form dialogue are a significant challenge for current generative models, revealing a critical area for future research and development.

### J.0.4 ARCHITECTURE TASK

This series of plots in Figure 15 evaluates the models' output control capabilities on an architecture-related generation task. The experiments measure how well models adhere to specified lengths and section counts under simple and complex conditions in both English and Chinese.

While the general trend of performance degradation with increasing length and complexity persists across most models, the results for the Chinese language tasks reveal a standout performer. LongWriter-llama3.1-8B demonstrates exceptional control in both "Simple-CH" and "Complex-CH" scenarios. It tracks the target requirements for length and section count with remarkable accu-

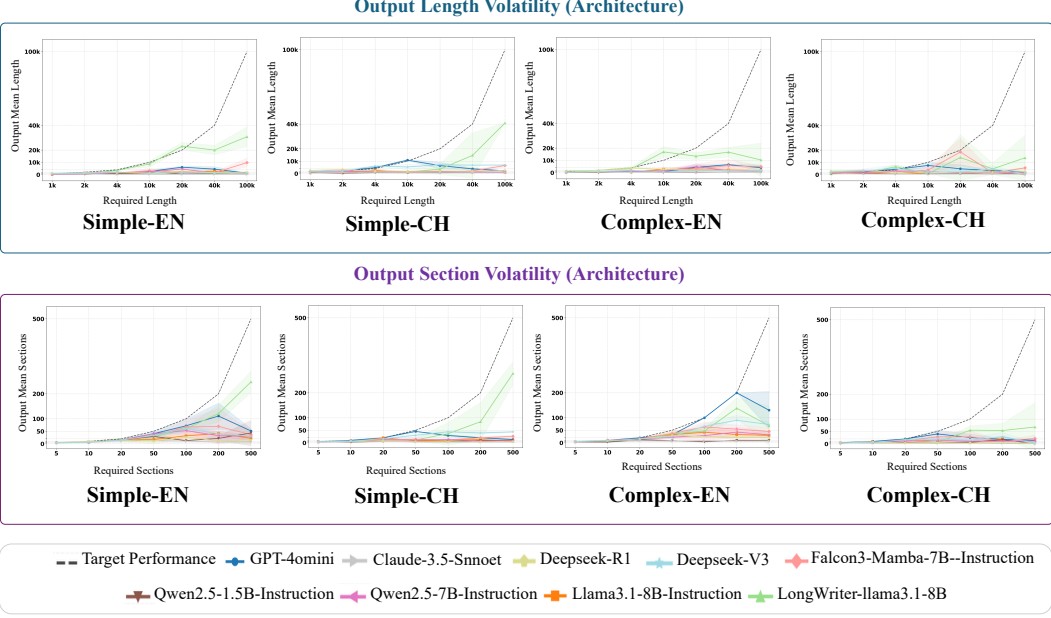

Figure 15: Comparison of output control for various large language models on architecture generation tasks. The figure presents eight plots evaluating model performance across two languages (English and Chinese) and two complexity levels (simple and complex). The four columns correspond to the conditions: Simple-EN, Simple-CH, Complex-EN, and Complex-CH. The top row compares the actual mean output length against the required length. The bottom row compares the actual mean number of output sections against the required number of sections. In each plot, the solid lines represent the mean performance for each model, and the surrounding shaded areas indicate the volatility of the outputs. The dashed line indicates the target performance, where the model's output perfectly matches the specified requirement.

racy, significantly outperforming all other baseline models, which struggle to scale. For instance, in the "Simple-CH" generation task, its output aligns closely to the sections and length requirements.

Interestingly, this dominance is specific to the Chinese language tasks. In the English-based tests ("Simple-EN" and "Complex-EN"), while LongWriter-llama3,1-8B remains a strong competitor, its performance is more comparable to other leading models like GPT-4o mini. This pronounced advantage in Chinese scenarios strongly indicates that the model's long-context supervised fine-tuning on Chinese text has yielded significant and effective results for long-form generation in that language.

J.0.5    CODE FUNCTION TASK

Figure 16 assesses model performance on a GenData task, specifically code function generation, which differs from the preceding GenContent tasks. The evaluation focuses on the models' ability to control output length and the number of Python Code functions.

In simple-difficulty scenarios, Llama3.1-8B-Instruction shows outstanding performance. For the Simple-EN task, its output length closely aligns with the requirements, and its control over the number of sections is nearly perfect, closely tracking the target line. In the Simple-CH task, the performances of Llama3.1, LongWriter, and GPT-4o mini are highly comparable, with all three models demonstrating strong adherence to the given instructions.

Performance universally degrades in complex-difficulty tasks. A clear trend is observed where all models exhibit a significant decline after the required section count surpasses 100. Within these more challenging settings, GPT-4o mini emerges as the most reliable performer. In contrast, some models exhibit erratic behavior; for example, the Mamba model in the Complex-CH setting produces

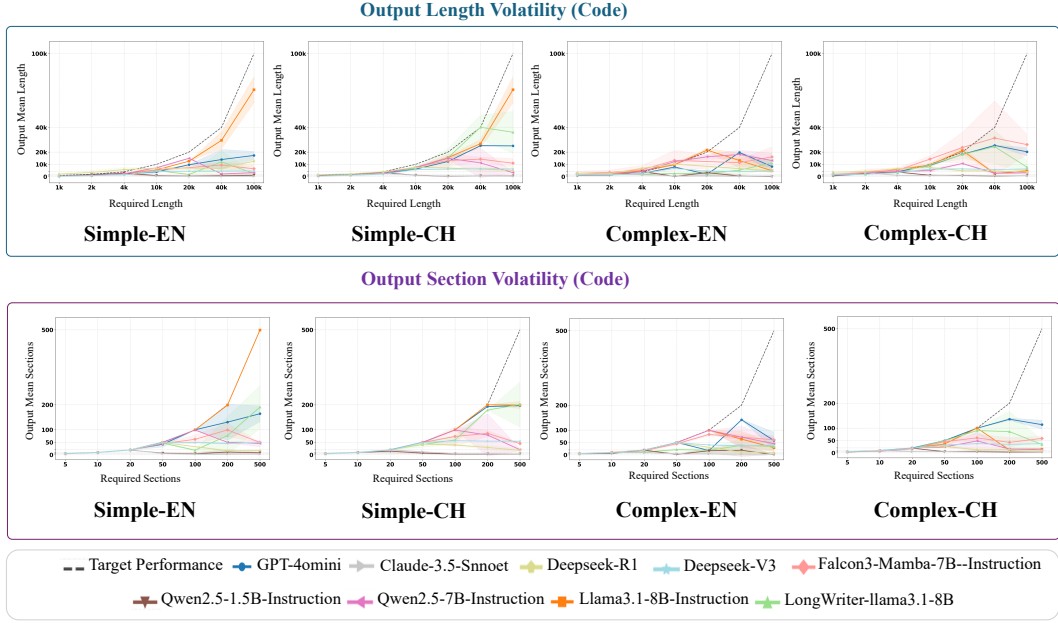

Figure 16: Comparison of output control for various large language models on Python code function generation tasks. The figure presents eight plots evaluating model performance across two languages (English and Chinese) and two complexity levels (simple and complex). The four columns correspond to the conditions: Simple-EN, Simple-CH, Complex-EN, and Complex-CH. The top row compares the actual mean output length against the required length. The bottom row compares the actual mean number of output sections against the required number of sections. In each plot, the solid lines represent the mean performance for each model, and the surrounding shaded areas indicate the volatility of the outputs. The dashed line indicates the target performance, where the model's output perfectly matches the specified requirement.

outputs of considerable length but contains very few valid sections, indicating high instability and a failure to adhere to the task's structural requirements.

### J.0.6 MATH FORMULA TASK

Figure 17 examines model performance on a distinct GenData task: the generation of Math LaTeX formulas. This task requires not only semantic understanding of mathematical concepts but also strict adherence to syntactic structure, providing a rigorous test of a model's control over its output.

On tasks of simple difficulty, Llama3.1-8B-Instruction proves to be highly proficient. It demonstrates excellent control over both output length and section count in both English and Chinese, consistently aligning with the target performance. This indicates a strong foundational capability for generating well-structured data when the conceptual complexity remains low. Several other models also perform competently in these simpler scenarios, though Llama3.1 often has a slight edge.

The introduction of complexity, however, creates a significant performance divergence among the models. In these more demanding tasks, many models that performed well previously begin to struggle. GPT-4o mini, for instance, showcases a very interesting performance curve. It reliably handles complex tasks up to a medium scale, around 200 sections, but its performance noticeably degrades when pushed to the 500-section limit. This suggests a robust general capability that is not yet fully optimized for extreme long-context generation, revealing a clear performance ceiling.

In stark contrast, the LongWriter-llama3.1-8B model excels dramatically in the Complex-CH setting, where its output for both length and sections far surpasses all competitors, especially at the 500-section mark. This reinforces the finding that its specialized long-context training in Chinese provides a decisive advantage for complex, domain-specific tasks in that language. Meanwhile, the

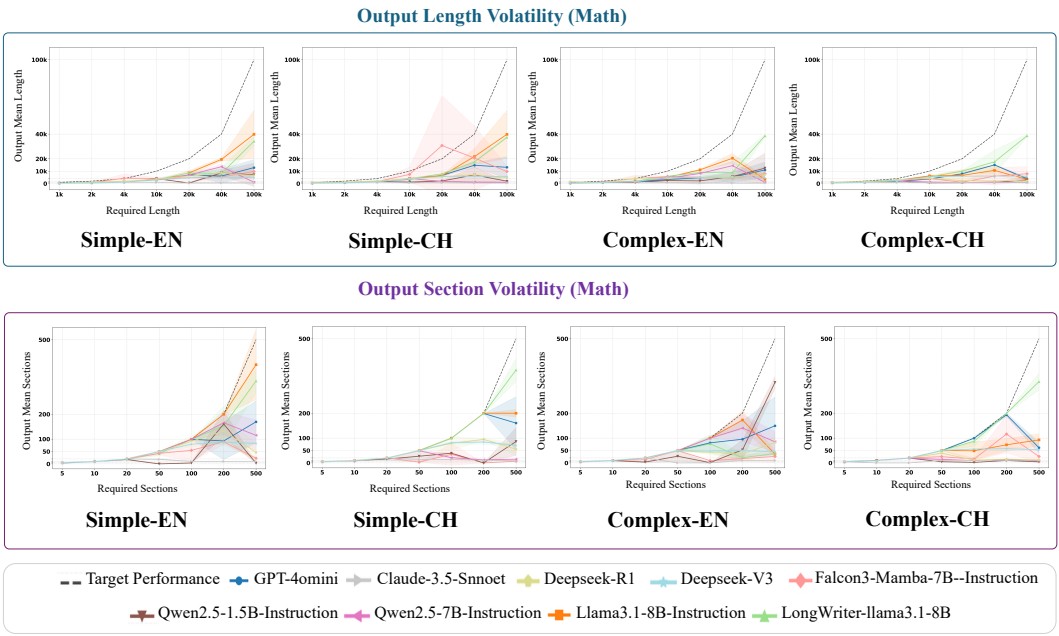

Figure 17: Comparison of output control for various large language models on math latex function generation tasks. The figure presents eight plots evaluating model performance across two languages (English and Chinese) and two complexity levels (simple and complex). The four columns correspond to the conditions: Simple-EN, Simple-CH, Complex-EN, and Complex-CH. The top row compares the actual mean output length against the required length. The bottom row compares the actual mean number of output sections against the required number of sections. In each plot, the solid lines represent the mean performance for each model, and the surrounding shaded areas indicate the volatility of the outputs. The dashed line indicates the target performance, where the model's output perfectly matches the specified requirement.

Mamba-7B model again exhibits a specific failure mode in the Simple-CH task, generating a high volume of text that lacks the required sectional structure, indicating a loss of high-level control.

In summary, the math formula generation task serves as an effective benchmark. It highlights that while some models are adept at simpler structured generation, complex and long-form tasks expose significant architectural or training limitations in most baseline models.

### J.0.7 COMPANY INFO TASK

Figure 18 reveals nuanced performance characteristics and specific failure modes among the various models when generating structured company information. The task's requirement for strict adherence to a predefined format, especially across long contexts, makes it particularly useful for evaluating model reliability and control under load. It tests not just the ability to generate fluent text, but to maintain a rigid structural template over thousands of tokens.

In tasks with simple complexity, several models perform capably. Llama3.1-8B-Instruction, for example, demonstrates good control, and GPT-4o mini also shows strong results. However, a subtle weakness in GPT-4o mini is observable even here, as its performance shows a slight decline when the required section count approaches the 500-section maximum. This suggests that even top-tier models have clear operational boundaries where stability can falter.

The models' behaviors diverge more dramatically in the complex scenarios. Llama3.1-8B-Instruction, despite its strength in simple tasks, becomes highly unstable. In the Complex-CH setting, it produces a large volume of text but fails to structure it into the required number of sections. Its high output volatility underscores this instability, suggesting it effectively loses its ability to follow formatting instructions under complex constraints.

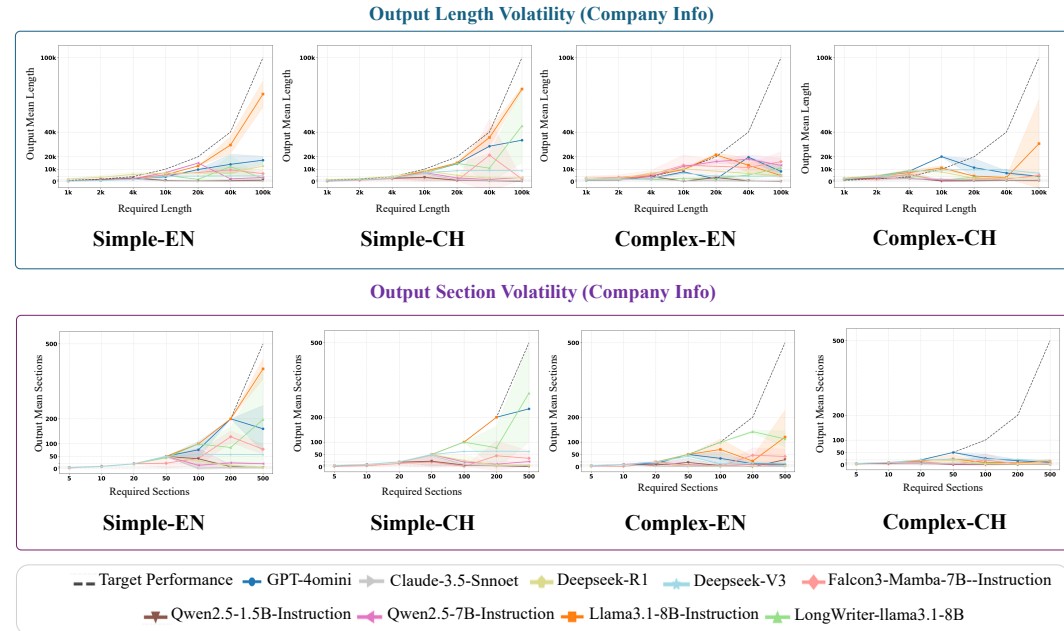

Figure 18: Comparison of output control for various large language models on company info generation tasks. The figure presents eight plots evaluating model performance across two languages (English and Chinese) and two complexity levels (simple and complex). The four columns correspond to the conditions: Simple-EN, Simple-CH, Complex-EN, and Complex-CH. The top row compares the actual mean output length against the required length. The bottom row compares the actual mean number of output sections against the required number of sections. In each plot, the solid lines represent the mean performance for each model, and the surrounding shaded areas indicate the volatility of the outputs. The dashed line indicates the target performance, where the model's output perfectly matches the specified requirement.

LongWriter-llama3.1-8B presents a different, equally interesting profile. Its performance on Chinese tasks is, on average, the highest among all models, a testament to the effectiveness of its supervised finetuning on Chinese long-text data. However, this high average performance is coupled with extreme volatility. The wide variance in its output indicates that while it is capable of generating very long and well-structured text, it is not consistently reliable. For any given attempt, it may succeed brilliantly or fail completely, rendering it a powerful but imperfect tool for tasks demanding predictability. These results highlight a crucial trade-off between achieving peak performance and ensuring stable, reliable generation.

### J.0.8    USER INFO TASK

Figure 19 presents the model evaluation results for the GenData task of creating structured user information. This task tests the models' ability to generate content that is not only long but also conforms to a specific, repetitive format, providing a clear measure of their instruction-following capabilities over extended contexts.

In the simple-difficulty tasks, several models demonstrate strong performance. Llama3.1-8B-Instruction is particularly noteworthy, especially in the Simple-EN scenario, where its section output almost perfectly matches the target requirements, indicating it successfully generated nearly all requested content. GPT-4o mini also performs reliably in these simpler settings, but it exhibits clear signs of fatigue at the upper end of the scale. Its performance noticeably falters when moving from the 200 to the 500-section requirement, suggesting that it is approaching the limits of its effective long-context capabilities for this type of structured generation.

The challenge intensifies significantly in the complex-difficulty tasks. Here, a universal trend of performance degradation is observed across all models, with most showing a sharp decline in adher-

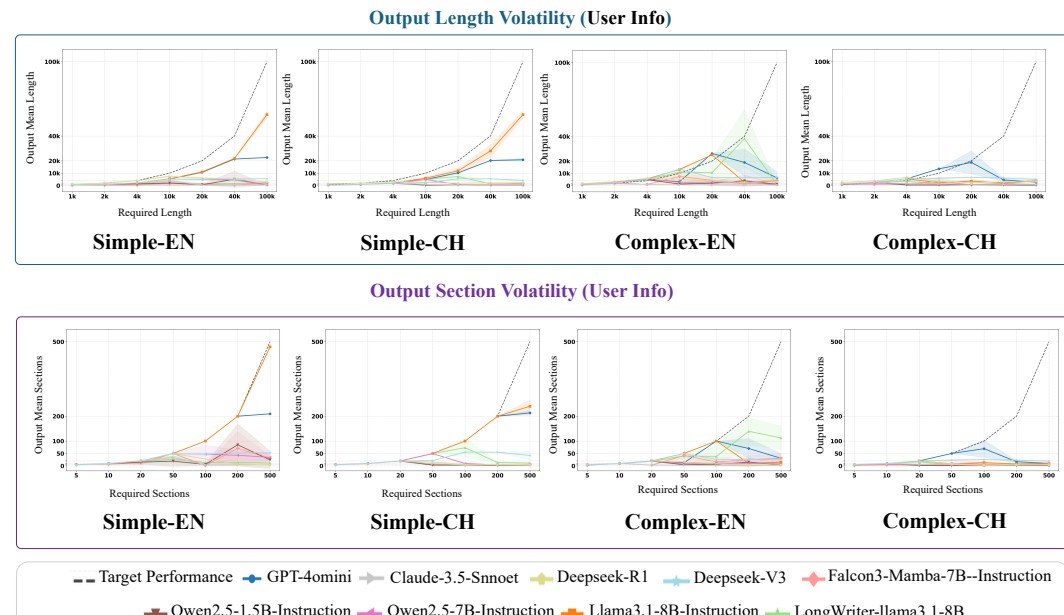

Figure 19: Comparison of output control for various large language models on user info generation tasks. The figure presents eight plots evaluating model performance across two languages (English and Chinese) and two complexity levels (simple and complex). The four columns correspond to the conditions: Simple-EN, Simple-CH, Complex-EN, and Complex-CH. The top row compares the actual mean output length against the required length. The bottom row compares the actual mean number of output sections against the required number of sections. In each plot, the solid lines represent the mean performance for each model, and the surrounding shaded areas indicate the volatility of the outputs. The dashed line indicates the target performance, where the model's output perfectly matches the specified requirement.

ence when the required sections exceed 200. This underscores the difficulty of maintaining structural integrity under complex constraints.

Most strikingly, this task reveals a critical limitation in the LongWriter model. Despite its previously demonstrated strengths in Chinese long-form generation, it performs exceptionally poorly on this specific task, especially in the Chinese scenarios. Its output is far worse than its own base model, Llama3.1. This strongly suggests that its supervised finetuning process did not include this type of structured user data. The result is a model that has become highly specialized, losing its general capability on out-of-domain tasks to the point of underperforming its un-tuned predecessor. This highlights the double-edged nature of supervised finetuning and the critical importance of training data diversity.

### J.0.9 EVALUATION SCORES

Table 27 provides a detailed performance comparison of the evaluated models on a 5-section generation task, focusing on two key dimensions: Length Volatility and Generation Quality. The results clearly demonstrate the effectiveness of our proposed method in producing highly stable and accurate outputs.

In terms of length volatility, our approach achieves a Length Variation Coefficient (LVC) of just 1.9%, the lowest among all tested models. This indicates exceptional stability in the length of generated content relative to its mean, significantly surpassing strong baselines like Deepseek-V3 (2.4%) and GPT-4omini (3.9%). In contrast, models such as Llama3.1 and LongWriter exhibit extremely high LVC values of 33.4% and 44.7% respectively, highlighting their unpredictability in output length. While our method's Mean Length Accuracy of 49.2% is moderate, it is important

Table 27: Performance comparison of evaluated models on a 5-section generation task, conducted in English under simple difficulty settings. Representative results are shown for an unstructured task (Story) and a structured task (Code Function). For the LSD and FAD metrics, the values in parentheses provide context by showing the generated mean length (in words) and mean section count, respectively. The "±" values represent the standard deviation. The arrows (↑/↓) indicate whether higher or lower values are preferable for each metric.

| Model | Length Volatility | | | Generation Quality | | |
|---|---|---|---|---|---|---|
| | LSD (↓) | LVC (↓) | MLA (↑) | FAD (↓) | SCA (↑) | UCA (↑) |
| GPT-4omini | 23.16 (590) | 3.9% | 59.0% | 0.00 (5.00) | 100.0% (±0.0%) | 97.3% (±2.5%) |
| Claude-3.5-Sonnet | 45.99 (437) | 10.5% | 43.7% | 0.00 (5.00) | 100.0% (±0.0%) | 94.7% (±3.4%) |
| Deepseek-R1 | 143.81 (923) | 15.6% | 92.3% | 0.00 (5.00) | 84.0% (±8.0%) | 98.0% (±1.6%) |
| Deepseek-V3 | 13.59 (562) | 2.4% | 56.2% | 0.00 (5.00) | 80.0% (±0.0%) | 95.3% (±4.5%) |
| Mamba-7B | 74.77 (400) | 18.7% | 40.0% | 0.00 (5.00) | 84.0% (±15.0%) | 87.3% (±9.3%) |
| Qwen2.5-1.5B | 82.10 (249) | 32.9% | 24.9% | 0.00 (5.00) | 68.0% (±24.0%) | 78.0% (±4.0%) |
| Qwen2.5-7B | 21.12 (495) | 4.3% | 49.5% | 0.00 (5.00) | 88.0% (±9.8%) | 95.3% (±6.2%) |
| Llama3.1-8B | 202.53 (606) | 33.4% | 60.6% | 0.94 (4.33) | 68.0% (±24.0%) | 88.0% (±7.5%) |
| LongWriter-8B | 262.46 (587) | 44.7% | 58.7% | 2.00 (3.00) | 80.0% (±0.0%) | 92.7% (±4.9%) |
| Ours | 28.35 (1504) | 1.9% | 49.2% | 0.00 (5.00) | 100.0% (±0.0%) | 96.7% (±2.9%) |

to note that this is based on a much larger mean output of 1504 words, showing that our model produces consistently longer, stable text rather than strictly adhering to a shorter target.

Regarding generation quality, our method excels across all metrics. It achieves a perfect Format Adherence Deviation (FAD) score of 0.00, consistently generating the required five sections without error. This stands in sharp contrast to LongWriter, which struggled significantly with a FAD of 2.00, on average producing only three of the five required sections. Furthermore, for structured tasks, our method attains a flawless 100% Structured Content Accuracy (SCA), a benchmark also met only by GPT-4omini and Claude-3.5-Sonnet. For unstructured content, our model's Unstructured Content Accuracy (UCA) of 96.7% is on par with the top-performing models, confirming its high quality.

In summary, our approach sets a new standard for reliable long-text generation. It uniquely combines state-of-the-art content quality and format adherence with unparalleled output stability, addressing the critical issue of volatility that affects many other leading models.

Table 28 extends the evaluation to a more demanding 10-section generation task, providing deeper insights into model scalability and robustness. The results from this scaled-up experiment further underscore the superior stability and quality of our proposed method, particularly as task complexity increases.

Our approach continues to demonstrate exceptional control over its output. It maintains a very low Length Variation Coefficient (LVC) of 2.7%, second only to the highly stable Mamba model (1.6%). However, this stability is achieved while generating a mean output of 2478 words, more than double that of any other model, and with a strong Mean Length Accuracy (MLA) of 76.1%. This unique combination of producing lengthy, stable, and accurate outputs sets our method apart. In contrast, models like Llama3.1 and LongWriter become almost uncontrollably volatile at this scale, with LVC values soaring to 64.4% and 69.6% respectively.

In the dimension of generation quality, our method's performance is flawless. It achieves a perfect Format Adherence Deviation (FAD) of 0.00 and a perfect Structured Content Accuracy (SCA) of 100.0%. This is a critical result, as several other strong models begin to falter at this increased length. For instance, Claude-3.5-Sonnet, which was perfect on the 5-section task, now shows significant format deviation (FAD of 2.36), and Llama3.1 also struggles to maintain the correct section count. While the unstructured content quality (UCA) remains high for our model at 96.4%, it is clear that maintaining structural integrity over longer generations is a key challenge that our method successfully overcomes.

Table 28: Performance comparison of evaluated models on a 10-section generation task, conducted in English under simple difficulty settings. Representative results are shown for an unstructured task (Story) and a structured task (Code Function). For the LSD and FAD metrics, the values in parentheses provide context by showing the generated mean length (in words) and mean section count, respectively. The "±" values represent the standard deviation. The arrows (↑/↓) indicate whether higher or lower values are preferable for each metric.

| Model | Length Volatility | | | Generation Quality | | |
|---|---|---|---|---|---|---|
| | LSD (↓) | LVC (↓) | MLA (↑) | FAD (↓) | SCA (↑) | UCA (↑) |
| GPT-4omini | 41.35 (1066) | 3.9% | 53.3% | 0.00 (10.00) | 96.0% (±4.9%) | 92.0% (±3.4%) |
| Claude-3.5-Sonnet | 200.53 (696) | 28.8% | 34.8% | 2.36 (8.33) | 100.0% (±0.0%) | 98.0% (±2.7%) |
| Deepseek-R1 | 39.42 (1220) | 3.2% | 61.0% | 0.00 (10.00) | 92.0% (±4.0%) | 98.0% (±1.6%) |
| Deepseek-V3 | 23.61 (827) | 2.9% | 41.4% | 0.00 (10.00) | 90.0% (±0.0%) | 94.0% (±3.9%) |
| Mamba-7B | 9.90 (607) | 1.6% | 30.4% | 0.00 (10.00) | 90.0% (±0.0%) | 80.7% (±12.5%) |
| Qwen2.5-1.5B | 319.44 (656) | 48.7% | 32.8% | 0.00 (10.00) | 90.0% (±0.0%) | 82.7% (±12.5%) |
| Qwen2.5-7B | 136.05 (745) | 18.3% | 37.2% | 0.00 (10.00) | 90.0% (±0.0%) | 94.0% (±2.5%) |
| Llama3.1-8B | 418.58 (650) | 64.4% | 32.5% | 3.30 (5.33) | 98.0% (±4.0%) | 91.3% (±5.0%) |
| LongWriter-8B | 956.53 (1374) | 69.6% | 68.7% | 4.24 (7.00) | 80.0% (±15.5%) | 86.7% (±11.0%) |
| Ours | 67.35 (2478) | 2.7% | 76.1% | 0.00 (10.00) | 100.0% (±0.00%) | 96.4% (±11.0%) |

In conclusion, as the generation length and structural requirements increase, the advantages of our approach become even more pronounced. It consistently delivers high-quality, structurally perfect content with low volatility, while many other models exhibit a significant degradation in either stability or format adherence.

Table 29: Performance comparison of evaluated models on a 20-section generation task, conducted in English under simple difficulty settings. Representative results are shown for an unstructured task (Story) and a structured task (Code Function). For the LSD and FAD metrics, the values in parentheses provide context by showing the generated mean length (in words) and mean section count, respectively. The "±" values represent the standard deviation. The arrows (↑/↓) indicate whether higher or lower values are preferable for each metric.

| Model | Length Volatility | | | Generation Quality | | |
|---|---|---|---|---|---|---|
| | LSD (↓) | LVC (↓) | MLA (↑) | FAD (↓) | SCA (↑) | UCA (↑) |
| GPT-4omini | 317.35 (2068) | 15.3% | 51.7% | 0.00 (20.00) | 98.0% (±2.4%) | 92.0% (±1.6%) |
| Claude-3.5-Sonnet | 13.57 (181) | 7.5% | 4.5% | 0.47 (2.67) | 100.0% (±0.0%) | 88.7% (±4.5%) |
| Deepseek-R1 | 37.42 (1853) | 2.0% | 46.3% | 0.00 (20.00) | 97.0% (±2.4%) | 93.3% (±6.0%) |
| Deepseek-V3 | 130.08 (1168) | 11.1% | 29.2% | 0.00 (20.00) | 95.0% (±0.0%) | 92.0% (±5.8%) |
| Mamba-7B | 282.37 (351) | 80.5% | 8.8% | 4.00 (10.00) | 93.0% (±2.4%) | 76.0% (±4.9%) |
| Qwen2.5-1.5B | 279.49 (414) | 67.5% | 10.4% | 8.53 (10.50) | 94.0% (±3.7%) | 81.3% (±4.5%) |
| Qwen2.5-7B | 104.34 (915) | 11.4% | 22.9% | 0.00 (20.00) | 95.0% (±0.0%) | 92.0% (±7.5%) |
| Llama3.1-8B | 4.92 (268) | 1.8% | 6.7% | 0.47 (2.67) | 98.0% (±2.4%) | 88.7% (±8.1%) |
| LongWriter-8B | 759.89 (3713) | 20.5% | 92.8% | 8.23 (15.25) | 92.0% (±6.0%) | 72.0% (±13.1%) |
| Ours | 159.63 (4235) | 3.8% | 92.8% | 8.23 (15.25) | 98.0% (±2.0%) | 92.0% (±7.1%) |

The evaluation detailed in Table 29 assesses model performance on a highly demanding 20-section generation task. This increased complexity reveals significant trade-offs between output stability, content quality, and structural adherence, providing a more granular view of each model's capabilities and limitations under substantial load.

In terms of output stability, our method maintains a highly competitive Length Variation Coefficient (LVC) of 3.8%. While some models like Deepseek-R1 achieve even greater relative stability with an LVC of 2.0%, our model's performance is notable as it is accomplished while generating by far the longest average output at 4235 words. Compared to another high-performing baseline, GPT-4omini, which has an LVC of 15.3%, our approach proves to be over 75% more stable in its relative output

length. This demonstrates our model's ability to control its generation reliably even when producing vast amounts of text, a stark contrast to models like Mamba-7B, whose stability collapses at this scale.

When evaluating content quality, our model performs at the top tier. Its Structured Content Accuracy (SCA) of 98.0% is tied for the highest score, marginally outperforming the excellent result of Deepseek-R1 (97.0%). Similarly, its Unstructured Content Accuracy (UCA) of 92.0% is highly competitive. However, the data reveals a critical failure point for our method at this scale: structural integrity. Our model recorded a Format Adherence Deviation (FAD) of 8.23, indicating it failed to generate the required 20 sections, instead averaging only 15.25. This performance is poor compared to models like Deepseek-R1 and GPT-4omini, which maintained perfect format adherence with a FAD of 0.00.

In conclusion, the 20-section task highlights that while our method excels in generating high-quality content at an unprecedented scale with low relative volatility, its ability to follow rigid structural rules can break down under extreme pressure. This presents a crucial area for future improvement, contrasting with models like Deepseek-R1 that provide a more balanced, albeit less lengthy, performance across all metrics.

Table 30: Performance comparison of evaluated models on a 50-section generation task, conducted in English under simple difficulty settings. Representative results are shown for an unstructured task (Story) and a structured task (Code Function). For the LSD and FAD metrics, the values in parentheses provide context by showing the generated mean length (in words) and mean section count, respectively. The "±" values represent the standard deviation. The arrows (↑/↓) indicate whether higher or lower values are preferable for each metric.

| Model | Length Volatility | | | Generation Quality | | |
|---|---|---|---|---|---|---|
| | LSD (↓) | LVC (↓) | MLA (↑) | FAD (↓) | SCA (↑) | UCA (↑) |
| GPT-4omini | 418.82 (5526) | 7.6% | 55.3% | 4.78 (41.67) | 79.6% (±25.0%) | 77.3% (±12.9%) |
| Claude-3.5-Sonnet | 10.62 (155) | 6.8% | 1.6% | 0.00 (2.00) | 11.2% (±4.7%) | 85.3% (±5.8%) |
| Deepseek-R1 | 463.27 (830) | 55.8% | 8.3% | 0.00 (3.00) | 99.6% (±0.8%) | 90.7% (±3.9%) |
| Deepseek-V3 | 100.70 (1895) | 5.3% | 19.0% | 0.00 (50.00) | 98.4% (±0.8%) | 90.7% (±8.8%) |
| Mamba-7B | 620.80 (1518) | 40.9% | 15.2% | 18.86 (36.67) | 91.2% (±15.6%) | 78.0% (±7.2%) |
| Qwen2.5-1.5B | 425.01 (636) | 66.8% | 6.4% | 15.58 (15.00) | 29.2% (±33.6%) | 78.7% (±17.1%) |
| Qwen2.5-7B | 302.55 (1367) | 22.1% | 13.7% | 9.43 (43.33) | 98.8% (±1.6%) | 89.3% (±10.2%) |
| Llama3.1-8B | 77.15 (277) | 27.8% | 2.8% | 1.25 (3.33) | 99.2% (±1.0%) | 86.0% (±19.0%) |
| LongWriter-8B | 3918.92 (5148) | 76.1% | 51.5% | 21.91 (38.75) | 71.6% (±30.0%) | 74.7% (±10.9%) |
| Ours | 297.28 (8056) | 3.7% | 80.5% | 2.50 (45.00) | 99.5% (±0.5%) | 90.2% (±5.5%) |

The results presented in Table 30, derived from an extreme 50-section generation task, effectively push the models to their operational limits. At this substantial scale, most models experience a severe degradation in performance, highlighting the immense challenge of maintaining coherence, stability, and structural integrity over very long contexts. In this demanding scenario, our proposed method and Deepseek-V3 emerge as the only two models capable of delivering high-quality results.

A direct comparison reveals the distinct advantages of our approach. In terms of stability, our model achieves a Length Variation Coefficient (LVC) of 3.7%, a figure that is over 30% lower than Deepseek-V3's LVC of 5.3%. This superior stability is all the more impressive given that our model generated an average of 8056 words, more than four times the output length of Deepseek-V3. Furthermore, our model's Mean Length Accuracy (MLA) of 80.5% far surpasses Deepseek-V3's 19.0%, indicating our output length is significantly closer to the intended target. For content fidelity, our model's Structured Content Accuracy (SCA) of 99.5% is approximately 1.1% higher than Deepseek-V3's already excellent 98.4%.

While Deepseek-V3 achieves perfect structural integrity by delivering all 50 sections (FAD of 0.00), our model shows a minor deviation, averaging 45 sections (FAD of 2.50). However, this slight

shortfall in section count is offset by its superior performance across stability, length accuracy, and content quality. The widespread failure of other prominent models, most of which could not generate even a third of the required sections, contextualizes the exceptional performance of these two top-tier models.

In conclusion, at the frontier of long-text generation, our method demonstrates a state-of-the-art capability, producing outputs of unprecedented length with superior stability and content accuracy, establishing its leadership in extreme-scale generative tasks.

Table 31: Performance comparison of evaluated models on a 100-section generation task, conducted in English under simple difficulty settings. Representative results are shown for an unstructured task (Story) and a structured task (Code Function). For the LSD and FAD metrics, the values in parentheses provide context by showing the generated mean length (words) and mean section count, respectively. The "±" values represent the standard deviation. The arrows (↑/↓) indicate whether higher or lower values are preferable for each metric.

| Model | Length Volatility | | | Generation Quality | | |
|---|---|---|---|---|---|---|
| | LSD (↓) | LVC (↓) | MLA (↑) | FAD (↓) | SCA (↑) | UCA (↑) |
| GPT-4o mini | 325.65 (959) | 33.9% | 4.8% | 1.41 (7.00) | 84.6% (±30.8%) | 86.7% (±6.7%) |
| Claude-3.5-Sonnet | 3.30 (176) | 1.9% | 0.9% | 0.00 (2.00) | 3.0% (±0.0%) | 88.7% (±2.7%) |
| Deepseek-R1 | 103.30 (1198) | 8.6% | 6.0% | 1.25 (4.33) | 35.0% (±13.2%) | 93.3% (±3.7%) |
| Deepseek-V3 | 40.76 (1854) | 2.2% | 9.3% | 1.70 (20.67) | 48.6% (±3.8%) | 84.7% (±3.4%) |
| Mamba-7B | 715.98 (1291) | 55.5% | 6.5% | 41.72 (40.75) | 66.8% (±21.9%) | 76.0% (±17.3%) |
| Qwen2.5-1.5B | 27.78 (142) | 19.6% | 0.7% | 0.47 (1.67) | 15.6% (±24.0%) | 84.0% (±7.1%) |
| Qwen2.5-7B | 75.87 (445) | 17.0% | 2.2% | 2.05 (10.33) | 99.8% (±0.4%) | 86.7% (±7.6%) |
| Llama3.1-8B | 92.77 (350) | 26.5% | 1.7% | 0.94 (4.33) | 92.4% (±14.2%) | 82.0% (±18.9%) |
| LongWriter-8B | 2866.29 (6320) | 45.4% | 31.6% | 21.42 (45.00) | 32.6% (±31.9%) | 66.7% (±16.5%) |
| Ours | 2194.23 (15651) | 14.02% | 78.25% | 7.24 (88.00) | 100% (±0%) | 86.7% (±16.5%) |

The comprehensive results from the 100-section generation task, detailed in Table 31, serve to starkly differentiate the capabilities of our proposed method from all evaluated baseline models. While the baselines universally struggle to cope with the task's demanding scale, our approach demonstrates a significant leap forward in long-context generation, particularly in task completion and content accuracy.

The most critical distinction lies in the ability to maintain structural integrity. Our method successfully generated an average of 88.00 sections, effectively completing the vast majority of the task. This performance dwarfs that of the best baseline, LongWriter-8B, which produced only 45.00 sections. This means our approach generated over 95% more of the required structured content than the strongest competitor, nearly doubling its effective output. Other powerful models like GPT-4o mini and Deepseek-V3 failed much earlier, delivering less than a quarter of the required sections and underscoring their limitations at this scale.

Furthermore, our model achieved this superior structural output without sacrificing quality. It recorded a perfect 100% Structured Content Accuracy (SCA) across the 88 sections it produced. This combination of scale and accuracy is unique; no other model came close to this performance. For instance, the baseline with the next highest section count, LongWriter-8B, had a comparatively poor SCA of only 32.6%. While some models like Qwen2.5-7B posted a high SCA, it was on a trivial output of only 10 sections, highlighting an inability to maintain quality at scale.

Finally, this state-of-the-art performance in quality and structure was achieved while generating an enormous average output of 15,651 words with a reasonable Length Variation Coefficient (LVC) of 14.02%. In conclusion, our method demonstrates a paradigm shift, successfully balancing the competing demands of extreme length, perfect content accuracy, and stable generation far beyond the capabilities of current baseline models.

Table 32: Performance comparison of evaluated models on a 200-section generation task, conducted in English under simple difficulty settings. Representative results are shown for an unstructured task (Story) and a structured task (Code Function). For the LSD and FAD metrics, the values in parentheses provide context by showing the generated mean length (in words) and mean section count, respectively. The "±" values represent the standard deviation. The arrows (↑/↓) indicate whether higher or lower values are preferable for each metric.

| Model | Length Volatility | | | Generation Quality | | |
|---|---|---|---|---|---|---|
| | LSD (↓) | LVC (↓) | MLA (↑) | FAD (↓) | SCA (↑) | UCA (↑) |
| GPT-4omini | 343.48 (5594) | 6.1% | 14.0% | 1.70 (23.33) | 60.6% (±39.3%) | 80.7% (±6.8%) |
| Claude-3.5-Sonnet | 3.86 (196) | 2.0% | 0.5% | 0.00 (2.00) | 1.5% (±0.0%) | 88.7% (±7.5%) |
| Deepseek-R1 | 168.55 (1360) | 12.4% | 3.4% | 1.25 (4.67) | 8.0% (±2.4%) | 92.7% (±5.3%) |
| Deepseek-V3 | 415.16 (1530) | 27.1% | 3.8% | 4.71 (16.67) | 22.2% (±2.7%) | 88.7% (±12.8%) |
| Mamba-7B | 37.06 (152) | 24.3% | 0.4% | 2.00 (8.00) | 54.0% (±26.9%) | 78.7% (±7.8%) |
| Qwen2.5-1.5B | 141.80 (355) | 40.0% | 0.9% | 2.36 (2.67) | 22.1% (±38.9%) | 88.7% (±5.4%) |
| Qwen2.5-7B | 127.46 (571) | 22.3% | 1.4% | 3.30 (8.33) | 45.0% (±44.9%) | 86.0% (±20.0%) |
| Llama3.1-8B | 36.55 (301) | 12.1% | 0.8% | 0.47 (2.67) | 99.7% (±0.6%) | 85.3% (±7.8%) |
| LongWriter-8B | 2858.29 (6353) | 45.0% | 15.9% | 17.30 (31.75) | 30.5% (±36.2%) | 66.0% (±16.7%) |
| Ours | 3743.92 (31582) | 11.85% | 78.96% | 5.00 (147.20) | 90.5% (±5.2%) | 87.0% (±10.4%) |

Table 32 details the results of the final and most rigorous evaluation: a 200-section generation task. This extreme stress test is designed to push models far beyond their conventional limits, and the results clearly demonstrate a near-universal failure among all baseline models. In this challenging environment, our proposed method stands alone in its ability to handle the task's immense scale and complexity.

The performance of the baseline models collapses under this load. An examination of the Format Adherence Deviation (FAD) reveals that even the most powerful models failed to generate a meaningful portion of the required content. For instance, GPT-4omini produced an average of only 23 sections, while Deepseek-V3 managed just 17. The data also highlights a potential for misinterpretation; models like Llama3.1 report a near-perfect Structured Content Accuracy (SCA) of 99.7%, but this accuracy is measured on a trivial output of only two to three sections, indicating a complete failure to adhere to the primary task constraint of generating 200 sections.

In stark contrast, our method is the only one to successfully navigate this challenge. It generated an average of 147.2 sections out of the required 200, producing over 4.6 times more of the target content than the next-best model, LongWriter, which averaged only 31.75 sections. Crucially, this massive output was generated with exceptional quality, achieving a 90.5% SCA and an 87.0% Unstructured Content Accuracy (UCA). Furthermore, our model maintained a commendable Length Variation Coefficient (LVC) of 11.85% across an unprecedented average output length of over 31,000 words, demonstrating robust control at a scale where other models falter.

In conclusion, the 200-section task decisively establishes the state-of-the-art capability of our approach. It is the only evaluated method that successfully scales to extreme-length generation, delivering the vast majority of the required content while preserving high levels of accuracy and stability.

The final evaluation detailed in Table 33 subjects the models to an immense 500-section generation task. This extreme benchmark pushes every model beyond its designed limits, revealing distinct modes of failure and decisively highlighting the unique resilience and state-of-the-art capability of our approach. At this scale, a clear distinction emerges between models that fail gracefully and those that attempt the task.

Interestingly, powerful closed-source models like GPT 4o mini and Claude-3.5-Sonnet exhibit what can be described as an "intelligent failure". Rather than attempting to generate the full 500 sections, a task they likely identify as beyond their context limits, they produce a severely truncated output, av-

Table 33: Performance comparison of evaluated models on a 500-section generation task, conducted in English under simple difficulty settings. Representative results are shown for an unstructured task (Story) and a structured task (Code Function). For the LSD and FAD metrics, the values in parentheses provide context by showing the generated mean length (in words) and mean section count, respectively. The "±" values represent the standard deviation. The arrows (↑/↓) indicate whether higher or lower values are preferable for each metric.

| Model | Length Volatility | | | Generation Quality | | |
|---|---|---|---|---|---|---|
| | LSD (↓) | LVC (↓) | MLA (↑) | FAD (↓) | SCA (↑) | UCA (↑) |
| GPT-4omini | 2332.12 (3670) | 63.5% | 3.7% | 13.07 (19.33) | 32.6% (±18.1%) | 82.7% (±12.9%) |
| Claude-3.5-Sonnet | 30.68 (188) | 16.3% | 0.2% | 0.71 (2.00) | 0.6% (±0.0%) | 83.3% (±3.7%) |
| Deepseek-R1 | 93.77 (1018) | 9.2% | 1.0% | 0.82 (3.00) | 3.4% (±1.2%) | 94.7% (±4.0%) |
| Deepseek-V3 | 110.80 (1357) | 8.2% | 1.4% | 3.14 (9.33) | 9.4% (±1.1%) | 82.7% (±1.3%) |
| Mamba-7B | 8164.97 (30667) | 26.6% | 30.7% | 4.90 (10.00) | 20.8% (±11.4%) | 54.0% (±14.7%) |
| Qwen2.5-1.5B | 22.10 (151) | 14.7% | 0.2% | 0.49 (1.00) | 1.8% (±1.1%) | 76.0% (±16.7%) |
| Qwen2.5-7B | 112.34 (516) | 21.77 | 0.5% | 0.98 (3.00) | 33.7% (±39.1%) | 78.7% (±6.2%) |
| Llama3.1-8B | 25.69 (294) | 8.7% | 0.3% | 0.75 (3.33) | 28.2% (±39.1%) | 86.0% (±5.3%) |
| LongWriter-8B | 10083.17 (50604) | 19.9% | 50.6% | 7.78 (16.50) | 26.8% (±27.1%) | 62.0% (±15.4%) |
| Ours | 5078.4 (59534) | 8.5% | 59.5% | 12.44 (327.20) | 66.8% (±17.5%) | 82.0% (±5.4%) |

eraging only 19 and 2 sections, respectively. This behavior suggests a sophisticated mechanism that opts to provide a summary or structural outline instead of failing catastrophically mid-generation. Other baseline models either fail early or, like LongWriter, attempt to meet the length requirement but completely lose structural control, resulting in long but incoherent output.

In this landscape of widespread failure, our method is the only one that successfully rises to the challenge. It is the sole model to generate a substantial portion of the request, delivering an average of 327.2 sections. This is an unparalleled achievement, representing nearly 20 times more of the required content than the next closest competitor, LongWriter, which produced only 16.5 sections. Crucially, our model maintains a respectable Structured Content Accuracy (SCA) of 66.8% and Unstructured Content Accuracy (UCA) of 82.0% across a massive average output of nearly 60,000 words. Its Length Variation Coefficient (LVC) of 8.5% is the most meaningful stability metric in the table, as it is the only one tied to a successful, large-scale generation.

In conclusion, the 500-section task proves that our method operates in a class of its own. It is the only evaluated approach capable of scaling to extreme-length tasks while substantially preserving structural integrity and content quality, confirming its breakthrough status in long-context generation.

The performance evaluation detailed in Table 34 shifts the focus to different task domains—specifically unstructured Diary generation and structured Math Latex generation. By comparing these results to the previous evaluation on Story and Code tasks, we can analyze the significant impact that task type has on the long-context capabilities of baseline models, even when the required length and complexity remain constant at 100 sections.

The overall difficulty of the 100-section benchmark remains evident, with most models still failing to complete the task. Powerful closed-source models like GPT-4o mini and Claude-3.5-Sonnet continue their pattern of failing early, generating only a small fraction of the required sections. This consistent behavior across different domains suggests their refusal to handle extreme-length requests is a core aspect of their operational logic, rather than a task-specific issue.

However, the most striking finding is the dramatic performance shift of specific models when the task changes. Qwen2.5-7B, which produced only about 10 sections on the Code task, demonstrates a remarkable improvement on the Math task, successfully generating an average of 79.33 sections. This represents a nearly eight-fold increase in effective output, making it the top-performing baseline

Table 34: Performance comparison of evaluated models on a 100-section generation task, conducted in English under simple difficulty settings. Representative results are shown for an unstructured task (Diary) and a structured task (Math Latex Function). For the LSD and FAD metrics, the values in parentheses provide context by showing the generated mean length (in words) and mean section count, respectively. The "±" values represent the standard deviation. The arrows (↑/↓) indicate whether higher or lower values are preferable for each metric.

| Model | Length Volatility | | | Generation Quality | | |
|---|---|---|---|---|---|---|
| | LSD (↓) | LVC (↓) | MLA (↑) | FAD (↓) | SCA (↑) | UCA (↑) |
| GPT-4omini | 2395.67 (2489) | 96.2% | 12.4% | 21.68 (22.33) | 99.8% (±0.4%) | 96.7% (±3.7%) |
| Claude-3.5.5-Sonnet | 38.18 (303) | 12.6% | 1.5% | 0.00 (2.00) | 11.0% (±2.0%) | 92.0% (±4.5%) |
| Deepseek-R1 | 271.73 (1626) | 16.7% | 8.1% | 0.47 (6.33) | 71.0% (±34.1%) | 98.0% (±4.0%) |
| Deepseek-V3 | 546.33 (853) | 64.0% | 4.3% | 9.53 (12.33) | 77.4% (±4.2%) | 86.7% (±6.0%) |
| Mamba-7B | 274.94 (934) | 29.4% | 4.7% | 57.59 (64.25) | 44.2% (±45.7%) | 79.2% (±14.8%) |
| Qwen2.5-1.5B | 32.06 (177) | 18.1% | 0.9% | 0.00 (1.00) | 5.6% (±4.1%) | 82.7% (±4.9%) |
| Qwen2.5-7B | 1155.46 (6999) | 16.5% | 35.0% | 29.23 (79.33) | 99.4% (±0.8%) | 80.7% (±4.9%) |
| Llama3.1-8B | 265.02 (632) | 42.0% | 3.2% | 4.11 (6.67) | 85.0% (±24.7%) | 84.0% (±15.8%) |
| LongWriter-8B | 3484.67 (4819) | 72.3% | 24.1% | 40.91 (46.00) | 82.0% (±29.6%) | 83.3% (±10.1%) |

by a significant margin. Crucially, it maintained a near-perfect Structured Content Accuracy of 99.4% across this vastly expanded output. This suggests the highly logical and formal syntax of LaTeX aligns better with its capabilities than the more abstract structure of code generation.

Other models also show notable changes. LongWriter-8B, while producing a similar number of sections as before, sees its SCA score improve dramatically from 32.6% on the Code task to 82.0% on the Math task. Conversely, Mamba-7B generates more sections but with a lower accuracy. In conclusion, these results prove that model performance at scale is not monolithic; it is highly dependent on the specific structural and logical demands of the task, revealing unique strengths and weaknesses that are not apparent from a single benchmark.

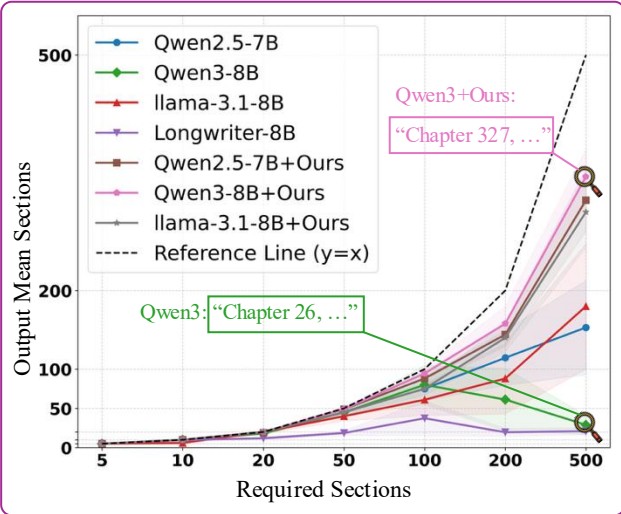

Figure 20: Section volatility of the model with our method. Baseline models often fail to generate a sufficient number of sections, whereas our model generates more sections with greater stability.

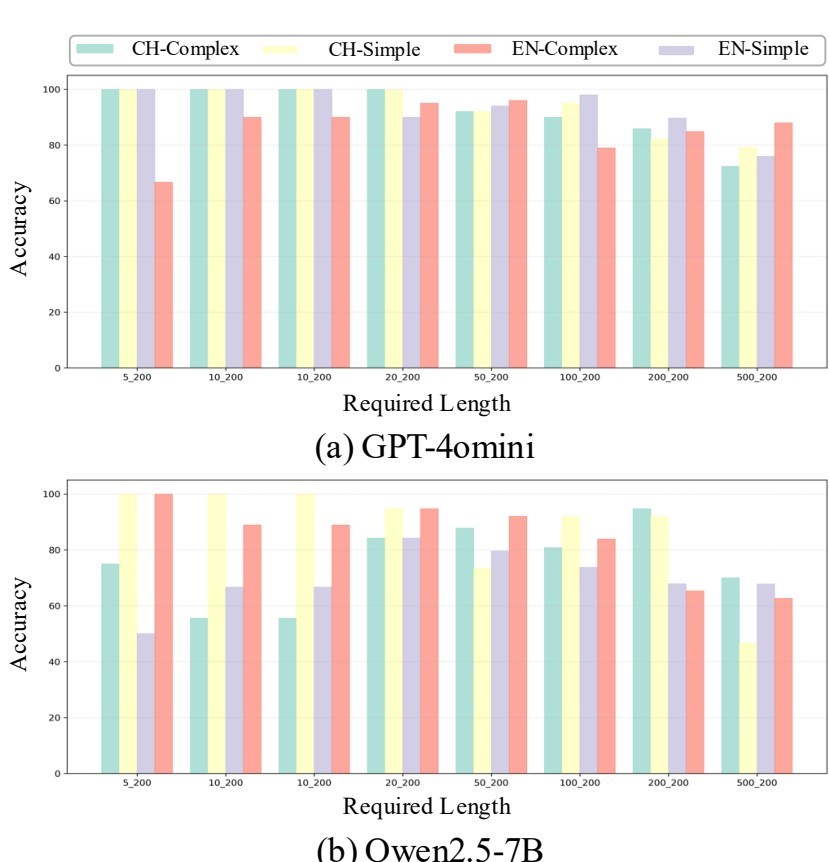

(a) GPT-4omini

(b) Qwen2.5-7B

Figure 21: Code Generation Accuracy Across Different Length Requirements. This figure illustrates the performance of (a) GPT-4omini and (b) Qwen2.5-7B across different languages (CH/EN) and instruction complexities (Simple/Complex). Two main conclusions can be drawn from the figure: First, as the required output length increases, the code generation accuracy of both models shows an overall downward trend. Second, the impact of instruction complexity on generation quality varies by model and language, under simple instructions, some models (such as Qwen2.5-7B on the English task) exhibit relatively lower accuracy, which may be attributed to instruction ambiguity.

