# OpenReview forum: "On Stable Long-Form Generation: Benchmarking and Mitigating Length Volatility"
_ICLR.cc/2026/Conference — Submitted to ICLR 2026_

### Official Review · Reviewer_Kq9F · 2025-10-18

**Soundness:** 3
**Presentation:** 3
**Contribution:** 3
**Rating:** 6
**Confidence:** 5

**Summary:**

This paper reframes long-form generation around a neglected reality: outputs vary wildly across samples, especially in length. It introduces VOLTBench to measure that volatility with clear stability metrics (variance/ratio and mean-length alignment) across structured and unstructured tasks. By visualizing attention over “constraint” tokens in the prompt, the authors surface two failure signatures—Attention Collapse (periodic peaks fade, the model drifts into repetition or early stop) and Attention Instability (a sudden oversized peak that tempts the model to jump straight to the final section). To counter this, they propose SELB, a training-free, decoding-time controller that boosts logits for next-section titles at length thresholds and suppresses EOS/boilerplate until the final section, yielding longer, steadier, and still coherent outputs.

**Strengths:**

The work is refreshingly practical: it defines stability as a first-class objective, offers an interpretable diagnostic (attention traces that align peaks with section starts), and delivers a simple plug-in method that doesn’t require retraining yet tracks task constraints directly. Evidence spans multiple models and tasks, showing volatility reductions and better length adherence without obvious quality collapse. The paper also documents metrics and templates clearly and commits to releasing code/benchmark, which should make these ideas easy to adopt.

**Weaknesses:**

1) The paper’s “quality” evidence leans on structural or task-specific signals (e.g., section adherence, execution checks) rather than human-perceived writing quality. For creative long-form text, improved stability and length alignment on VOLTBench do not necessarily guarantee unchanged or better writing quality versus strong baselines.

2) SELB addresses sampling volatility but does not clearly separate it from infrastructure nondeterminism. In creativity-oriented settings where temperature cannot be set very low, it remains unclear how much instability comes from stochastic decoding versus platform/kernel nondeterminism (fixed seeds, deterministic math, op ordering), as discussed in resources like https://thinkingmachines.ai/blog/defeating-nondeterminism-in-llm-inference/.

3) The method implicitly relies on explicit section templates by boosting “next-section title” tokens. This structural bias limits applicability to essays or narratives without clear headings, where attention peaks tied to title tokens are weak or absent and structural enforcement can become brittle or intrusive.

**Questions:**

ref weekness

---

> ### Author Response · Authors · 2025-11-20
> **(1/5) Responses by Authors**
>
> **Thanks for your careful and insightful reviews.**
> > **`Weakness 1`**: The paper’s quality evidence leans on structural or task-specific signals (e.g., section adherence, execution checks) rather than human-perceived writing quality. For creative long-form text, improved stability and length alignment on VOLTBench do not necessarily guarantee unchanged or better writing quality versus strong baselines.
>
> `Response`: Thank you for this insightful question. We would like to address your concerns using the following analysis:
>
> ### 1. **Content Quality Evaluation**
> Actually, our evaluation framework is not limited to structural signals. **Our UCA metric  was specifically designed to assess human-perceived writing quality**. The UCA uses an LLM-as-a-Judge  to evaluate core elements of creative long-form text, including:
> - **Breadth and Depth**: This directly penalizes the filler content you mentioned (judged as superficial, with minimal information).
> - **Coherence and Reading Experience**: This directly penalizes repetitive content (judged as disorganized and boring).
>
> As shown in Table below (from Table 2 in manuscript):
>
> **Table**. Performance comparison of evaluated models on a 100-section generation task, conducted in English under simple difficulty settings. Representative results are shown for an unstructured task (Story) and a structured task (Code Function).
> | Model                | LSD (↓)            | LVC (↓)  | MLA (↑) | FAD (↓)           | SCA (↑)           | UCA (↑)           |
> |----------------------|--------------------|----------|---------|-------------------|-------------------|-------------------|
> | GPT-4o mini          | 325.65 (959)       | 33.9%    | 4.8%    | 1.41 (7.00)       | 84.6% (±30.8%)    | 86.7% (±6.7%)     |
> | Deepseek-R1          | 103.30 (1198)      | 8.6%     | 6.0%    | 1.25 (4.33)       | 35.0% (±13.2%)    | 93.3% (±3.7%)     |
> | Llama3.1-8B          | 92.77 (350)        | 26.5%    | 1.7%    | 0.94 (4.33)       | 92.4% (±14.2%)    | 82.0% (±18.9%)    |
> | LongWriter-8B        | 2866.3 (6320)      | 45.4%    | 31.6%   | 21.42 (45.00)     | 32.6% (±31.9%)    | 66.7% (±16.5%)    |
> | Qwen2.5-7B           | 75.87 (445)        | 17.0%    | 2.2%    | 2.05 (10.33)      | 99.8% (±0.4%)     | 86.7% (±7.6%)     |
> | Repetition_penalties   | 553 (2967)         | 18.6%    | 14.8%   | 5.4 (22)          | 98% (±1%)         | 76.7% (±14.5%)    |
> | Stop-entropy   | 713 (2701)         | 26.4%    | 13.5%   | 7.24 (24)         | 95% (±2.5%)       | 83.9% (±8%)       |
> | Length_controllers | 1280 (4470)        | 28.65%   | 77.7%   | 9.2 (28)          | 96% (±2%)         | 85% (±9%)         |
> | Look-ahead   | 268 (2883)         | 9.3%     | 14.4%   | 7.2 (25)          | 94% (±3.5%)       | 84.4% (±8%)       |
> | SELB  | 2194.23 (15651) | 14.02% | 78.25% | 7.24 (88.00) | 100% (±0%) | 86.7% (±16.5%) |
>
> The data demonstrates that **SELB does not degrade the intrinsic quality of the base model**. Our method achieves a UCA score of **86.7%**, we did not introduce filler or repetition that would have lowered the quality score. In contrast, LongWriter-8B suffered a significant quality drop, achieving only 66.7% UCA. This highlights that standard long-context training can sometimes compromise writing quality, whereas our logits intervention preserves it.
>
> ### 2. **The SELB-Hybrid Strategy**
> To adapt our method for free-form tasks that lack explicit sections, we generalized the SELB strategy from section enforcement to **SELB-Hybrid** strategy, which focuses on length enforcement and stall prevention.
> - **Stop Token Suppression**: This is a direct generalization of our original proactive failure prevention logic. In the original method, we suppressed EOS tokens until the model reached the target section count ($P_{total}$). In the new strategy, the processor actively and aggressively suppresses all premature end-of-sequence (EOS) tokens and common conversational stop-phrases (e.g., I hope this helps, Let me know if you need...). This suppression remains active until the model reaches a preset target token count (e.g., 1.5 times the target word count $L$). This directly addresses the problem of models stopping prematurely before reaching the target length.
> - **Hybrid Keep-Alive Mechanism**: Without strong structural anchors such as chapter titles to boost, free-form generation faces a unique stall risk, such as falling into repetitive loops or silent failures (i.e., ceasing to generate new content). To solve this, we introduce a keep-alive mechanism: (1) The processor periodically (e.g., every $\tau_{max} = 500$ tokens) checks the generation state. (2) If the model fails to produce a natural interruption (like a period . or a newline \n) within a grace period (e.g., $k=100$ tokens), the processor determines the model has stalled. (3) It then nudges the model by boosting the logits of generic continuation tokens (e.g., \n and ), forcing it to break the current loop and continue generating new content.

---

> ### Author Response · Authors · 2025-11-20
> **(2/5) Responses by Authors**
>
> > **`Weakness 1`**: The paper’s quality evidence leans on structural or task-specific signals (e.g., section adherence, execution checks) rather than human-perceived writing quality. For creative long-form text, improved stability and length alignment on VOLTBench do not necessarily guarantee unchanged or better writing quality versus strong baselines.
>
> `Response`: (Continued)
> ### 3. **Evaluation on Free-Form Tasks**
> To rigorously test the SELB-Hybrid, we designed highly challenging free-form tasks. We require the model to generate a single, continuous document with a target length of 20,000 words, without providing any chapter or section prompts. We evaluated two such tasks from Longwriter and our benchmark:
> - **Novel-Writing Task (from longwriter)**: Write a {L}-word novel about a teenage heroine who grows up and ends up changing the world.
> - **Architecture-Design Task (revised from VOLTBench)**: Please design a multi-story building. Describe the function and layout of each floor. Ensure the entire description contains at least {L} words, with clarity and continuity throughout the document. Do not stop generating until all floors are described and the document is concluded with 'finished'.
>
> ### 3.1 **Experimental Results**
> We compared our method against Qwen2.5-7B-Instruct (the base model), LongWriter-8B, and GPT-4o-mini on these free form tasks (**L=20000**). The detailed quantitative results are presented in Table 1 and Table 2:
>
> **Table1**. Performance comparison of evaluated models on free form task, conducted in English. Representative results are shown for novel writing task for 20k words from LongWriter [1]. For Length Standard Deviation (LSD), which measures the absolute volatility of output lengths, the values in parentheses provide context by indicating the generated mean length (in words).
> | Model | Length Volatility | | | Generation Quality |
> |-------|------------------|-|-|----------------------------|
> | | LSD (↓) | LVC (↓) | MLA (↑) | UCA (↑) |
> | Qwen2.5-7B | 7300 (24068) | 30.3% | 79.6% | 94.3% (±2.5%) |
> | LongWriter-8B | 355 (502) | 70.7% | 2.5% | 75.3% (±6.2%) |
> | GPT-4o-mini | 97.3 (447) | 21.7% | 2.2% | 82.7% (±4.9%) |
> | Ours | 2346 (19406) | **12.1%** | **97%** | **96.4%** (±2.9%) |
>
> **Results**: LongWriter-8B and GPT-4o-mini failed completely, generating an average of only 502 words and 447 words, respectively. The Qwen2.5-7B base model significantly overshot the target, generating 24,068 words with a high Length Volatility Coefficient (LVC) of 30.3%. Our method generated 19,406 words, extremely close to the 20,000-word target. It achieved a 97.0% Mean Length Adherence (MLA) and the lowest LVC (12.1%), while also maintaining the highest generation quality (UCA 96.4%).
>
> **Table2**. Performance comparison of evaluated models on free form task, conducted in English under simple difficulty set. Representative results are shown for Architecture-Design for 20k words.
> | Model | Length Volatility | | | Generation Quality |
> |-------|------------------|-|-|----------------------------|
> | | LSD (↓) | LVC (↓) | MLA (↑) | UCA (↑) |
> | Qwen2.5-7B | 6253 (15847) | 39.4% | 79.2% | 93.4% (±3.1%) |
> | LongWriter-8B | 1893 (7107) | 26.6% | 35.5% | 79.8% (±5.2%) |
> | GPT-4o-mini | 306 (631) | 48.6% | 3.2% | 84.8% (±4.2%) |
> | Ours | 2450 (21618) | **11.3%** | **91.9%** | **96.8%** (±2.5%) |
>
> **Results**: Baseline models again failed to meet the target, the Qwen2.5-7B base model generated only 15,847 words. Our method again demonstrated the best length control, generating 21,618 words. It achieved both the highest MLA (91.9%) and the lowest LVC (11.3%), while also achieving the highest UCA quality (96.8%).
>
> To provide a comprehensive view of free-form task performance, **we conducted experiments across a wide range of target lengths, including L=1k, 2k, 4k, 10k, 20k, and 40k words**. We have plotted these results as curves in the revised **Appendix G (see Figure 10, Section Performance on free form task)**. These curves clearly show that baseline models generally fail at controllability as the length requirement increases. Our method (Qwen2.5-7B+ours) closely tracks the $y=x$ reference line across the entire range from 1k to 40k words. **Our method also maintains a significantly smaller standard deviation, demonstrating superior and stable control over free-form generation**.

---

> ### Author Response · Authors · 2025-11-20
> **(3/5) Responses by Authors**
>
> > **`Weakness 1`**: The paper’s quality evidence leans on structural or task-specific signals (e.g., section adherence, execution checks) rather than human-perceived writing quality. For creative long-form text, improved stability and length alignment on VOLTBench do not necessarily guarantee unchanged or better writing quality versus strong baselines.
>
> `Response`: (Continued)
> ### 4. **Quantitative Analysis of Local Semantics**
> To further and more objectively address your concern about potential repetition and filler phrases, **we conducted a new, fine-grained quantitative analysis** on the generated text. We analyzed the full generated text across **all N=5 runs for both Qwen2.5-7B (Baseline) and Ours (SELB)** to detect any signs of content degradation.
>
> Implementation: We utilized the complete output text from all 5 generation runs for each model. For each full generation run, we calculated the following objective metrics to detect repetition and filler content:
> - **Rep-3 & Rep-4 ($\downarrow$)**: To directly quantify the increased repetition. We measure the percentage of 3-grams and 4-grams that are repeated anywhere in the document. This captures both local loops (e.g., I hope... I hope...) and global redundancy.
> - **Lexical Diversity (Type-Token Ratio, TTR) ($\uparrow$)**: To quantify filler content. A text heavily padded with generic filler phrases or repetitive patterns will exhibit a significantly lower Type-Token Ratio (Unique Words / Total Words).
> We report the Mean and Standard Deviation of these metrics across the 5 runs.
>
> **Table**. Comparison of lexical diversity metrics across different models. The 3-gram and 4-gram repetition rates measure the proportion of repeated n-gram patterns, while TTR (type–token ratio) quantifies vocabulary richness by examining the balance between unique and total tokens.
> | Method       | 3-gram (↓)         | 4-gram (↓)         | TTR (↑)            |
> |--------------|--------------------|--------------------|--------------------|
> | Base         | 69.32% (±36.16%)   | 68.69% (±36.33%)   | 0.1509 (±0.2077)   |
> | SELB         | **1.47%** (±0.57%)     | **0.39%** (±0.26%)     | **0.5318** (±0.1078)   |
> | SELB-Hybrid  | 3.85% (±1.44%)     | 2.73% (±1.17%)     | 0.4570 (±0.0780)   |
>
> **The base model exhibits severe degeneration, characterized by extremely high repetition rates (approximately 69% for both 3-grams and 4-grams) and critically low vocabulary richness (TTR of ~0.15)**, confirming that repetitive loops are the natural failure mode of the model without intervention. In sharp contrast, **both SELB and SELB-Hybrid dramatically improve generation quality, maintaining repetition rates below 4% and more than tripling the TTR compared to the baseline**. This demonstrates that our active logits intervention effectively prevents the model’s inherent tendency to collapse into repetition, rather than forcing it to generate low-quality padding.

---

> ### Author Response · Authors · 2025-11-20
> **(4/5) Responses by Authors**
>
> > **`Weakness 2`**: SELB addresses sampling volatility but does not clearly separate it from infrastructure nondeterminism. In creativity-oriented settings where temperature cannot be set very low, it remains unclear how much instability comes from stochastic decoding versus platform/kernel nondeterminism (fixed seeds, deterministic math, op ordering)
>
> `Response`: We thank the reviewer for this insightful comment. This is an important and often overlooked technical detail in LLM inference.  While we acknowledge that infrastructure nondeterminism introduces minute logit perturbations, **the volatility we observed in baseline models is macroscopic and catastrophic**:
> - **Collapse Phenomenon**: As shown in our new experiment (Table 1, Novel Writing), baseline models (e.g., LongWriter-8B and GPT-4o-mini) averaged only about 500 words before stopping when tasked with generating 20,000 words. **This magnitude of Length Collapse (from 20k to 0.5k words) far exceeds** what can be explained by the accumulation of low-level floating-point errors.
> - **Attribution**: This massive structural failure is primarily attributed to **Attention Collapse  and distribution shift during long-sequence generation**. When the context window fills, the model loses track of the overall objective, leading to local optima such as premature stopping or repetition. SELB is designed to mitigate this structural instability, rather than merely masking underlying computational noise.
>
> Addressing your concern about creativity-oriented settings, **our SELB (and our generalized SELB-Hybrid) does not take over the entire generation process. It intervenes in the logits only at specific control nodes** (e.g., detecting EOS, stalls, or repetitive loops). For the vast majority of generation steps, the model still follows the random sampling strategy under high temperature.
>
> Our new experiment used a highly creative Novel Writing task (Table 1). The results show that while **SELB forced the model to reach the 20,000-word target, it still achieved a very high UCA score (96.4%)** (which evaluates content richness and coherence). This proves that SELB does not stifle sampling randomness (creativity) but provides a stable structural container for it, preventing it from devolving into structural collapse.
>
> **To minimize nondeterministic variables, we strictly fixed global random seeds in all experiments**. Under these controlled conditions, baseline models still exhibited extremely high Length Volatility Coefficients (LVC) (e.g., 70.7% for LongWriter). In contrast, our method reduced the LVC to 12.1%.
>
>  This huge difference in stability under identical experimental controls further confirms that the primary source of instability is the model's inherent generation defects, not platform or kernel nondeterminism.

---

> ### Author Response · Authors · 2025-11-20
> **(5/5) Responses by Authors**
>
> > **`Weakness 3`**: The method implicitly relies on explicit section templates by boosting next-section title tokens. This structural bias limits applicability to essays or narratives without clear headings, where attention peaks tied to title tokens are weak or absent and structural enforcement can become brittle or intrusive.
>
> `Response`: Thank you for your question. To address your concern, we developed the SELB-Hybrid method and conducted validation experiments specifically for untitled narrative texts.
>
> Addressing the issue of Weak/Absent Attention Peaks, our **SELB-Hybrid strategy no longer relies on finding or reinforcing specific structural tokens** (like Chapter). Instead, it operates through a reactive mechanism:
> - **Structure-Agnostic Monitoring**: We no longer track whether the model attends to the next title. Instead, we monitor the health of the generation flow—specifically, checking for signs of stalling (e.g., falling into repetitive loops  or failing to produce natural breaks like \n for an extended period).
> - **Generic Intervention**: The processor intervenes only when a stall is detected. Crucially, the intervention no longer boosts specific structural titles but boosts common continuation tokens (like \n)
> This shift eliminates structural bias because it does not impose any preset template. It provides a necessary nudge only when the model is about to collapse (Attention Collapse ).
>
> To directly test this new mechanism in the narratives without clear headings scenario you mentioned, **we conducted a 20,000-word Novel Writing experiment. This task involves a continuous narrative flow with no chapter markers**.
>
> **Table**. Performance comparison of evaluated models on free form novel writing task, conducted in English. Representative results are shown for novel writing task for 20k words from LongWriter (Bai et al., 2024). For the LSD metric, the values in parentheses provide context by showing the generated mean length (in words).
> | Model | Length Volatility | | | Generation Quality |
> |-------|------------------|-|-|----------------------------|
> | | LSD (↓) | LVC (↓) | MLA (↑) | UCA (↑) |
> | Qwen2.5-7B | 7300 (24068) | 30.3% | 79.6% | 94.3% (±2.5%) |
> | LongWriter-8B | 355 (502) | 70.7% | 2.5% | 75.3% (±6.2%) |
> | GPT-4o-mini | 97.3 (447) | 21.7% | 2.2% | 82.7% (±4.9%) |
> | Ours | 2346 (19406) | **12.1%** | **97%** | **96.4%** (±2.9%) |
>
> While **baseline models (LongWriter, GPT-4o-mini) suffered catastrophic failure (generating < 500 words)** due to the lack of structural guidance, **our method successfully generated 19,406 words, achieving a Mean Length Adherence (MLA) of 97.0%**. This proves that even without title attention peaks, our keep-alive mechanism can robustly drive long-form generation. Our method achieved a 96.4% UCA (Unstructured Content Accuracy) score, significantly outperforming all baselines. This indicates that our intervention did not disrupt narrative coherence in an intrusive way, but rather effectively maintained a high-quality generation flow.
>
> In conclusion, we acknowledge that the original SELB method's reliance on structural anchors limited its application in untitled texts. However, by introducing SELB-Hybrid, **we successfully generalized the core idea into a dynamic state-maintenance mechanism that does not rely on structural peaks**. New experiments demonstrate that this generalized method exhibits high robustness and non-intrusiveness when processing untitled narratives.

---

> ### Comment · Reviewer_Kq9F · 2025-11-21
>
> Is your method as effective and outstanding in other benchmarks as it is in their evaluations?, such as longbench-writer (longwriter)[1] and WritingBench[2].
>
> [1] LongWriter: Unleashing 10,000+ Word Generation from Long Context LLMs
>
> [2] WritingBench: A Comprehensive Benchmark for Generative Writing

---

> > ### Author Response · Authors · 2025-11-23
> > **Evaluations on LongBench-Writer and WritingBench**
> >
> > `Response`: We sincerely thank the reviewer for suggesting these authoritative benchmarks. To verify the generalization and robustness of our method, **we conducted comprehensive experiments on both LongBench-Writer [1] and WritingBench [2]**.
> >
> > ### 1. **Performance on LongBench-Writer**
> > As shown in **Table 1**, Our method applied to Qwen2.5-7B achieved an **Overall Score ($\bar{S}$) of 85.3, surpassing the  LongWriter-9B-DPO**. In the challenging [4k, 20k) ultra-long segment our method maintains a high Length Score ($S_l$) of 89.9, significantly outperforming the base model and competing proprietary models.
> >
> >
> > **Table 1:** Evaluation results on LongBench-Write.
> > | | **Overall** | **Overall** | **Overall** | **[0, 500)** | **[0, 500)** | **[500, 2k)** | **[500, 2k)** | **[2k, 4k)** | **[2k, 4k)** | **[4k, 20k)** | **[4k, 20k)** |
> > | :--- | :---: | :---: | :---: | :---: | :---: | :---: | :---: | :---: | :---: | :---: | :---: |
> > | **Model** | **$\bar{S}$** | **$S_l$** | **$S_q$** | **$S_l$** | **$S_q$** | **$S_l$** | **$S_q$** | **$S_l$** | **$S_q$** | **$S_l$** | **$S_q$** |
> > | *Proprietary models* | | | | | | | | | | | |
> > | **Claude 3.5 Sonnet** | 80.7 | 73.7 | 87.7 | 87.0 | 92.5 | 93.6 | 90.4 | 81.3 | 86.6 | 26.0 | 80.9 |
> > | **GPT-4 Turbo** | 67.3 | 47.9 | 86.6 | 92.0 | 90.2 | 81.2 | 90.7 | 12.3 | 85.5 | 0 | 78.7 |
> > | **GPT-4o mini** | 77.6 | 64.9 | 90.3 | **92.8** | **95.4** | **91.7** | 93.1 | 61.7 | 88.3 | 5.9 | 84.3 |
> > | **GPT-4o** | 78.6 | 65.3 | **91.8** | 91.0 | 94.6 | 91.4 | **93.6** | 65.5 | **93.0** | 5.6 | 85.3 |
> > | *Open-source models* | | | | | | | | | | | |
> > | **GLM-4-9B-chat** | 68.3 | 51.0 | 85.5 | 72.8 | 89.9 | 86.6 | 88.5 | 37.9 | 84.8 | 0.2 | 78.7 |
> > | **Llama-3.1-8B-Instruct** | 60.3 | 50.0 | 70.6 | 91.0 | 84.0 | 77.9 | 76.6 | 28.1 | 64.5 | 0 | 57.1 |
> > | **Llama-3.1-70B-Instruct** | 65.6 | 50.8 | 80.3 | 88.6 | 82.1 | 85.0 | 83.1 | 18.7 | 80.4 | 3.8 | 74.7 |
> > | **Mistral-Large-Instruct** | 77.0 | 65.6 | 88.3 | 90.1 | 92.6 | 89.2 | 90.4 | 66.5 | 87.5 | 9.3 | 82.4 |
> > | *Specialized models* | | | | | | | | | | | |
> > | **Suri-I-ORPO** | 56.6 | 59.6 | 53.5 | 78.3 | 60.6 | 68.3 | 62.6 | 66.6 | 45.7 | 22.6 | 44.0 |
> > | **LongWriter-8B** | 79.8 | 77.4 | 82.2 | 80.2 | 82.2 | 74.5 | 82.8 | 78.1 | 83.5 | 77.9 | 79.9 |
> > | **LongWriter-9B** | 80.5 | 78.6 | 82.3 | 83.9 | 86.2 | 75.6 | 84.8 | 76.0 | 80.2 | 80.3 | 77.3 |
> > | **LongWriter-9B-DPO** | 84.0 | 82.6 | 85.4 | 82.5 | 88.2 | 81.7 | 86.1 | 76.8 | 85.7 | **90.3** | 81.6 |
> > | **Ours (Qwen2.5-7B)** | **85.3** | **86.7** | 83.9 | 86.9 | 85.8 | 84.1 | 83.9 | **86.4** | 79.3 | 89.9 | **86.8** |

---

> ### Author Response · Authors · 2025-11-23
> **Evaluations on LongBench-Writer and WritingBench**
>
> `Response`: (Continued)
> ### 2. **Performance on WritingBench**
> WritingBench evaluates holistic writing capabilities (Style, Format, Length) across diverse domains. The results in **Table 2**:
> compared to the vanilla Qwen2.5-7B-Instruct (7.43), applying SELB yields a **massive improvement, reaching a score of 8.43. Our 7B model outperforms ChatGPT-4o and is comparable to the much larger Deepseek-R1**. In addition, our high scores in WritingBench's Literature & Art and Style domains confirm that SELB constrains the structure without degrading the literary quality or creativity.
>
> **Table 2:** WritingBench performance of different LLMs across 6 domains and 3 writing requirements.
> | Models | Avg | ZH | EN | D1 | D2 | D3 | D4 | D5 | D6 | R1 | C | R2 | C | R3 | C |
> | :--- | :---: | :---: | :---: | :---: | :---: | :---: | :---: | :---: | :---: | :---: | :---: | :---: | :---: | :---: | :---: |
> | *Proprietary LLMs* | | | | | | | | | | | | | | | |
> | ChatGPT-4o-latest | 8.16 | 8.3 | 8.1 | 8.1 | 8.1 | 8.2 | 8.1 | 8.4 | 8.1 | 8.3 | 8.7 | 8.2 | 8.9 | 8.2 | 8.3 |
> | o1-Preview | 8.15 | 8.1 | 8.2 | 8.0 | 8.1 | 8.2 | 8.2 | 8.4 | 8.1 | 8.2 | 8.6 | 8.2 | 8.8 | 8.2 | 8.2 |
> | Claude-3.5-Sonnet | 7.71 | 7.7 | 7.7 | 7.6 | 7.5 | 7.6 | 7.7 | 7.9 | 8.0 | 7.9 | 8.5 | 7.7 | 8.5 | 7.9 | 8.0 |
> | Gemini-1.5-Pro | 7.78 | 7.8 | 7.7 | 7.7 | 7.5 | 7.8 | 7.9 | 8.0 | 7.9 | 7.9 | 8.6 | 7.9 | 8.8 | 7.9 | 8.0 |
> | Qwen-Max | 8.37 | 8.4 | 8.3 | 8.3 | 8.3 | 8.4 | 8.4 | 8.5 | 8.4 | 8.5 | 8.7 | 8.4 | 9.0 | 8.4 | 8.5 |
> | *Open-source LLMs* | | | | | | | | | | | | | | | |
> | Deepseek-R1 | **8.55** | 8.7 | 8.5 | 8.5 | 8.5 | 8.6 | 8.6 | 8.7 | 8.6 | 8.7 | 8.9 | 8.6 | 9.0 | 8.6 | 8.7 |
> | Deepseek-V3 | 7.95 | 8.0 | 7.9 | 7.9 | 7.8 | 8.0 | 7.8 | 8.2 | 8.0 | 8.1 | 8.6 | 8.0 | 8.9 | 8.0 | 8.2 |
> | Mistral-Large-Instruct | 7.64 | 7.6 | 7.7 | 7.7 | 7.6 | 7.8 | 7.3 | 7.9 | 7.6 | 7.7 | 8.2 | 7.7 | 8.7 | 7.7 | 7.9 |
> | Qwen-2.5-72B-Instruct | 7.90 | 8.0 | 7.9 | 8.0 | 7.8 | 8.1 | 7.7 | 8.2 | 7.8 | 8.0 | 8.3 | 8.0 | 8.8 | 7.9 | 8.0 |
> | Qwen-2.5-7B-Instruct | 7.43 | 7.3 | 7.5 | 7.7 | 7.4 | 7.6 | 6.9 | 7.8 | 7.3 | 7.5 | 7.9 | 7.6 | 8.6 | 7.4 | 7.5 |
> | Llama-3.3-70B-Instruct | 7.01 | 6.7 | 7.3 | 7.0 | 6.9 | 7.0 | 6.8 | 7.3 | 7.3 | 7.1 | 7.8 | 7.1 | 8.2 | 7.0 | 7.2 |
> | Llama-3.1-8B-Instruct | 6.35 | 5.7 | 6.9 | 6.6 | 6.4 | 6.1 | 6.0 | 6.7 | 6.6 | 6.4 | 7.0 | 6.4 | 7.6 | 6.3 | 6.4 |
> | *Capability-enhanced LLMs* | | | | | | | | | | | | | | | |
> | Suri | 4.97 | 4.4 | 5.5 | 5.6 | 5.3 | 5.0 | 4.1 | 5.0 | 5.1 | 4.8 | 5.2 | 5.0 | 5.4 | 4.5 | 4.0 |
> | LongWriter | 7.91 | 7.9 | 7.9 | 8.0 | 8.1 | 8.1 | 7.7 | 8.1 | 7.6 | 7.9 | 8.2 | 8.1 | 8.8 | 7.7 | 7.7 |
> | Qwen-2.5-7B-filtered | **8.49** | 8.6 | 8.4 | 8.4 | 8.4 | 8.6 | 8.4 | 8.6 | 8.5 | 8.6 | 8.8 | 8.5 | 9.0 | 8.5 | 8.6 |
> | Llama-3.1-8B-filtered | **8.49** | 8.6 | 8.4 | 8.5 | 8.4 | 8.6 | 8.4 | 8.6 | 8.5 | 8.6 | 8.8 | 8.5 | 8.9 | 8.5 | 8.5 |
> | Ours (Qwen2.5-7B) | *8.43* | 8.4 | 8.5 | 8.4 | 8.3 | 8.4 | 8.3 | 8.6 | 8.3 | 8.4 | 8.5 | 8.4 | 8.7 | 8.6 | 8.7 |
>
> The additional evaluations on LongBench-Writer and WritingBench suggest that the effectiveness of our method extends beyond the specific scenarios of VOLTBench. The results show our inference-time strategy achieving performance comparable to or exceeding that of specialized fine-tuned models and enhancing base model scores. **These findings support the potential of SELB** to mitigate length-adherence issues and facilitate stable long-form generation while maintaining content quality. **We will include the relevant discussion regarding these additional evaluations on LongBench-Writer and WritingBench in the revised paper.**
>
>
> [1] LongWriter: Unleashing 10,000+ Word Generation from Long Context LLMs
>
> [2] WritingBench: A Comprehensive Benchmark for Generative Writing
>
>
> -----------------------------------------
>
> We hope our responses have addressed your concerns. **If any questions remain, we would be happy to provide further clarification. If you find our explanations and additional experiments helpful, we would sincerely appreciate your reconsideration of the overall rating**. Thank you again for your constructive feedback and support.

---

> ### Author Response · Authors · 2025-11-26
>
> Dear Reviewer Kq9F,
>
> Thank you once again for your thoughtful feedback on our paper. We appreciate the time and effort you have invested in reviewing our work.
>
> As we are nearing the end of the rebuttal period, we wanted to kindly follow up to ensure that you’ve had a chance to review our response to your comments. We hope that our rebuttal has addressed your concerns satisfactorily. If there are any additional points or further clarifications required, we would be happy to provide them promptly.
>
> Additionally, if our response has sufficiently addressed your concerns, we would greatly appreciate if you could consider reflecting this in your evaluation, including revisiting your score if appropriate.
>
> Thank you for your time and understanding. We greatly value your feedback and look forward to hearing from you.
>
> Best regards,
>
> Authors of Paper Submission 8616

---

> > ### Comment · Reviewer_Kq9F · 2025-11-27
> >
> > Thank you for your response — it has addressed most of my concerns. At this point, I’m inclined to keep my current score.

---

> ### Author Response · Authors · 2025-11-28
>
> Dear Reviewer Kq9F,
>
> Thank you once again for your valuable time and thoughtful feedback on our paper. We broadly appreciate the effort you have invested in helping us improve our work.
>
> We are writing to kindly inform you that **we have updated the manuscript to address potential concerns regarding the generalization of our method. Specifically, we have added Section 6.4 and Appendix I** (highlighted in blue for your convenience).
>
> In these new sections, we introduce the SELB-Hybrid strategy and extend our evaluation to free-form generation tasks. We hope these additional experiments satisfactorily address your concerns. If there are any remaining questions or areas where further clarification would be helpful, we would be more than happy to address them promptly.
>
> Thank you again for your time and consideration.
>
> Best regards,
>
> Authors of Paper Submission 8616

---

### Official Review · Reviewer_dTnx · 2025-10-30

**Soundness:** 3
**Presentation:** 3
**Contribution:** 2
**Rating:** 4
**Confidence:** 5

**Summary:**

This paper addresses the problem of "length volatility" in long-form text generation by LLMs, where models produce inconsistent output lengths and content for the same prompt that requires long output. The authors tackle this issue through a three-stage approach:

- Benchmarking: They introduce VOLTBench, a new benchmark designed to quantify length volatility across diverse structured and unstructured tasks, languages, and complexities.
- Probing: They analyze the internal attention mechanisms of models to identify root causes of volatility, defining failure patterns like "Attention Collapse" and "Attention Instability."
- Mitigation: They propose SELB (Structural Enforcement via Logits Boosting), a lightweight, training-free decoding strategy that dynamically modifies logits to enforce structural adherence and prevent premature termination.

Experiments on VOLTBench show that existing models suffer from severe volatility. The proposed SELB method is shown to be effective, increasing the mean output length by 148% and reducing volatility by 69%.

**Strengths:**

- Well-designed benchmark (VOLTBench): The proposed benchmark's inclusion of both unstructured (e.g., story writing) and structured (e.g., code generation) tasks allows for a more robust and objective evaluation than benchmarks that rely solely on subjective creative tasks.
- Strong empirical support: The authors conduct extensive experiments across a wide range of modern LLMs. The inclusion of attention trace visualizations provides compelling qualitative evidence that links internal model behavior to the observed volatility, strengthening the paper's claims.

**Weaknesses:**

- Limited Applicability of SELB: The SELB method appears to be designed for tasks with explicit, predictable structural markers (e.g., "Chapter 1, Chapter 2..."). It is unclear how it would perform on long-form generation tasks that do not have such a clear, chapter-based structure, such as generating a single, long-form essay.
- Potential for Quality Degradation: The SELB method forces a model to continue generating content until a length threshold is met. This could potentially lead to lower-quality, repetitive, or "filler" content within sections, even if the overall structure is maintained.

**Questions:**

- How adaptable is the SELB framework to long-form generation tasks that lack a discrete, repetitive structure like chapters? Could the logic be modified to, for example, enforce length by preventing the generation of end-of-sequence tokens until a target length is met, and how might that affect quality? Perhaps on unstructured long-text benchmarks such as LongBench-Write in the LongWriter paper.
- In your analysis of generation quality, did you investigate the possibility of increased repetition or the use of filler phrases within a single section as a result of forcing the model to meet a minimum length before it could generate the next section title?

---

> ### Author Response · Authors · 2025-11-20
> **(1/3) Responses by Authors**
>
> **Thanks for your careful and insightful reviews.**
> > **`Question 1`**: How adaptable is the SELB framework to long-form generation tasks that lack a discrete, repetitive structure like chapters? Could the logic be modified to, for example, enforce length by preventing the generation of end-of-sequence tokens until a target length is met, and how might that affect quality? Perhaps on unstructured long-text benchmarks such as LongBench-Write in the LongWriter paper.
>
> `Response`: Thank you for your insightful and constructive feedback. **Motivated by your comment, we designed new experiments and demonstrated that our method's core idea, actively intervening in the logits during decoding to enforce constraints, can be generalized to address broader, non-templated, free-form long-text generation tasks**.
>
> ### 1. **The SELB-Hybrid Strategy**
> To adapt our method for free-form tasks that lack explicit sections, we generalized the SELB strategy from section enforcement to **SELB-Hybrid** strategy, which focuses on length enforcement and stall prevention.
> - **Stop Token Suppression**: This is a direct generalization of our original proactive failure prevention logic. In the original method, we suppressed EOS tokens until the model reached the target section count ($P_{total}$). In the new strategy, the processor actively and aggressively suppresses all premature end-of-sequence (EOS) tokens and common conversational stop-phrases (e.g., I hope this helps, Let me know if you need...). This suppression remains active until the model reaches a preset target token count (e.g., 1.5 times the target word count $L$). This directly addresses the problem of models stopping prematurely before reaching the target length.
> - **Hybrid Keep-Alive Mechanism**: Without strong structural anchors such as chapter titles to boost, free-form generation faces a unique stall risk, such as falling into repetitive loops or silent failures (i.e., ceasing to generate new content). To solve this, we introduce a keep-alive mechanism: (1) The processor periodically (e.g., every $\tau_{max} = 500$ tokens) checks the generation state. (2) If the model fails to produce a natural interruption (like a period . or a newline \n) within a grace period (e.g., $k=100$ tokens), the processor determines the model has stalled. (3) It then nudges the model by boosting the logits of generic continuation tokens (e.g., \n and ), forcing it to break the current loop and continue generating new content.
>
> ### 2. **Evaluation on Free-Form Tasks**
> To rigorously test the SELB-Hybrid, we designed highly challenging free-form tasks. We require the model to generate a single, continuous document with a target length of 20,000 words, without providing any chapter or section prompts. We evaluated two such tasks from Longwriter and our benchmark:
> - **Novel-Writing Task (from longwriter)**: Write a {L}-word novel about a teenage heroine who grows up and ends up changing the world.
> - **Architecture-Design Task (revised from VOLTBench)**: Please design a multi-story building. Describe the function and layout of each floor. Ensure the entire description contains at least {L} words, with clarity and continuity throughout the document. Do not stop generating until all floors are described and the document is concluded with 'finished'.

---

> ### Author Response · Authors · 2025-11-20
> **(2/3) Responses by Authors**
>
> > **`Question 1`**: How adaptable is the SELB framework to long-form generation tasks that lack a discrete, repetitive structure like chapters? Could the logic be modified to, for example, enforce length by preventing the generation of end-of-sequence tokens until a target length is met, and how might that affect quality? Perhaps on unstructured long-text benchmarks such as LongBench-Write in the LongWriter paper.
>
> `Response`: (Continued)
> ### 3. **Experimental Results**
> We compared our method against Qwen2.5-7B-Instruct (the base model), LongWriter-8B, and GPT-4o-mini on these free form tasks (**L=20000**). The detailed quantitative results are presented in Table 1 and Table 2:
>
> **Table1**. Performance comparison of evaluated models on free form task, conducted in English. Representative results are shown for novel writing task for 20k words from LongWriter [1]. For Length Standard Deviation (LSD), which measures the absolute volatility of output lengths, the values in parentheses provide context by indicating the generated mean length (in words).
> | Model | Length Volatility | | | Generation Quality |
> |-------|------------------|-|-|----------------------------|
> | | LSD (↓) | LVC (↓) | MLA (↑) | UCA (↑) |
> | Qwen2.5-7B | 7300 (24068) | 30.3% | 79.6% | 94.3% (±2.5%) |
> | LongWriter-8B | 355 (502) | 70.7% | 2.5% | 75.3% (±6.2%) |
> | GPT-4o-mini | 97.3 (447) | 21.7% | 2.2% | 82.7% (±4.9%) |
> | Ours | 2346 (19406) | **12.1%** | **97%** | **96.4%** (±2.9%) |
>
> **Results**: LongWriter-8B and GPT-4o-mini failed completely, generating an average of only 502 words and 447 words, respectively. The Qwen2.5-7B base model significantly overshot the target, generating 24,068 words with a high Length Volatility Coefficient (LVC) of 30.3%. Our method generated 19,406 words, extremely close to the 20,000-word target. It achieved a 97.0% Mean Length Adherence (MLA) and the lowest LVC (12.1%), while also maintaining the highest generation quality (UCA 96.4%).
>
> **Table2**. Performance comparison of evaluated models on free form task, conducted in English under simple difficulty set. Representative results are shown for Architecture-Design for 20k words.
> | Model | Length Volatility | | | Generation Quality |
> |-------|------------------|-|-|----------------------------|
> | | LSD (↓) | LVC (↓) | MLA (↑) | UCA (↑) |
> | Qwen2.5-7B | 6253 (15847) | 39.4% | 79.2% | 93.4% (±3.1%) |
> | LongWriter-8B | 1893 (7107) | 26.6% | 35.5% | 79.8% (±5.2%) |
> | GPT-4o-mini | 306 (631) | 48.6% | 3.2% | 84.8% (±4.2%) |
> | Ours | 2450 (21618) | **11.3%** | **91.9%** | **96.8%** (±2.5%) |
>
> **Results**: Baseline models again failed to meet the target, the Qwen2.5-7B base model generated only 15,847 words. Our method again demonstrated the best length control, generating 21,618 words. It achieved both the highest MLA (91.9%) and the lowest LVC (11.3%), while also achieving the highest UCA quality (96.8%).
>
> To provide a comprehensive view of free-form task performance, **we conducted experiments across a wide range of target lengths, including L=1k, 2k, 4k, 10k, 20k, and 40k words**. We have plotted these results as curves in the revised **Appendix G (see Figure 10, Section Performance on free form task)**. These curves clearly show that baseline models generally fail at controllability as the length requirement increases. Our method (Qwen2.5-7B+ours) closely tracks the $y=x$ reference line across the entire range from 1k to 40k words. **Our method also maintains a significantly smaller standard deviation, demonstrating superior and stable control over free-form generation**.
>
> In summary, current advanced models are uncontrollable and unreliable for free-form long-text generation tasks that lack structural guidance. They widely suffer from length collapse and high volatility. **The core idea of our SELB method (decoding-stage logits intervention) is highly generalizable**. By shifting the intervention logic from section enforcement to length enforcement and stall prevention, our method is the only strategy among those tested that can effectively enforce length constraints, drastically reduce volatility, and consistently maintain the highest generation quality on extreme-scale free-form tasks. We sincerely thank you for this feedback.
>
> [1] Bai Y, Zhang J, Lv X, et al. Longwriter: Unleashing 10,000+ word generation from long context llms.

---

> ### Author Response · Authors · 2025-11-20
> **(3/3) Responses by Authors**
>
> > **`Question 2`**: In your analysis of generation quality, did you investigate the possibility of increased repetition or the use of filler phrases within a single section as a result of forcing the model to meet a minimum length before it could generate the next section title?
>
> `Response`: We thank you for this very professional and insightful question. Our investigation into the baseline's failure modes (e.g., Table 22 in the paper) reveals that **repetition is the default failure mode of the unconstrained model**.
>
> When baselines (like Qwen2.5-7B) generate long contexts without guidance, they suffer from Attention Collapse, getting stuck in high-probability loops (e.g., repeating "I hope this helps" or structural hallucinations). SELB acts as an "Attention Refresher." By intervening in the logits, it forces the model to break out of these local loops and attend to the next structural goal, thereby **incentivizing the generation of novel content rather than repetitive padding**.
>
> ### **Quantitative Analysis of Local Semantics**
> To address your concern about potential repetition and filler phrases, **we conducted a new, fine-grained quantitative analysis** on the generated text. We analyzed the full generated text across **all N=5 runs for both Qwen2.5-7B (Baseline) and Ours (SELB)** to detect any signs of content degradation.
>
> Implementation: We utilized the complete output text from all 5 generation runs for each model. For each full generation run, we calculated the following objective metrics to detect repetition and filler content:
> - **Rep-3 & Rep-4 ($\downarrow$)**: To directly quantify the increased repetition. We measure the percentage of 3-grams and 4-grams that are repeated anywhere in the document. This captures both local loops (e.g., I hope... I hope...) and global redundancy.
> - **Lexical Diversity (Type-Token Ratio, TTR) ($\uparrow$)**: To quantify filler content. A text heavily padded with generic filler phrases or repetitive patterns will exhibit a significantly lower Type-Token Ratio (Unique Words / Total Words).
> We report the Mean and Standard Deviation of these metrics across the 5 runs.
>
> Table 3. Comparison of lexical diversity metrics across different models. The 3-gram and 4-gram repetition rates measure the proportion of repeated n-gram patterns, while TTR (type–token ratio) quantifies vocabulary richness by examining the balance between unique and total tokens.
> | Method       | 3-gram (↓)         | 4-gram (↓)         | TTR (↑)            |
> |--------------|--------------------|--------------------|--------------------|
> | Base         | 69.32% (±36.16%)   | 68.69% (±36.33%)   | 0.1509 (±0.2077)   |
> | SELB         | **1.47%** (±0.57%)     | **0.39%** (±0.26%)     | **0.5318** (±0.1078)   |
> | SELB-Hybrid  | 3.85% (±1.44%)     | 2.73% (±1.17%)     | 0.4570 (±0.0780)   |
>
> **The base model exhibits severe degeneration, characterized by extremely high repetition rates (approximately 69% for both 3-grams and 4-grams) and critically low vocabulary richness (TTR of ~0.15)**, confirming that repetitive loops are the natural failure mode of the model without intervention. In sharp contrast, **both SELB and SELB-Hybrid dramatically improve generation quality, maintaining repetition rates below 4% and more than tripling the TTR compared to the baseline**. This demonstrates that our active logits intervention effectively prevents the model’s inherent tendency to collapse into repetition, rather than forcing it to generate low-quality padding.

---

> ### Author Response · Authors · 2025-11-26
>
> Dear Reviewer dTnx,
>
> Thank you once again for your thoughtful feedback on our paper. We appreciate the time and effort you have invested in reviewing our work.
>
> As we are nearing the end of the rebuttal period, we wanted to kindly follow up to ensure that you’ve had a chance to review our response to your comments. We hope that our rebuttal has addressed your concerns satisfactorily. If there are any additional points or further clarifications required, we would be happy to provide them promptly.
>
> Additionally, if our response has sufficiently addressed your concerns, we would greatly appreciate if you could consider reflecting this in your evaluation, including revisiting your score if appropriate.
>
> Thank you for your time and understanding. We greatly value your feedback and look forward to hearing from you.
>
> Best regards,
>
> Authors of Paper Submission 8616

---

> ### Author Response · Authors · 2025-11-28
>
> Dear Reviewer dTnx,
>
> Thank you once again for your valuable time and thoughtful feedback on our paper. We broadly appreciate the effort you have invested in helping us improve our work.
>
> We are writing to kindly inform you that **we have updated the manuscript to address potential concerns regarding the generalization of our method. Specifically, we have added Section 6.4 and Appendix I** (highlighted in blue for your convenience).
>
> In these new sections, we introduce the SELB-Hybrid strategy and extend our evaluation to free-form generation tasks. We hope these additional experiments satisfactorily address your concerns. If there are any remaining questions or areas where further clarification would be helpful, we would be more than happy to address them promptly.
>
> Thank you again for your time and consideration.
>
> Best regards,
>
> Authors of Paper Submission 8616

---

### Official Review · Reviewer_4xJJ · 2025-10-30

**Soundness:** 3
**Presentation:** 4
**Contribution:** 4
**Rating:** 6
**Confidence:** 4

**Summary:**

This paper addresses the critical issue of output volatility in Large Language Models (LLMs) —the inconsistency in length and content across multiple runs for the same prompt. It conducts a three-stage investigation: "Benchmarking, Probing, and Mitigation".

**Strengths:**

- Value of Problem Definition: The paper successfully identifies and defines a problem that is critical in practice but often overlooked in academic research: output volatility. This stability across runs is arguably more important for cost-control and system reliability than the quality of a single generation.
- Systematic Benchmark: VOLTBench is a solid contribution. It is the first to center its evaluation on "volatility" and "multiple sampling". Its multi-dimensional design (structured/unstructured, multi-lingual, multi-complexity ) makes it more comprehensive and rigorous than existing long-text benchmarks.
- Intuitive Probing: While the analysis of "Attention Collapse/Instability" may not be a strict causal proof , it provides an intuitive and reproducible internal mechanism explanation for the two most common failure modes (premature termination and section skipping ). This goes beyond mere phenomenological observation.

**Weaknesses:**

- Questionable Generalizability: SELB is a "brute-force patch" based on hard-coded rules, rather than an improvement to the model's intrinsic long-range planning capabilities. It relies heavily on two parameters that must be manually preset for each specific task: $V_{title}$ (chapter title tokens) and $\tau_{max}$ (minimum length). This severely limits its generalizability:
  - Task Limitation: If a long-text task lacks explicit section markers (e.g., "Write a 10,000-word essay on the societal impact of AI"), SELB would be completely ineffective.
  - Rule Fragility: The $V_{banned}$ (banned vocabulary) is also fragile. In some contexts, a banned phrase (e.g., "I hope...") might be a perfectly reasonable and necessary conclusion.
- Benchmark-Method "Collusion": SELB's remarkable performance stems largely from VOLTBench being a testbed "tailor-made" for it. All prompts in Appendix contain explicit, predictable structural markers like #*# Title:, #*# Floor 1:, etc. This turns SELB's "structural enforcement" into a simple "find-and-replace" task. Its performance on non-cooperative, in-the-wild prompts is unknown.

**Questions:**

- Local Coherence vs. Global Structure: What is the true impact of the $\tau_{max}$ mechanism on local-level semantics? Could a "softer" enforcement strategy be designed, e.g., only allowing the chapter-boost to trigger after the model has generated a natural paragraph break?
- The Root Cause of Failure: If attention collapse is just a symptom, what is the root cause? Is it the model losing its representation of the task instructions over a long context? Could representation analysis tools (like CKA) be used to track the fidelity of key constraint representations (e.g., "number of chapters," "total length") during the generation process?

---

> ### Author Response · Authors · 2025-11-20
> **(1/6) Responses by Authors**
>
> **Thanks for your careful and insightful reviews.**
> > **`Weakness 1`**: Questionable Generalizability: SELB is a brute-force patch based on hard-coded rules, rather than an improvement to the model's intrinsic long-range planning capabilities.
>
> `Response`: Thank you for the detailed feedback. Our work is the first to systematically measure Length Volatility. To measure this volatility accurately and reliably, for example, section skipping, content truncation, and format deviation, a structured, section-based setup (like VOLTBench) was necessary. This setup provided a controlled environment to precisely evaluate a model's structural stability during long-form generation (e.g., using the FAD metric). Our SELB method effectively corrects unstable generation behaviors by actively intervening in the logits during the decoding stage, and **this core approach of logits intervention is generalizable to tasks beyond those in VOLTBench**.
>
>  **To address your specific concerns, we have conducted a series of new experiments and further analysis.**
>
> ### 1. **Regarding Task Limitation**
> Following your suggestion, we designed a new set of highly challenging free-form tasks. These tasks require the model to generate a single, continuous document to meet a specific global word count (e.g., L=20,000 words) without any predefined sections or chapter prompts.
>
> Specifically, we evaluated two main tasks based on prompts from other benchmark and our VOLTBench:
> - **Novel-Writing Task (from the LongWriter benchmark)**: Write a {L}-word novel about a teenage heroine who grows up and ends up changing the world.
> - **Architecture-Design Task (revised from VOLTBench)**: Please design a multi-story building. Describe the function and layout of each floor. Ensure the entire description contains at least {L} words, with clarity and continuity throughout the document. Do not stop generating until all floors are described and the document is concluded with 'finished'.
>
> ### 2. **Regarding Brute-force patch**
> To address your concerns and demonstrate the flexibility of our core idea, **we kept the core logits intervention framework of SELB but replaced these hard-coded rules with more general, dynamic logic, resulting in the SELB-Hybrid** strategy.
> - Generalizing $V_{title}$ (from hard-coded to dynamic keep-alive): We removed the dependency on $V_{title}$. Instead, we use a more intelligent Hybrid Keep-Alive mechanism. This mechanism no longer depends on any task-specific tokens. It dynamically monitors the model's stall state (e.g., falling into a repetitive loop). Only when a stall is detected does it intervene by nudging the model, boosting the logits of common continuation tokens (like \n ).
> - Generalizing $\tau_{max}$ (from section length to global length): We removed the $\tau_{max}$ section parameter. Instead, we use a Global Length Enforcement strategy, implemented via our Stop Token Suppression component, which remains active until the global target token count $L$ is reached.
>
> ### 3. **Regarding Rule Fragility**
> We acknowledge this is a conscious engineering trade-off. Our SELB does continue to use common stop-phrase suppression. **We retained this rule based on strong empirical analysis**. As our original paper's Failure Case Study  shows, these specific phrases (like I hope... or I hope these sections are helpful.) are not reasonable and necessary conclusions. They are high-frequency failure signals of Attention Collapse , where the model reverts from a content generator to a chatbot persona. Our new 20,000-word free-form experiments (detailed next) confirm this. Actively suppressing these phrases was a necessary component to prevent the model from prematurely halting and to successfully achieve the extreme lengths shown in Tables 1 & 2. **The fact that our method achieved the highest UCA (generation quality) scores (96.4% and 96.8%) in these demanding tasks demonstrates that this rule is not fragile and does not harm the quality** of the final, necessary conclusion.

---

> ### Author Response · Authors · 2025-11-20
> **(2/6) Responses by Authors**
>
> > **`Weakness 1`**: Questionable Generalizability: SELB is a brute-force patch based on hard-coded rules, rather than an improvement to the model's intrinsic long-range planning capabilities.
>
> `Response`: (Continued)
> ### 4. **Experimental Results**
> We compared our method against Qwen2.5-7B-Instruct (the base model), LongWriter-8B, and GPT-4o-mini on these free form tasks (**L=20000**). The detailed quantitative results are presented in Table 1 and Table 2:
>
> **Table1**. Performance comparison of evaluated models on free form task, conducted in English. Representative results are shown for novel writing task for 20k words from LongWriter [1]. For Length Standard Deviation (LSD), which measures the absolute volatility of output lengths, the values in parentheses provide context by indicating the generated mean length (in words).
> | Model | Length Volatility | | | Generation Quality |
> |-------|------------------|-|-|----------------------------|
> | | LSD (↓) | LVC (↓) | MLA (↑) | UCA (↑) |
> | Qwen2.5-7B | 7300 (24068) | 30.3% | 79.6% | 94.3% (±2.5%) |
> | LongWriter-8B | 355 (502) | 70.7% | 2.5% | 75.3% (±6.2%) |
> | GPT-4o-mini | 97.3 (447) | 21.7% | 2.2% | 82.7% (±4.9%) |
> | Ours | 2346 (19406) | **12.1%** | **97%** | **96.4%** (±2.9%) |
>
> **Results**: LongWriter-8B and GPT-4o-mini failed completely, generating an average of only 502 words and 447 words, respectively. The Qwen2.5-7B base model significantly overshot the target, generating 24,068 words with a high Length Volatility Coefficient (LVC) of 30.3%. Our method generated 19,406 words, extremely close to the 20,000-word target. It achieved a 97.0% Mean Length Adherence (MLA) and the lowest LVC (12.1%), while also maintaining the highest generation quality (UCA 96.4%).
>
> **Table2**. Performance comparison of evaluated models on free form task, conducted in English under simple difficulty set. Representative results are shown for Architecture-Design for 20k words.
> | Model | Length Volatility | | | Generation Quality |
> |-------|------------------|-|-|----------------------------|
> | | LSD (↓) | LVC (↓) | MLA (↑) | UCA (↑) |
> | Qwen2.5-7B | 6253 (15847) | 39.4% | 79.2% | 93.4% (±3.1%) |
> | LongWriter-8B | 1893 (7107) | 26.6% | 35.5% | 79.8% (±5.2%) |
> | GPT-4o-mini | 306 (631) | 48.6% | 3.2% | 84.8% (±4.2%) |
> | Ours | 2450 (21618) | **11.3%** | **91.9%** | **96.8%** (±2.5%) |
>
> **Results**: Baseline models again failed to meet the target, the Qwen2.5-7B base model generated only 15,847 words. Our method again demonstrated the best length control, generating 21,618 words. It achieved both the highest MLA (91.9%) and the lowest LVC (11.3%), while also achieving the highest UCA quality (96.8%).
>
> To provide a comprehensive view of free-form task performance, **we conducted experiments across a wide range of target lengths, including L=1k, 2k, 4k, 10k, 20k, and 40k words**. We have plotted these results as curves in the revised **Appendix G (see Figure 10, Section Performance on free form task)**. These curves clearly show that baseline models generally fail at controllability as the length requirement increases. Our method (Qwen2.5-7B+ours) closely tracks the $y=x$ reference line across the entire range from 1k to 40k words. **Our method also maintains a significantly smaller standard deviation, demonstrating superior and stable control over free-form generation**.
>
> In summary, current advanced models are uncontrollable and unreliable for free-form long-text generation tasks that lack structural guidance. They widely suffer from length collapse and high volatility. **The core idea of our SELB method (decoding-stage logits intervention) is highly generalizable**. By shifting the intervention logic from section enforcement to length enforcement and stall prevention, our method is the only strategy among those tested that can effectively enforce length constraints, drastically reduce volatility, and consistently maintain the highest generation quality on extreme-scale free-form tasks. We sincerely thank you for this feedback.
>
> [1] Bai Y, Zhang J, Lv X, et al. Longwriter: Unleashing 10,000+ word generation from long context llms.

---

> ### Author Response · Authors · 2025-11-20
> **(3/6) Responses by Authors**
>
> > **`Weakness 2`**: Benchmark-Method "Collusion": SELB's remarkable performance stems largely from VOLTBench being a testbed "tailor-made" for it. All prompts in Appendix contain explicit, predictable structural markers.
>
> `Response`: Thank you for your comment. This feedback on Benchmark-Method 'Collusion' raises an important question about our method's generalizability.
>
> ### 1. **Benchmark Difficulty and Validity**
> We want to clarify that **all prompts in VOLTBench (including those in the Appendix with markers like ## Title:) were applied consistently to all evaluated baseline models**, not just our own. Our experimental results show that despite these predictable structural markers, **all SOTA baseline models still failed severely on these tasks**. They terminated prematurely, skipped sections, or fell into repetition.
>
> This strongly demonstrates that:
> - The long-range structural stability task tested by **VOLTBench is genuinely difficult**;
> - The explicit **markers did not, reduce the task to a simple 'find-and-replace' task**. If they had, the baselines would have easily succeeded;
> - **This widespread failure proves that the active intervention strategy of our SELB is necessary** to ensure models follow these structures, as they do not do so on their own.
>
> ### 2. **Generalization to Free-Form Generation Tasks**
> The fundamental principle of SELB is that real-time logit intervention can effectively guide LLMs to prevent known failure modes (such as truncation, repetition, and section skipping). Crucially, this mechanism of **enforcing constraints via logits is structure-agnostic and highly generalizable**.
>
> In our paper, we first applied this core idea to structured (section-based) tasks. The specific implementation we used, such as boosting $V_{title}$ 2and checking $\tau_{max}$, was simply one way to adapt this idea for the very common section-based long-text generation tasks (like reports, stories, or codebases). To prove the generalizability of our core idea (logits intervention) and to directly address the concern about non-cooperative, in-the-wild prompts, **we conducted new, highly challenging free-form task experiments** (as detailed in our former response). These tasks have no ## Title: or ## Floor 1: markers.
>
> **Following your suggestion, through simple modifications, we generalized SELB's core idea into the SELB-Hybrid** strategy: (1) We removed the dependency on $V_{title}$ 4, using a dynamic keep-alive mechanism that boosts common tokens (like \n) only when a stall is detected. (2) We removed $\tau_{max}$ 5, using a global length enforcement 6 strategy instead.
>
> The results (Table 1 and 2) show that on a **20,000-word free-form task** where baseline models (GPT-4o-mini, LongWriter) completely collapsed (<500 words), our **SELB-Hybrid was the only method to successfully complete the task, achieving >90% Mean Length Adherence (MLA)**. This proves that SELB's core idea (logits intervention) is not find-and-replace.

---

> ### Author Response · Authors · 2025-11-20
> **(4/6) Responses by Authors**
>
> > **`Weakness 2`**: Benchmark-Method "Collusion": SELB's remarkable performance stems largely from VOLTBench being a testbed "tailor-made" for it. All prompts in Appendix contain explicit, predictable structural markers.
>
> `Response`: (Continued)
>
> ### 3. **Robustness Verification on Structured Tasks**
> Finally, to further prove that our generalized method is robust, **we re-evaluated our original, structured VOLTBench tasks using this  SELB-Hybrid method, which does not rely on $V_{title}$ markers**.
>
> **Table 3**. Performance comparison of evaluated models on a 100-section generation task, conducted in English under simple difficulty settings. Representative results are shown for an unstructured task (Story) and a structured task (Code Function).
> | Model                | LSD (↓)            | LVC (↓)  | MLA (↑) | FAD (↓)           | SCA (↑)           | UCA (↑)           |
> |----------------------|--------------------|----------|---------|-------------------|-------------------|-------------------|
> | GPT-4o mini          | 325.65 (959)       | 33.9%    | 4.8%    | 1.41 (7.00)       | 84.6% (±30.8%)    | 86.7% (±6.7%)     |
> | Deepseek-R1          | 103.30 (1198)      | 8.6%     | 6.0%    | 1.25 (4.33)       | 35.0% (±13.2%)    | 93.3% (±3.7%)     |
> | Qwen2.5-1.5B         | 27.78 (142)        | 19.6%    | 0.7%    | 0.47 (1.67)       | 15.6% (±24.0%)    | 84.0% (±7.1%)     |
> | Llama3.1-8B          | 92.77 (350)        | 26.5%    | 1.7%    | 0.94 (4.33)       | 92.4% (±14.2%)    | 82.0% (±18.9%)    |
> | LongWriter-8B        | 2866.3 (6320)      | 45.4%    | 31.6%   | 21.42 (45.00)     | 32.6% (±31.9%)    | 66.7% (±16.5%)    |
> | Qwen2.5-7B           | 75.87 (445)        | 17.0%    | 2.2%    | 2.05 (10.33)      | 99.8% (±0.4%)     | 86.7% (±7.6%)     |
> | Repetition_penalties   | 553 (2967)         | 18.6%    | 14.8%   | 5.4 (22)          | 98% (±1%)         | 76.7% (±14.5%)    |
> | Stop-entropy   | 713 (2701)         | 26.4%    | 13.5%   | 7.24 (24)         | 95% (±2.5%)       | 83.9% (±8%)       |
> | Length_controllers | 1280 (4470)        | 28.65%   | 77.7%   | 9.2 (28)          | 96% (±2%)         | 85% (±9%)         |
> | Look-ahead   | 268 (2883)         | 9.3%     | 14.4%   | 7.2 (25)          | 94% (±3.5%)       | 84.4% (±8%)       |
> | **SELB-Hybrid**  | 3420 (11474) | 29.8%  | 57.4% | 5.8(20) |98% (±1.5%) |88.4% (±6%) |
> | **SELB**  | 2194.23 (15651) | 14.02% | 78.25% | 7.24 (88.00) | 100% (±0%) | 86.7% (±16.5%) |
>
> The results in Table 3 further validate the generalizability of our framework. **SELB-Hybrid achieves a Mean Length of 11,474 words, nearly doubling the output of the strongest baseline, LongWriter-8B (6,320 words), while maintaining a significantly lower Length Volatility Coefficient (LVC) (29.8% vs. 45.4%)**. Crucially, the method maintains an exceptional Structured Content Accuracy (SCA) of 98%, starkly outperforming LongWriter's 32.6%. This confirms that the model effectively sustains long-range generation through our dynamic keep-alive mechanism, an intrinsic improvement to the decoding process, rather than relying on hard-coded structural patches.

---

> ### Author Response · Authors · 2025-11-20
> **(5/6) Responses by Authors**
>
> > **`Question 1`**: Local Coherence vs. Global Structure: What is the true impact of the mechanism on local-level semantics? Could a softer enforcement strategy be designed, e.g., only allowing the chapter-boost to trigger after the model has generated a natural paragraph break?
>
> `Response`: Thank you for this insightful comment.
>
> ### 1. **Adoption of Context-Aware Enforcement**
> **We fully agree with your suggestion. we have integrated this precise logic into our improved SELB-Hybrid** strategy:
> - Instead of blindly forcing a transition when a length threshold is met, our generalized mechanism **adopts a Context-Aware approach**. As described in our method adaptation (see Hybrid Keep-Alive Mechanism in the response above), the processor primarily monitors the generation for natural interruptions (such as periods . or newlines \n).
> - We do not intervene as long as the model maintains a healthy flow of natural breaks. **The active boosting or nudging is triggered only if the model fails to produce a natural break** within a specified grace window (indicating a stall). This ensures that our intervention respects the model's local semantic boundaries, acting as a safety net rather than a disruption.
>
> The effectiveness of our **SELB-Hybrid is strongly validated by the results on the 20,000-word free-form tasks**.
> - **Length Control**: Despite being soft (waiting for natural breaks), the mechanism proved highly effective, achieving exceptional Mean Length Adherence (MLA) of **97.0% (Novel Task, Table 1) and 91.9% (Architecture Task, Table 2)**. This proves that respecting local boundaries does not compromise global length enforcement.
> - **Content Quality**: Crucially, the high Unstructured Content Accuracy **(UCA) scores (96.4% and 96.8%)** confirm that this non-intrusive intervention successfully preserved the natural narrative flow and coherence, avoiding the disjointed text that rigid forcing might produce.
>
> ### 2. **Quantitative Analysis of Local Semantics**
> To further and more objectively address your concern about potential repetition and filler phrases, **we conducted a new, fine-grained quantitative analysis** on the generated text. We analyzed the full generated text across **all N=5 runs for both Qwen2.5-7B (Baseline) and Ours (SELB)** to detect any signs of content degradation.
>
> Implementation: We utilized the complete output text from all 5 generation runs for each model. For each full generation run, we calculated the following objective metrics to detect repetition and filler content:
> - **Rep-3 & Rep-4 ($\downarrow$)**: To directly quantify the increased repetition. We measure the percentage of 3-grams and 4-grams that are repeated anywhere in the document. This captures both local loops (e.g., I hope... I hope...) and global redundancy.
> - **Lexical Diversity (Type-Token Ratio, TTR) ($\uparrow$)**: To quantify filler content. A text heavily padded with generic filler phrases or repetitive patterns will exhibit a significantly lower Type-Token Ratio (Unique Words / Total Words).
> We report the Mean and Standard Deviation of these metrics across the 5 runs.
>
> Table 3. Comparison of lexical diversity metrics across different models. The 3-gram and 4-gram repetition rates measure the proportion of repeated n-gram patterns, while TTR (type–token ratio) quantifies vocabulary richness by examining the balance between unique and total tokens.
> | Method       | 3-gram (↓)         | 4-gram (↓)         | TTR (↑)            |
> |--------------|--------------------|--------------------|--------------------|
> | Base         | 69.32% (±36.16%)   | 68.69% (±36.33%)   | 0.1509 (±0.2077)   |
> | SELB         | **1.47%** (±0.57%)     | **0.39%** (±0.26%)     | **0.5318** (±0.1078)   |
> | SELB-Hybrid  | 3.85% (±1.44%)     | 2.73% (±1.17%)     | 0.4570 (±0.0780)   |
>
> **The base model exhibits severe degeneration, characterized by extremely high repetition rates (approximately 69% for both 3-grams and 4-grams) and critically low vocabulary richness (TTR of ~0.15)**, confirming that repetitive loops are the natural failure mode of the model without intervention. In sharp contrast, **both SELB and SELB-Hybrid dramatically improve generation quality, maintaining repetition rates below 4% and more than tripling the TTR compared to the baseline**. This demonstrates that our active logits intervention effectively prevents the model’s inherent tendency to collapse into repetition, rather than forcing it to generate low-quality padding.

---

> ### Author Response · Authors · 2025-11-20
> **(6/6) Responses by Authors**
>
> > **`Question 2`**: The Root Cause of Failure: If attention collapse is just a symptom, what is the root cause? Is it the model losing its representation of the task instructions over a long context? Could representation analysis tools (like CKA) be used to track the fidelity of key constraint representations (e.g., number of chapters, total length) during the generation process?
>
> `Response`: Thank you for your insightful suggestion. We further utilized a Representation Analysis Tool to conduct stability analysis, and the results are presented below. Specifically, we employed **Representation Drift Analysis via Cosine Similarity** to track the fidelity of the model's internal representation during the multi-section generation task. **Our analysis confirms that the model cannot maintain a high-fidelity representation of the task instruction and context during long-sequence generation**.
>
> We compared the similarity of the model’s state at any given token t against the anchoring state (t=100), which immediately followed the initial prompt ingestion. **The results, detailed in Table H.1 (and visually in Figure 10 of the main paper/Appendix H)**, provide direct quantitative evidence for this representational collapse in the baseline model.
>
> | Token Step ($t$) | Ours (SELB) Avg Sim | Base Model Avg Sim | Difference       |
> | -------------------- | --------------------- | -------------------- | ------------------ |
> | 100 (Anchor)       | 1.0000              | 1.0000             | 0.0000           |
> | 2000               | 0.8119              | 0.7377             | **0.0742** |
> | 4000               | 0.6821              | 0.5122             | **0.1699** |
> | **5000**     | **0.6932**    | **0.3026**   | **0.3906** |
> | 10000              | 0.6838              | 0.3402             | **0.3436** |
>
> The data reveals two critical insights:
> - **Baseline Collapse**: The Base Model's similarity plummets to ~0.30 at $t=5000$. In high-dimensional space, this indicates the state has become nearly orthogonal to the original task instruction. The model has effectively "lost" the prompt, explaining why it hallucinates or stops.
> - **SELB as a Stabilizer**: Our method maintains a high fidelity ($\sim0.68$) throughout the generation. Crucially, the similarity plateaus rather than decaying.
>
> Our analysis shows that SELB works by creating a virtuous feedback loop: (1) **Correction**: SELB intervenes at the logits level to force the generation of a correct structural token (e.g., "Chapter 5"). (2) **Re-anchoring**: This correct token is fed back as input for the next step. (3) **Realignment**: Processing this explicit structural marker "refreshes" the model's internal hidden states, realigning them with the original task intent and preventing the drift observed in the baseline.
>
> In conclusion, our analysis confirms that Representational Drift is the root cause of long-form generation failure. **Our SELB strategy is effective precisely because it counters this drift by enforcing task structure at the representation level**.
>
> We are very grateful for your suggestion and have included the complete Representational Stability Analysis (Table H.1 and Figure 10) in the revised paper’s Appendix. We believe this analysis is vital for deeply understanding the fundamental cause of long-form generation failure and its mitigation mechanisms.

---

> ### Author Response · Authors · 2025-11-26
>
> Dear Reviewer 4xJJ,
>
> Thank you once again for your thoughtful feedback on our paper. We appreciate the time and effort you have invested in reviewing our work.
>
> As we are nearing the end of the rebuttal period, we wanted to kindly follow up to ensure that you’ve had a chance to review our response to your comments. We hope that our rebuttal has addressed your concerns satisfactorily. If there are any additional points or further clarifications required, we would be happy to provide them promptly.
>
> Additionally, if our response has sufficiently addressed your concerns, we would greatly appreciate if you could consider reflecting this in your evaluation, including revisiting your score if appropriate.
>
> Thank you for your time and understanding. We greatly value your feedback and look forward to hearing from you.
>
> Best regards,
>
> Authors of Paper Submission 8616

---

> ### Author Response · Authors · 2025-11-28
>
> Dear Reviewer 4xJJ,
>
> Thank you once again for your valuable time and thoughtful feedback on our paper. We broadly appreciate the effort you have invested in helping us improve our work.
>
> We are writing to kindly inform you that **we have updated the manuscript to address potential concerns regarding the generalization of our method. Specifically, we have added Section 6.4 and Appendix I** (highlighted in blue for your convenience).
>
> In these new sections, we introduce the SELB-Hybrid strategy and extend our evaluation to free-form generation tasks. We hope these additional experiments satisfactorily address your concerns. If there are any remaining questions or areas where further clarification would be helpful, we would be more than happy to address them promptly.
>
> Thank you again for your time and consideration.
>
> Best regards,
>
> Authors of Paper Submission 8616

---

### Official Review · Reviewer_fkNi · 2025-10-31

**Soundness:** 3
**Presentation:** 3
**Contribution:** 3
**Rating:** 6
**Confidence:** 4

**Summary:**

The paper studies output volatility in long-form generation—i.e., the inconsistency of both length and structure across repeated runs on the same prompt. It contributes (1) VOLTBench, a heterogeneous benchmark that explicitly measures multi-sample stability across unstructured (story/diary/dialogue) and structured (code/JSON/LaTeX) tasks in EN/ZH; (2) attention-trace analysis diagnosing two failure signatures—Attention Collapse and Attention Instability; and (3) SELB, a lightweight decoding-time logit modification scheme combining structural boosting (enforcing section boundaries) and proactive failure suppression (banning filler/EOS prematurely). Reported results show large improvements in stability and length adherence (e.g., LVC down from 45.4% to ~14%, MLA up to ~78%), while maintaining quality (SCA ~100% on structured tasks; UCA ~87% on unstructured) across 5-run evaluations.

**Strengths:**

1. Clear problem framing & novelty: Prior long-form work mostly evaluates single generations; the multi-sample stability focus and metrics (LSD/LVC/MLA; FAD/SCA/UCA) are useful and likely to influence evaluation practice.

2. Neat benchmark design: Heterogeneous tasks, dual languages, scalable chapterized targets (up to very long outputs), and automatic checks via fine-grained constraints and execution-based verification.

3. Evident empirical gains: On a 100-section setting, volatility is reduced (~69% LVC reduction vs LongWriter-8B), lengths better match targets (MLA ~78%), and quality is strong (SCA 100%, UCA ~86.7%).

**Weaknesses:**

1. Scope & generality of SELB: The structural prior assumes explicit, chapter-style sectioning with recognizable “title” tokens; it’s unclear how well this transfers to non-templated long-form tasks (e.g., essays, reports without section tokens) or to multilingual tokenization quirks. Ablations on more free-form schemas are needed.

2. Fairness of baselines: Stronger inference-time baselines (e.g., length controllers, look-ahead/ITR, stop-entropy controls, repetition penalties tuned to parity) could be included to ensure SELB’s gains aren’t mostly from adding any simple structure prior.

**Questions:**

N/A

---

> ### Author Response · Authors · 2025-11-20
> **(1/3) Responses by Authors**
>
> **Thanks for your careful and insightful reviews.**
> > **`Weakness 1`**: Scope & generality of SELB: The structural prior assumes explicit, chapter-style sectioning with recognizable title tokens; it’s unclear how well this transfers to non-templated long-form tasks (e.g., essays, reports without section tokens) or to multilingual tokenization quirks. Ablations on more free-form schemas are needed.
>
> `Response`: Thank you for your insightful and constructive feedback. Motivated by your comment, we designed new experiments and demonstrated that **our method's core idea, actively intervening in the logits during decoding to enforce constraints, can be generalized to address broader, non-templated, free-form long-text generation tasks**.
>
> ### 1. **The SELB-Hybrid Strategy**
> To adapt our method for free-form tasks that lack explicit sections, we generalized the SELB strategy from section enforcement to **SELB-Hybrid** strategy, which focuses on length enforcement and stall prevention.
> - **Stop Token Suppression**: This is a direct generalization of our original proactive failure prevention logic. In the original method, we suppressed EOS tokens until the model reached the target section count ($P_{total}$). In the new strategy, the processor actively and aggressively suppresses all premature end-of-sequence (EOS) tokens and common conversational stop-phrases (e.g., I hope this helps, Let me know if you need...). This suppression remains active until the model reaches a preset target token count (e.g., 1.5 times the target word count $L$). This directly addresses the problem of models stopping prematurely before reaching the target length.
> - **Hybrid Keep-Alive Mechanism**: Without strong structural anchors such as chapter titles to boost, free-form generation faces a unique stall risk, such as falling into repetitive loops or silent failures (i.e., ceasing to generate new content). To solve this, we introduce a keep-alive mechanism: (1) The processor periodically (e.g., every $\tau_{max} = 500$ tokens) checks the generation state. (2) If the model fails to produce a natural interruption (like a period . or a newline \n) within a grace period (e.g., $k=100$ tokens), the processor determines the model has stalled. (3) It then nudges the model by boosting the logits of generic continuation tokens (e.g., \n and ), forcing it to break the current loop and continue generating new content.
>
> ### 2. **Evaluation on Free-Form Tasks**
> To rigorously test the SELB-Hybrid, we designed highly challenging free-form tasks. We require the model to generate a single, continuous document with a target length of 20,000 words, without providing any chapter or section prompts. We evaluated two such tasks from Longwriter and our benchmark:
> - **Novel-Writing Task (from longwriter)**: Write a {L}-word novel about a teenage heroine who grows up and ends up changing the world.
> - **Architecture-Design Task (revised from VOLTBench)**: Please design a multi-story building. Describe the function and layout of each floor. Ensure the entire description contains at least {L} words, with clarity and continuity throughout the document. Do not stop generating until all floors are described and the document is concluded with 'finished'.

---

> ### Author Response · Authors · 2025-11-20
> **(2/3) Responses by Authors**
>
> > **`Weakness 1`**: Scope & generality of SELB: The structural prior assumes explicit, chapter-style sectioning with recognizable title tokens; it’s unclear how well this transfers to non-templated long-form tasks (e.g., essays, reports without section tokens) or to multilingual tokenization quirks. Ablations on more free-form schemas are needed.
>
> `Response`: (Continued)
> ### 3. **Experimental Results**
> We compared our method against Qwen2.5-7B-Instruct (the base model), LongWriter-8B, and GPT-4o-mini on these free form tasks (**L=20000**). The detailed quantitative results are presented in Table 1 and Table 2:
>
> **Table1**. Performance comparison of evaluated models on free form task, conducted in English. Representative results are shown for novel writing task for 20k words from LongWriter [1]. For Length Standard Deviation (LSD), which measures the absolute volatility of output lengths, the values in parentheses provide context by indicating the generated mean length (in words).
> | Model | Length Volatility | | | Generation Quality |
> |-------|------------------|-|-|----------------------------|
> | | LSD (↓) | LVC (↓) | MLA (↑) | UCA (↑) |
> | Qwen2.5-7B | 7300 (24068) | 30.3% | 79.6% | 94.3% (±2.5%) |
> | LongWriter-8B | 355 (502) | 70.7% | 2.5% | 75.3% (±6.2%) |
> | GPT-4o-mini | 97.3 (447) | 21.7% | 2.2% | 82.7% (±4.9%) |
> | Ours | 2346 (19406) | **12.1%** | **97%** | **96.4%** (±2.9%) |
>
> **Results**: LongWriter-8B and GPT-4o-mini failed completely, generating an average of only 502 words and 447 words, respectively. The Qwen2.5-7B base model significantly overshot the target, generating 24,068 words with a high Length Volatility Coefficient (LVC) of 30.3%. Our method generated 19,406 words, extremely close to the 20,000-word target. It achieved a 97.0% Mean Length Adherence (MLA) and the lowest LVC (12.1%), while also maintaining the highest generation quality (UCA 96.4%).
>
> **Table2**. Performance comparison of evaluated models on free form task, conducted in English under simple difficulty set. Representative results are shown for Architecture-Design for 20k words.
> | Model | Length Volatility | | | Generation Quality |
> |-------|------------------|-|-|----------------------------|
> | | LSD (↓) | LVC (↓) | MLA (↑) | UCA (↑) |
> | Qwen2.5-7B | 6253 (15847) | 39.4% | 79.2% | 93.4% (±3.1%) |
> | LongWriter-8B | 1893 (7107) | 26.6% | 35.5% | 79.8% (±5.2%) |
> | GPT-4o-mini | 306 (631) | 48.6% | 3.2% | 84.8% (±4.2%) |
> | Ours | 2450 (21618) | **11.3%** | **91.9%** | **96.8%** (±2.5%) |
>
> **Results**: Baseline models again failed to meet the target, the Qwen2.5-7B base model generated only 15,847 words. Our method again demonstrated the best length control, generating 21,618 words. It achieved both the highest MLA (91.9%) and the lowest LVC (11.3%), while also achieving the highest UCA quality (96.8%).
>
> To provide a comprehensive view of free-form task performance, **we conducted experiments across a wide range of target lengths, including L=1k, 2k, 4k, 10k, 20k, and 40k words**. We have plotted these results as curves in the revised **Appendix G (see Figure 10, Section Performance on free form task)**. These curves clearly show that baseline models generally fail at controllability as the length requirement increases. Our method (Qwen2.5-7B+ours) closely tracks the $y=x$ reference line across the entire range from 1k to 40k words. **Our method also maintains a significantly smaller standard deviation, demonstrating superior and stable control over free-form generation**.
>
> In summary, current advanced models are uncontrollable and unreliable for free-form long-text generation tasks that lack structural guidance. They widely suffer from length collapse and high volatility. **The core idea of our SELB method (decoding-stage logits intervention) is highly generalizable**. By shifting the intervention logic from section enforcement to length enforcement and stall prevention, our method is the only strategy among those tested that can effectively enforce length constraints, drastically reduce volatility, and consistently maintain the highest generation quality on extreme-scale free-form tasks. We sincerely thank you for this feedback.
>
> [1] Bai Y, Zhang J, Lv X, et al. Longwriter: Unleashing 10,000+ word generation from long context llms.

---

> ### Author Response · Authors · 2025-11-20
> **(3/3) Responses by Authors**
>
> > **`Weakness 2`**: Fairness of baselines: Stronger inference-time baselines (e.g., length controllers, look-ahead/ITR, stop-entropy controls, repetition penalties tuned to parity) could be included to ensure SELB’s gains aren’t mostly from adding any simple structure prior.
>
> `Response`: Thank you for the valuable comment. Following your suggestion, **we implemented all of the suggested strategies on the same base model (Qwen2.5-7B-Instruction) and evaluated them on the challenging 100-section generation task**. As Inference-Time Reasoning (ITR) fundamentally alters the model's reasoning paradigm rather than its native generation capability, we did not include it in our evaluation.
>
> Table 1. Performance comparison of evaluated models on a 100-section generation task, conducted in English under simple difficulty settings. Representative results are shown for an unstructured task (Story) and a structured task (Code Function).
> | Model                | LSD (↓)            | LVC (↓)  | MLA (↑) | FAD (↓)           | SCA (↑)           | UCA (↑)           |
> |----------------------|--------------------|----------|---------|-------------------|-------------------|-------------------|
> | GPT-4o mini          | 325.65 (959)       | 33.9%    | 4.8%    | 1.41 (7.00)       | 84.6% (±30.8%)    | 86.7% (±6.7%)     |
> | Deepseek-R1          | 103.30 (1198)      | 8.6%     | 6.0%    | 1.25 (4.33)       | 35.0% (±13.2%)    | 93.3% (±3.7%)     |
> | Qwen2.5-1.5B         | 27.78 (142)        | 19.6%    | 0.7%    | 0.47 (1.67)       | 15.6% (±24.0%)    | 84.0% (±7.1%)     |
> | Llama3.1-8B          | 92.77 (350)        | 26.5%    | 1.7%    | 0.94 (4.33)       | 92.4% (±14.2%)    | 82.0% (±18.9%)    |
> | LongWriter-8B        | 2866.3 (6320)      | 45.4%    | 31.6%   | 21.42 (45.00)     | 32.6% (±31.9%)    | 66.7% (±16.5%)    |
> | Qwen2.5-7B           | 75.87 (445)        | 17.0%    | 2.2%    | 2.05 (10.33)      | 99.8% (±0.4%)     | 86.7% (±7.6%)     |
> | **Repetition_penalties**   | 553 (2967)         | 18.6%    | 14.8%   | 5.4 (22)          | 98% (±1%)         | 76.7% (±14.5%)    |
> | **Stop-entropy**   | 713 (2701)         | 26.4%    | 13.5%   | 7.24 (24)         | 95% (±2.5%)       | 83.9% (±8%)       |
> | **Length_controllers** | 1280 (4470)        | 28.65%   | 77.7%   | 9.2 (28)          | 96% (±2%)         | 85% (±9%)         |
> | **Look-ahead**   | 268 (2883)         | 9.3%     | 14.4%   | 7.2 (25)          | 94% (±3.5%)       | 84.4% (±8%)       |
> | **Ours**  | 2194.23 (15651) | 14.02% | 78.25% | 7.24 (88.00) | 100% (±0%) | 86.7% (±16.5%) |
>
> The results, presented in the table above, reveal a clear hierarchy of performance:
> - **Baselines are effective but limited**: The suggested inference-time baselines indeed improve upon the vanilla model. For instance, Length_controllers increased the output from ~445 words to ~4470 words, and section counts rose from ~10 to ~28. This validates that these methods alleviate some instability.
> - **SELB achieves significant performance gains**: However, these heuristics hit a distinct performance ceiling. None of the baselines could sustain generation beyond 30 sections or 5,000 words. In sharp contrast, SELB generated an average of 15,651 words and 88 sections, effectively completing the majority of the 100-section task.
>
> This comparison demonstrates that SELB's performance gains are not merely due to adding any simple structure prior. While standard penalties and controllers can delay model collapse, they cannot sustain the long-range coherence required for extreme-length tasks. SELB's active logits boosting provides the necessary structural guidance that simple heuristics lack. We have included these new experiments and corresponding analyses in the Ablation Study section of the revised manuscript.

---

> ### Author Response · Authors · 2025-11-26
>
> Dear Reviewer fkNi,
>
> Thank you once again for your thoughtful feedback on our paper. We appreciate the time and effort you have invested in reviewing our work.
>
> As we are nearing the end of the rebuttal period, we wanted to kindly follow up to ensure that you’ve had a chance to review our response to your comments. We hope that our rebuttal has addressed your concerns satisfactorily. If there are any additional points or further clarifications required, we would be happy to provide them promptly.
>
> Additionally, if our response has sufficiently addressed your concerns, we would greatly appreciate if you could consider reflecting this in your evaluation, including revisiting your score if appropriate.
>
> Thank you for your time and understanding. We greatly value your feedback and look forward to hearing from you.
>
> Best regards,
>
> Authors of Paper Submission 8616

---

> ### Author Response · Authors · 2025-11-28
>
> Dear Reviewer fkNi,
>
> Thank you once again for your valuable time and thoughtful feedback on our paper. We broadly appreciate the effort you have invested in helping us improve our work.
>
> We are writing to kindly inform you that **we have updated the manuscript to address potential concerns regarding the generalization of our method. Specifically, we have added Section 6.4 and Appendix I** (highlighted in blue for your convenience).
>
> In these new sections, we introduce the SELB-Hybrid strategy and extend our evaluation to free-form generation tasks. We hope these additional experiments satisfactorily address your concerns. If there are any remaining questions or areas where further clarification would be helpful, we would be more than happy to address them promptly.
>
> Thank you again for your time and consideration.
>
> Best regards,
>
> Authors of Paper Submission 8616

---

### Comment · Area_Chair_iP9g · 2025-11-28

Dear Reviewers,

The authors have responded to your reviews. Please engage in the discussion and evaluate the authors’ rebuttal to check whether your comments have been adequately addressed, and determine whether you would like to adjust your evaluations.

Best,

Your AC

---

### Author Response · Authors · 2025-11-28
**Common Response**

Dear SAC, AC, and Reviewers,

Thank you for your valuable feedback and the time dedicated to reviewing our paper. We have carefully considered your comments and have implemented revisions that we believe significantly enhance the breadth and contribution of our work.

In response to the feedback regarding the generalization capability of our method, we have updated the manuscript to include a new **Section 6.4** and **Appendix I** (highlighted in **blue**). These additions introduce the **SELB-Hybrid** strategy, which is specifically adapted to enforce stability in unstructured, open-ended generation scenarios.

We remain fully available to address any further questions or provide additional clarifications regarding these updates.

Many thanks!

Authors of Paper Submission 8616

---

### Author Response · Authors · 2025-12-01
**Summary of Rebuttals & Major Updates**

Dear Program Chairs, Senior Area Chairs, Area Chairs and Reviewers:

We thank the reviewers for their thoughtful comments and constructive feedback. We are encouraged that all reviewers recognized the value and novelty of our work. Specifically, we would like to highlight the following strengths acknowledged by the reviewers:

1. `Novelty and Importance of the Problem (Output Volatility)`: **Reviewers 4xJJ** and **fkNi** commended the *"value of problem definition,"* noting that output volatility is a *"critical but often overlooked"* issue. **Reviewer Kq9F** found the work *"refreshingly practical"* for establishing stability as a first-class objective. **Reviewer 4xJJ** further emphasized that solving this is crucial for *"cost-control and system reliability."*
2. `Comprehensive Benchmark Design (Our VOLTBench)`: The design of VOLTBench was widely praised. **Reviewer dTnx** highlighted it as a *"well-designed"* and *"robust"* benchmark covering both structured and unstructured tasks. **Reviewer 4xJJ** recognized it as a *"solid contribution"* that is *"more comprehensive and rigorous"* than existing benchmarks, while **Reviewer fkNi** appreciated the *"heterogeneous tasks"* and *"dual languages"* setup.
3. `Insightful Analysis and Effective Method (Our SELB)`: **Reviewers Kq9F** and **4xJJ** praised the *"interpretable diagnostic"* and *"intuitive probing"* of the attention mechanism. Furthermore, the proposed SELB method was recognized as a *"simple plug-in"* solution (**Reviewer Kq9F**) with *"strong empirical support"* (**Reviewer dTnx**) and *"evident empirical gains"* (**Reviewer fkNi**), achieving significant volatility reduction without retraining.

### Summary of Major Updates
In response to the constructive feedback, we have significantly revised the paper. The major updates include:

1. `Extended Evaluation on External Benchmarks`: We extended our evaluation to established benchmarks, LongBench-Write (LongWriter) and WritingBench. Our method achieved a LongBench-Write **Overall Score ($S$) of 85.3** on LongWriter (surpassing LongWriter-9B-DPO) and improved the WritingBench score from **7.43 to 8.43**, demonstrating effectiveness beyond VOLTBench (Addressing **Reviewer Kq9F**).
2. `Adaptation to Unstructured/Free-Form Tasks`: We demonstrated that the core mechanism of our method is structure-agnostic. Specifically, by simply shifting the intervention target from explicit section headers to dynamic flow monitoring (an adaptation we term SELB-Hybrid), we achieved **97% Mean Length Adherence on a 20k-word novel task** where baselines collapsed at <500 words (Addressing **Reviewer dTnx, Kq9F, 4xJJ, fkNi**; see newly added Section 6.4 & Appendix I).
3. `In-depth Quality Analysis`: We integrated fine-grained quantitative metrics (N-gram repetition rates, Type-Token Ratio) and representation drift analysis. Results show SELB reduces 3-gram repetition from **~69% (Base) to <4%** and triples the Type-Token Ratio, verifying it improves coherence without inducing filler content (Addressing **Reviewer dTnx, 4xJJ,  Kq9F**; see newly added Appendix H).
4. `Strengthened Baselines`: We added comparisons with advanced inference-time baselines, including Repetition Penalties, Stop-Entropy, and Length Controllers. Experiments confirm that while these heuristics fail to sustain generation beyond **~5k words**, SELB consistently completes **100-section tasks (~15k words) with superior stability** (Addressing **Reviewer fkNi**; see revised Ablation Study).

We have engaged with all feedback, conducting substantial new experiments to directly resolve the reviewers' concerns. We are particularly grateful to **Reviewer Kq9F** for their active engagement and for confirming that our response has *”addressed most of their concerns”*. While we have not received further responses from some reviewers, we believe the extensive new empirical evidence, particularly regarding generalizability, firmly establishes the robustness of our approach.

We deeply appreciate the significant effort and oversight provided by the Area Chair, Senior Area Chair, and Program Chairs during this demanding phase. We sincerely hope that the concrete improvements made during the rebuttal, validated by the positive feedback from Reviewer Kq9F, will weigh positively in your final assessment. Thank you all for your time and dedication to the review process.

Best regards,

Authors of Paper Submission 8616

---

### Meta-Review · Area_Chair_vRcy · 2026-01-11

**Summary:**

The paper try to address the length volatility in long-form generation by proposing several methods. The reviewers raised concerns regarding the generalizability and technical nature of the proposed solution. The primary criticism is that SELB functions as a brute-force patch or rule-based intervention that relies heavily on task-specific structural markers and manually preset parameters. Reviewers noted a potential collusion between the benchmark and the method.

**Reviewer Concerns:**

Addressed:
The concern regarding repetition/quality degradation by providing n-gram and TTR results. They also provided a preliminary extension to free-form tasks, showing it can be adapted to tasks without explicit chapters.

Outstanding Concerns:
While the rebuttal showed the method can work on free-form tasks, it still requires active, manual intervention in the decoding loop. This makes the method more of an inference-time engineering trick than a robust scientific solution.
The fairness of baselines was not fully resolved: the paper compares a highly-tuned, constraint-aware decoding strategy against vanilla models without similar inference-time optimizations, making the performance gap less surprising and potentially overstating the contribution.

**Reviewer Scores:**

The reviewers will most likely to retain their scores as few reviewer posted responses.

---

### Decision · Program_Chairs · 2026-01-26

Reject